# Universal Rates of Empirical Risk Minimization

**Steve Hanneke**
Department of Computer Science
Purdue University
`steve.hanneke@gmail.com`

**Mingyue Xu**
Department of Computer Science
Purdue University
`xu1864@purdue.edu`

## Abstract

The well-known *empirical risk minimization* (ERM) principle is the basis of many widely used machine learning algorithms, and plays an essential role in the classical PAC theory. A common description of a learning algorithm's performance is its so-called "learning curve", that is, the decay of the expected error as a function of the input sample size. As the PAC model fails to explain the behavior of learning curves, recent research has explored an alternative universal learning model and has ultimately revealed a distinction between optimal universal and uniform learning rates (Bousquet et al., 2021). However, a basic understanding of such differences with a particular focus on the ERM principle has yet to be developed.

In this paper, we consider the problem of universal learning by ERM in the realizable case and study the possible universal rates. Our main result is a fundamental *tetrachotomy*: there are only four possible universal learning rates by ERM, namely, the learning curves of any concept class learnable by ERM decay either at $e^{-n}$, $1/n$, $\log(n)/n$, or arbitrarily slow rates. Moreover, we provide a complete characterization of which concept classes fall into each of these categories, via new complexity structures. We also develop new combinatorial dimensions which supply sharp asymptotically-valid constant factors for these rates, whenever possible.

## 1 Introduction

The classical statistical learning theory mainly focuses on the celebrated PAC (Probably Approximately Correct) model (Vapnik and Chervonenkis, 1974; Valiant, 1984) with emphasis on supervised learning. A particular setting therein, called the *realizable* case, has been extensively studied. Complemented by the "no-free-lunch" theorem (Antos and Lugosi, 1996), the PAC framework, which adopts a minimax perspective, can only explain the best *worst-case* learning rate by a learning algorithm over all realizable distributions. Such learning rates are thus also called the *uniform* rates. However, the uniform rates can only capture the upper envelope of all learning curves, and are too coarse to explain practical machine learning performance. This is because real-world data is rarely worst-case, and the data source is typically fixed in a given learning scenario. Indeed, Cohn and Tesauro (1990, 1992) observed from experiments that practical learning rates can be much faster than is predicted by PAC theory. Moreover, many theoretical works (Schuurmans, 1997; Koltchinskii and Beznosova, 2005; Audibert and Tsybakov, 2007, etc.) were able to prove faster-than-uniform rates for certain learning problems, though requiring additional modelling assumptions. To distinguish from the uniform rates, these rates are named the *universal* rates and was formalized recently by Bousquet et al. (2021) via a distribution-dependent framework. Unlike the simple *dichotomy* of the optimal uniform rates: every concept class $\mathcal{H}$ has a uniform rate being either linear $\mathrm{VC}(\mathcal{H})/n$ or "bounded away from zero", the optimal universal rates are captured by a *trichotomy*: every concept class $\mathcal{H}$ has a universal rate being either exponential, linear or arbitrarily slow (see Thm.1.6 Bousquet et al., 2021).

In supervised learning, a family of successful learners called the *empirical risk minimization* (ERM) consist of all learning algorithms that output a sample-consistent classifier. In other words, an ERM

38th Conference on Neural Information Processing Systems (NeurIPS 2024).

algorithm is any learning rule, which outputs a concept in $\mathcal{H}$ that minimizes the empirical error (see Appendix A for a formal definition). For notation simplicity, we first introduce

**Definition 1** (**Version space, Mitchell, 1977**). *Let $\mathcal{H}$ be a concept class and $S_n := \{(x_i, y_i)\}_{i=1}^n$ be a dataset, the version space induced by $S_n$, denoted by $V_{S_n}(\mathcal{H})$ (or $V_n(\mathcal{H})$ for short), is defined as $V_{S_n}(\mathcal{H}) := \{\overline{h \in \mathcal{H} : h(x_i) = y_i, \forall i \in [n]}\}$.*

Now given labeled samples $S_n := \{(x_i, y_i)\}_{i=1}^n$, an ERM algorithm is any learning algorithm that outputs a concept in the sample-induced *version space*, that is, a sequence of universally measurable functions $\mathcal{A}_n : S_n \to \hat{h}_n \in V_{S_n}(\mathcal{H}), n \in \mathbb{N}$. Throughout this paper, we will simply denote an ERM algorithm by its output predictors $\{\hat{h}_n\}_{n \in \mathbb{N}}$.

It is well-known that the ERM principle plays an important role in understanding general uniform learnability: a concept class is uniformly learnable if and only if it can be learned by ERM. However, while the optimal $\text{VC}(\mathcal{H})/n$ rate is achievable by some improper learner (Hanneke, 2016a), ERM algorithms can at best achieve a uniform rate of $(\text{VC}(\mathcal{H})/n) \log (n/\text{VC}(\mathcal{H}))$. Moreover, such a gap has been shown to be unavoidable in general (Auer and Ortner, 2007), which leaves a challenging question to study: what are the sufficient and necessary conditions on $\mathcal{H}$ for the entire family of ERM algorithms to achieve the optimal error? Indeed, many subsequent works have devoted to improving the logarithmic factor in specific scenarios. The work of Giné and Koltchinskii (2006) refined the bound by replacing $\log (n/\text{VC}(\mathcal{H}))$ with $\log (\theta(\text{VC}(\mathcal{H})/n))$, where $\theta(\cdot)$ is called the *disagreement coefficient*. Based on this, Hanneke and Yang (2015) proposed a new data-dependent bound with $\log (\hat{n}_{1:n}/\text{VC}(\mathcal{H}))$, where $\hat{n}_{1:n}$ is a quantity related to the *version space compression set size* (a.k.a. the *empirical teaching dimension*). As a milestone, the work of Hanneke (2016b) proved an upper bound $(\text{VC}(\mathcal{H})/n) \log (\mathfrak{s}_{\mathcal{H}}/\text{VC}(\mathcal{H}))$ and a lower bound $(\text{VC}(\mathcal{H}) + \log (\mathfrak{s}_{\mathcal{H}}))/n$, where $\mathfrak{s}_{\mathcal{H}}$ is called the *star number* of $\mathcal{H}$ (see Definition 4 in Section 2). Though not quite matching, these two bounds together yield an optimal linear rate when $\mathfrak{s}_{\mathcal{H}} < \infty$. Thereafter, the uniform rates by ERM can be described as a *trichotomy*, namely, every concept class $\mathcal{H}$ has a uniform rate by ERM being exactly one of the following: $1/n$, $\log (n)/n$ and "bounded away from zero".

From a practical perspective, many ERM-based algorithms are designed and are widely applied in different areas of machine learning, such as the logistic regression and SVM, the CAL algorithm in active learning, the gradient descent (GD) algorithm in deep learning. Since the worst-case nature of the PAC model is too pessimistic to reflect the practice of machine learning, understanding the distribution-dependent performance of ERM algorithms is of great significance. However, unlike that a distinction between the optimal uniform and universal rates has been fully understood, how fast universal learning can outperform uniform learning in particular by ERM remains unclear. Furthermore, we are lacking a complete theory to the characterization of the universal rates by ERM, though certain specific scenarios that admit faster rates by ERM have been discovered (Schuurmans, 1997; van Handel, 2013). In this paper, we aim to answer the following fundamental question:

**Question 1.** *Given a concept class $\mathcal{H}$, what are the possible rates at which $\mathcal{H}$ can be universally learned by ERM?*

We start with some basic preliminaries of this paper. We consider an *instance space* $\mathcal{X}$ and a *concept class* $\mathcal{H} \subseteq \{0, 1\}^{\mathcal{X}}$. Given a probability distribution $P$ on $\mathcal{X} \times \{0, 1\}$, the *error rate* of a classifier $h : \mathcal{X} \to \{0, 1\}$ is defined as $\text{er}_P(h) := P((x, y) \in \mathcal{X} \times \{0, 1\} : h(x) \neq y)$. A distribution $P$ is called *realizable* with respect to $\mathcal{H}$, denoted by $P \in \text{RE}(\mathcal{H})$, if $\inf_{h \in \mathcal{H}} \text{er}_P(h) = 0$. Note that in this definition, $h^*$ satisfying $\text{er}_P(h^*) = \inf_{h \in \mathcal{H}} \text{er}_P(h)$ is called the *target concept* of the learning problem, and is not necessary in $\mathcal{H}$. We may also say that $P$ is a realizable distribution centered at $h^*$. Given an integer $n$, we denote by $S_n := \{(x_i, y_i)\}_{i=1}^n \sim P^n$ a i.i.d. $P$-distributed dataset. In the universal learning framework, the performance of a learning algorithm is commonly measured by its *learning curve* (Bousquet et al., 2021; Hanneke et al., 2022; Bousquet et al., 2023), that is, the decay of the *expected error rate* $\mathbb{E}[\text{er}_P(\hat{h}_n)]$ as a function of sample size $n$. With these settings settled, we are now able to formalize the problem of universal learning by ERM.

**Definition 2** (**Universal learning by ERM**). *Let $\mathcal{H}$ be a concept class, and $R(n) \to 0$ be a rate function. We say*

- *$\mathcal{H}$ is underline{universally learnable at rate $R$ by ERM}, if for every distribution $P \in \text{RE}(\mathcal{H})$, there exist parameters $C, c > 0$ such that for every ERM algorithm, $\mathbb{E}[\text{er}_P(\hat{h}_n)] \leq CR(cn)$, for all $n \in \mathbb{N}$.*

- $\mathcal{H}$ is not universally learnable at rate faster than $R$ by ERM, if there exists a distribution $P \in RE(\mathcal{H})$ and parameters $C, c > 0$ such that there exists an ERM algorithm satisfying $\mathbb{E}[er_P(\hat{h}_n)] \geq CR(cn)$, for infinitely many $n \in \mathbb{N}$.
- $\mathcal{H}$ is universally learnable with exact rate $R$ by ERM, if $\mathcal{H}$ is universally learnable at rate $R$ by ERM, and is not universally learnable at rate faster than $R$ by ERM.
- $\mathcal{H}$ requires arbitrarily slow rates to be universally learned by ERM, if for every rate function $R(n) \to 0$, $\mathcal{H}$ is not universally learnable at rate faster than $R$ by ERM.

**Remark 1.** *The above definition inherits the structure of the definition to the optimal universal learning (Definition 1.4 Bousquet et al., 2021). Here, we are actually considering the "worst-case" ERM algorithm, which is consistent with the PAC theory. A crucial difference between this definition and the PAC one is that here the constants $C, c > 0$ are allowed to depend on the distribution $P$. In other words, the PAC model can be defined similarly, but requires uniform constants $C, c > 0$. Consequently, $\mathcal{H}$ is universally learnable at rate $R$ by ERM if it is PAC learnable at rate $R$ by ERM.*

**Remark 2.** *It is not hard to see that the error rate achieved by any ERM algorithm, given $S_n \sim P^n$ as input, is at most $\sup_{h \in V_{S_n}(\mathcal{H})} er_P(h)$. Furthermore, for any distribution $P \in RE(\mathcal{H})$, there exist ERM algorithms obtaining error rates arbitrarily close to this value. Hence, to obtain upper bounds of the universal rates by ERM, it requires us to bound the random variable $\sup_{h \in V_{S_n}(\mathcal{H})} er_P(h)$, where a common technique is to bound $\mathbb{P}(\sup_{h \in V_{S_n}(\mathcal{H})} er_P(h) > \epsilon) = \mathbb{P}(\exists h \in V_{S_n}(\mathcal{H}) : er_P(h) > \epsilon)$. To obtain lower bounds, it requires us to construct specific "hard" distributions.*

## 1.1 Basic examples

In order to develop some initial intuition for what universal learning rates are possible for ERM, we first introduce several basic examples that illustrate the possibilities in the following Section 1.2. To convince the reader, we provide direct analysis (without using our characterization in Section 1.2) for those examples (see details in Appendix B.1).

**Example 1** ($e^{-n}$ **learning rate**). *Any finite class $\mathcal{H}$ is universally learnable at exponential rate (Schuurmans, 1997). Indeed, according to their analysis, such exponential rates can also be achieved by any ERM algorithm.*

**Example 2** ($1/n$ **learning rate**). *Let $\mathcal{H}_{thresh,\mathbb{N}} := \{h_t : t \in \mathbb{N}\}$ be the class of all threshold classifiers on natural numbers, where $h_t(x) := \mathbb{1}(x \geq t)$, for all $x \in \mathbb{N}$. $\mathcal{H}_{thresh,\mathbb{N}}$ is universally learnable at exponential rate since this class does not have an infinite Littlestone tree (Bousquet et al., 2021). However, ERM algorithms cannot guarantee such exponential rates but at best linear rates, when encountering certain realizable distributions centered at the target concept $h_{all\text{-}0's}$, which is the function that labels zero everywhere (see Appendix A).*

**Example 3** ($\log(n)/n$ **learning rate**). *Let $\mathcal{X} = \mathbb{N}$ and $\mathcal{H}_{singleton,\mathbb{N}} := \{h_t : t \in \mathcal{X}\}$ be the class of all singletons on $\mathcal{X}$, where $h_t(x) := \mathbb{1}(x = t)$, for all $x \in \mathbb{N}$. It is clear that $VC(\mathcal{H}_{singleton,\mathbb{N}}) = 1$. Note that $\mathcal{H}_{singleton,\mathbb{N}}$ is universally learnable at exponential rate since it has finite Littlestone dimension $LD(\mathcal{H}_{singleton,\mathbb{N}}) = 1$. However, the exact universal rate by ERM is instead $\log(n)/n$. This is because $\mathcal{H}_{singleton,\mathbb{N}}$ admits certain realizable distributions centered at $h_{all\text{-}0's}$. Indeed, it is an example where the universal rate by ERM matches the uniform rate, up to a distribution-dependent constant.*

**Example 4** (**Arbitrarily slow learning rate**). *Let $\mathcal{X} = \bigcup_{i \in \mathbb{N}} \mathcal{X}_i$ be the disjoint union of finite sets with $|\mathcal{X}_i| = 2^i$, for all $i \in \mathbb{N}$. For each $i \in \mathbb{N}$, let $\mathcal{H}_i := \{h_S := \mathbb{1}_S : S \subseteq \mathcal{X}_i, |S| \geq 2^{i-1}\}$ and consider the concept class $\mathcal{H} = \bigcup_{i \in \mathbb{N}} \mathcal{H}_i$. $\mathcal{H}$ is universally learnable at exponential rate since it does not have an infinite Littlestone tree. However, a bad ERM algorithm can perform arbitrarily slowly.*

**Example 5** (**Not Glivenko-Cantelli but learnable by ERM**). *Let $\mathcal{X} = [0, 1]$, $\mathcal{H} := \{\mathbb{1}_S : S \subset \mathcal{X}, |S| < \infty\}$, and $P$ be the uniform (Lebesgue) distribution on $[0, 1]$. $\mathcal{H}$ is universally learnable at exponential rate (no infinite Littlestone tree). Moreover, $\mathcal{H}$ is not a universal Glivenko-Cantelli class for $P$ (van Handel, 2013), but is still universally learnable by ERM. However, if we consider the class $\mathcal{H} \cup \{h_{all\text{-}1's}\}$, which is still not a universal Glivenko-Cantelli class for $P$, but no longer universally learnable by any ERM algorithm since $er_P(\hat{h}_n) = 1$ regardless of the sample size.*

The above examples indicate that the cases of universal learning by ERM do not match the uniform learning, but contains at least five possible cases: every nontrivial concept class $\mathcal{H}$ is either universally learnable at exponential rate (but not faster), or is universally learnable at linear rate (but not faster), or is universally learnable at slightly slower than linear rate $\log(n)/n$ (but not faster), or is universally

learnable but necessarily with arbitrarily slow rates, or is not universally learnable at all. Throughout this paper, we only consider the case where the given concept class is universally learnable by ERM. We leave it an open question whether there exists a nice characterization that determines the universal learnability by ERM.

## 1.2 Main results

In this section, we summarize the main results of this paper. The examples in Section 1.1 reveal that there are at least four possible universal rates by ERM. Interestingly, we find that these are also the only possibilities. The following two theorems consist of the main results of this work. In particular, Theorem 1 gives out a complete answer to Question 1. It expresses a fundamental *tetrachotomy*: there are exactly four possibilities for the universal learning rates by ERM: being either exponential, or linear, or $\log(n)/n$, or arbitrarily slow rates. Moreover, Theorem 2 specifies the answer by pointing out for what realizable distributions (targets), those universal rates are sharp.

**Theorem 1** (**Universal rates for ERM**). *For every class $\mathcal{H}$ with $|\mathcal{H}| \geq 3$, the following hold:*

- *$\mathcal{H}$ is universally learnable by ERM with exact rate $e^{-n}$ if and only if $|\mathcal{H}| < \infty$.*
- *$\mathcal{H}$ is universally learnable by ERM with exact rate $1/n$ if and only if $|\mathcal{H}| = \infty$ and $\mathcal{H}$ does not have an infinite star-eluder sequence.*
- *$\mathcal{H}$ is universally learnable by ERM with exact rate $\log(n)/n$ if and only if $\mathcal{H}$ has an infinite star-eluder sequence and $VC(\mathcal{H}) < \infty$.*
- *$\mathcal{H}$ requires at least arbitrarily slow rates to be learned by ERM if and only if $VC(\mathcal{H}) = \infty$.*

**Remark 3.** *The formal definition of the star-eluder sequence can be found in Section 2. Unlike the separation between exact $e^{-n}$ and $1/n$ rates is determined by the cardinality of the class, and the separation between exact $\log(n)/n$ and arbitrarily slow rates is determined by the VC dimension of the class, whether there exists a simple combinatorial quantity that determines the separation between exact $1/n$ and $\log(n)/n$ rates is unclear and might be an interesting direction for future work. We thought that it is likely the star number $\mathfrak{s}_{\mathcal{H}}$ (Definition 4) is the correct characterization here, but it turns out not unfortunately (see details in Section 4 and Appendix B.3).*

Based on Theorem 1, a distinction between the performance of ERM algorithms and the optimal universal learning algorithms can be revealed, which we present in the following table (the required definitions in "Case" are deferred to Section 2, and examples can be found in Appendix B.2).

| Optimal rate | Exact rate by ERM | Case | Example |
|---|---|---|---|
| $e^{-n}$ | $1/n$ | infinite eluder sequence but no infinite Littlestone tree | Example 12 |
| $e^{-n}$ | $\log(n)/n$ | infinite star-eluder sequence but no infinite Littlestone tree | Example 13 |
| $e^{-n}$ | arbitrarily slow | infinite VC-eluder sequence but no infinite Littlestone tree | Example 15 |
| $1/n$ | $\log(n)/n$ | infinite star-eluder sequence but no infinite VCL tree | Example 14 |
| $1/n$ | arbitrarily slow | infinite VC-eluder sequence but no infinite VCL tree | Example 16 |

Furthermore, the distinction between the universal rates and the uniform rates by ERM can also be fully captured, and are depicted schematically in Figure 1 as an analogy to the Fig.4 of Bousquet et al. (2021). Besides the examples in Section 1.1, we also need the following additional example concerning the Littlestone dimension to appear in the diagram.

**Example 6** ($\log(n)/n$ **learning rate and unbounded Littlestone dimension**). *We consider here the class of two-dimensional halfspaces, that is, $\mathcal{X} := \mathbb{R}^2$ and $\mathcal{H}_{halfspaces,\mathbb{R}} := \{\mathbb{1}(\boldsymbol{w} \cdot \boldsymbol{x} + b \geq 0) : \boldsymbol{w} \in \mathbb{R}^2, b \in \mathbb{R}\}$. It is a classical fact that for any integer $d$, the class of halfspaces on $\mathbb{R}^d$ has a finte VC dimension $d$, but has an infinite Littlestone tree, and thus having unbounded Littlestone dimension (Shalev-Shwartz and Ben-David, 2014). Finally, to show that this class is universally learnable by ERM at exact $\log(n)/n$ rate, we simply consider the subspace $\mathbb{S}^1 \subset \mathcal{X}$, this is indeed an infinite star set of $\mathcal{H}_{halfspaces,\mathbb{R}}$ centered at $h_{all-0's}$ and thus an infinite star-eluder sequence.*

As a complement to Theorem 1, the following Theorem 2 gives out target-specified universal rates. We say a target concept $h^*$ is universally learnable by ERM with exact rate $R$ if all realizable distribution $P$ considered in Definition 2 are centered at $h^*$. In other words, $\mathcal{H}$ is universally learnable with exact rate $R$ is equivalent to say all realizable target concepts are universally learnable with exact rate $R$. Concretely, for each of the four possible rates stated in Theorem 1, Theorem 2 specifies the target concepts that can be learned at such exact rate by ERM.

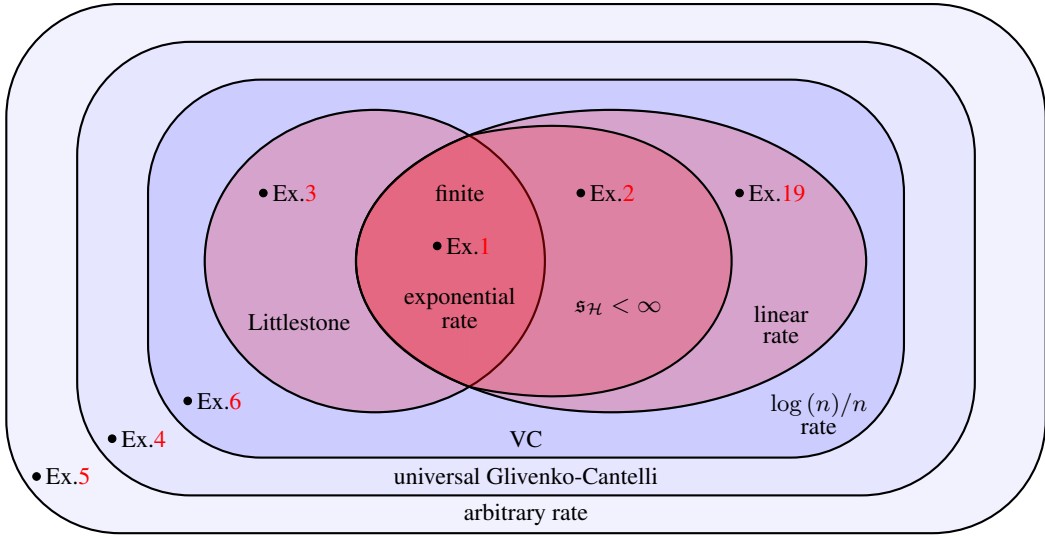

Figure 1: A venn diagram depicting the tetrachotomy of the universal rates by ERM and its relation with the uniform rates characterized by the VC dimension and the star number.

**Theorem 2** (**Target specified universal rates**). *For every class $\mathcal{H}$ with $|\mathcal{H}| \geq 3$, and a target concept $h^*$, the following hold:*

- *$h^*$ is universally learnable by ERM with exact rate $e^{-n}$ if and only if $\mathcal{H}$ does not have an infinite eluder sequence centered at $h^*$.*
- *$h^*$ is universally learnable by ERM with exact rate $1/n$ if and only if $\mathcal{H}$ has an infinite eluder sequence centered at $h^*$, but does not have an infinite star-eluder sequence centered at $h^*$.*
- *$h^*$ is universally learnable by ERM with exact rate $\log(n)/n$ if and only if $\mathcal{H}$ has an infinite star-eluder sequence centered at $h^*$, but does not have an infinite VC-eluder sequence centered at $h^*$.*
- *$h^*$ requires at least arbitrarily slow rates to be universally learned by ERM if and only if $\mathcal{H}$ has an infinite VC-eluder sequence centered at $h^*$.*

All detailed proofs appear in Appendix D. We also provide a brief overview of the main idea of each proof as well as some related concepts in Section 2.

An additional part of this work presents a *fine-grained analysis* (Bousquet et al., 2023) of the universal rates by ERM, which complements the *coarse rates* used in Theorem 1. Concretely, we provide a characterization of sharp distribution-free constant factors of the ERM universal rates, whenever possible. The characterization is based on two newly-developed combinatorial dimensions, called the *star-eluder dimension* (or SE dimension) and the *VC-eluder dimension* (or VCE dimension) (Definition 9). We say "whenever possible" because distribution-free constants are unavailable for certain cases (Remark 16). Such a characterization can also be considered as a refinement to the classical PAC theory, in a sense that it is sometimes better but only asymptotically-valid. Due to space limitation, we defer the definition of fine-grained rates and related results to Appendix C.

### 1.3   Related works

**PAC learning by ERM.** The performance of consistent learning rules (including the ERM algorithm) in the PAC (distribution-free) framework has been extensively studied. For VC classes, Blumer et al. (1989) gave out a $\log(n)/n$ upper bound of the uniform learning rate. Despite the well-known equivalence between uniform learnability and uniform learnability by the ERM principle (Vapnik and Chervonenkis, 1971), the best upper bounds for general ERM algorithms differ from the optimal sample complexity by an unavoidable logarithmic factor (Auer and Ortner, 2007). By analyzing the *disagreement coefficient* of the version space, the work of Giné and Koltchinskii (2006); Hanneke (2009) refined the logarithmic factor in certain scenarios. Furthermore, not only being a relevant

measure in the context of active learning (Cohn et al., 1994; El-Yaniv and Wiener, 2012; Hanneke, 2011, 2014), the *region of disagreement* of the version space was found out to have an interpretation of *sample compression scheme* with its size known as the *version space compression set size* (Wiener et al., 2015; Hanneke and Yang, 2015). Based on this, the label complexity of the CAL algorithm can be converted into a bound on the error rates of all consistent PAC learners (Hanneke, 2016b). Finally, Hanneke and Yang (2015); Hanneke (2016b) introduced a simple combinatorial quantity named the *star number*, and guaranteed that a concept class with finite star number can be uniformly learned at linear rate.

**Universal Learning.** Observed from empirical experiments, the actual learning rates on real-world data can be much faster than the one described by the PAC theory (Cohn and Tesauro, 1990, 1992). The work of Benedek and Itai (1988) considered a setting lies in between the PAC setting and the universal setting called *nonuniform learning*, in which the learning rate may depend on the target concept but still uniform over the marginal distributions. After that, Schuurmans (1997) studied classes of *concept chains* and revealed a distinction between exponential and linear rates along with a theoretical guarantee. Later, more improved learning rates have been obtained for various practical learning algorithms such as stochastic gradient decent and kernel methods (Koltchinskii and Beznosova, 2005; Audibert and Tsybakov, 2007; Pillaud-Vivien et al., 2018, etc.). Additionally, van Handel (2013) studied the uniform convergence property from a universal perspective, and proposed the *universal Glivenko-Cantelli property*. Until very recently, the universal (distribution-dependent) learning framework was formalized by Bousquet et al. (2021), in which a complete theory of the (optimal) universal learnability was obtained as well. After that, Bousquet et al. (2023) carried out a fine-grained analysis on the "distribution-free tail" of the universal *learning curves* by characterizing the optimal constant factor. As generalizations, Kalavasis et al. (2022); Hanneke et al. (2023) studied the universal rates for multiclass classification, and Hanneke et al. (2022) studied the universal learning rates under an interactive learning setting.

## 2 Technical overview

In this section, we discuss some technical aspects in the derivation of our main results in Section 1.2. Our analysis to the universal learning rates by ERM is based on three new types of complexity structures named the *eluder sequence*, the *star-eluder sequence* and the *VC-eluder sequence*. More details can be found in Sections 3-4 and all technical proofs are deferred to Appendix D.

**Definition 3** (**Realizable data**). *Let $\mathcal{H}$ be a concept class on an instance space $\mathcal{X}$, we say that a (finite or infinite) data sequence $\{(x_1, y_1), (x_2, y_2), \ldots\} \in (\mathcal{X} \times \{0, 1\})^{\infty}$ is realizable (with respect to $\mathcal{H}$), if for every $n \in \mathbb{N}$, there exists $h_n \in \mathcal{H}$ such that $h_n(x_i) = y_i$, for all $i \in [n]$.*

**Definition 4** (**Star number**). *Let $\mathcal{X}$ be an instance space and $\mathcal{H}$ be a concept class. We define the region of disagreement of $\mathcal{H}$ as $DIS(\mathcal{H}) := \{x \in \mathcal{X} : \exists h, g \in \mathcal{H} \text{ s.t. } h(x) \neq g(x)\}$. Let $h$ be a classifier, the star number of $h$, denoted by $\mathfrak{s}_h(\mathcal{H})$ or $\mathfrak{s}_h$ for short, is defined to be the largest integer $s$ such that there exist distinct points $x_1, \ldots, x_s \in \mathcal{X}$ and concepts $h_1, \ldots, h_s \in \mathcal{H}$ satisfying $DIS(\{h, h_i\}) \cap \{x_1, \ldots, x_s\} = \{x_i\}$, for every $1 \leq i \leq s$. (We say $\{x_1, \ldots, x_s\}$ is a star set of $\mathcal{H}$ centered at $h$.) If no such largest integer $s$ exists, we define $\mathfrak{s}_h = \infty$. The star number of $\mathcal{H}$, denoted by $\mathfrak{s}(\mathcal{H})$ or $\mathfrak{s}_{\mathcal{H}}$, is defined to be the maximum possible cardinality of a star set of $\mathcal{H}$, or $\mathfrak{s}_{\mathcal{H}} = \infty$ if no such maximum exists.*

**Remark 4.** *From this definition, it is clear that the star number $\mathfrak{s}_{\mathcal{H}}$ of $\mathcal{H}$ satisfies $\mathfrak{s}_{\mathcal{H}} \geq VC(\mathcal{H})$. Indeed, any set $\{x_1, \ldots, x_d\}$ that shattered by $\mathcal{H}$ is also a star set of $\mathcal{H}$ based on the following reasoning: Since $\{x_1, \ldots, x_d\}$ is shattered by $\mathcal{H}$, there exists $h \in \mathcal{H}$ such that $h(x_1) = \cdots = h(x_d) = 0$. Moreover, for any $i \in [d]$, there exists $h_i \in \mathcal{H}$ satisfying $h_i(x_i) = 1$ and $h_i(x_j) = 0$ for all $j \neq i$. An immediate implication is that a VC-eluder sequence is always a star-eluder sequence (see Definition 6 and Definition 7 below).*

With these basic definitions in hand, we next define the three aforementioned sequences:

**Definition 5** (**Eluder sequence**). *Let $\mathcal{H}$ be a concept class, we say that $\mathcal{H}$ has an eluder sequence $\{(x_1, y_1), \ldots, (x_d, y_d)\}$, if it is realizable and for every integer $k \in [d]$, there exists $h_k \in \mathcal{H}$ such that $h_k(x_i) = y_i$ for all $i < k$ and $h_k(x_k) \neq y_k$. The eluder dimension of $\mathcal{H}$, denoted by $E(\mathcal{H})$, is defined to be the largest integer $d \geq 1$ such that $\mathcal{H}$ has an eluder sequence $\{(x_1, y_1), (x_2, y_2), \ldots, (x_d, y_d)\}$. If no such largest $d$ exists, we say $\mathcal{H}$ has an infinite eluder sequence and define $E(\mathcal{H}) = \infty$. We say an infinite eluder sequence $\{(x_1, y_1), (x_2, y_2), \ldots\}$ is centered at $h$, if $h(x_i) = y_i$ for all $i \in \mathbb{N}$.*

**Remark 5.** *It has been proved that* $\max\{\mathfrak{s}_{\mathcal{H}}, \log{(LD(\mathcal{H}))}\} \leq E(\mathcal{H}) \leq 4^{\max\{\mathfrak{s}_{\mathcal{H}}, 2^{LD(\mathcal{H})}\}}$ *(Li et al., 2022, Thm.8), where $LD(\mathcal{H})$ is the Littlestone dimension of $\mathcal{H}$. Moreover, the very recent work of Hanneke (2024) proved that $E(\mathcal{H}) \leq |\mathcal{H}| \leq 2^{\mathfrak{s}_{\mathcal{H}} \cdot LD(\mathcal{H})}$, which implies that any concept class with finite star number and finite Littlestone dimension must be a finite class.*

Before proceeding to the next two definitions, we define a sequence of integers $\{n_k\}_{k \in \mathbb{N}}$ as $n_1 = 0$, $n_k := \binom{k}{2}$ for all $k > 1$, which satisfies $n_{k+1} - n_k = k$ for all $k \in \mathbb{N}$.

**Definition 6 (Star-eluder sequence).** *Let $\mathcal{H}$ be a concept class and $h$ be a classifier. We say that $\mathcal{H}$ has an infinite star-eluder sequence $\{(x_1, y_1), (x_2, y_2), \ldots\}$ centered at $h$ , if it is realizable and for every integer $k \geq 1$, $\{x_{n_k+1}, \ldots, x_{n_k+k}\}$ is a star set of $V_{n_k}(\mathcal{H})$ centered at $h$.*

**Definition 7 (VC-eluder sequence).** *Let $\mathcal{H}$ be a concept class and $h$ be a classifier. We say that $\mathcal{H}$ has an infinite VC-eluder sequence $\{(x_1, y_1), (x_2, y_2), \ldots\}$ centered at $h$ , if it is realizable and labelled by $h$, and for every integer $k \geq 1$, $\{x_{n_k+1}, \ldots, x_{n_k+k}\}$ is a shattered set of $V_{n_k}(\mathcal{H})$.*

**Remark 6.** *In the definitions of eluder sequence and VC-eluder sequence, "$\{(x_1, y_1), (x_2, y_2), \ldots\}$ centered at $h$" simply means the sequence is labelled by $h$. However, the words "centered at" in the definition of star-eluder sequence is more meaningful. In this paper, we give them a uniform name in order to make Theorem 2 look consistent.*

**Remark 7.** *An infinite star-eluder (VC-eluder) sequence requires the version space to keep on having star (shattered) sets with infinitely increasing sizes. If the size cannot grow infinitely, the largest possible size of the star (shattered) set is called the star-eluder (VC-eluder) dimension (Definition 9), which plays an important role in our fine-grained analysis (Appendix C). To distinguish the notion of star-eluder (VC-eluder) sequence here from the $d$-star-eluder ($d$-VC-eluder) sequence defined in Appendix C, we may call the construction in Definition 6 an infinite strong star-eluder sequence, and the construction in Definition 7 an infinite strong VC-eluder sequence.*

*Proof Sketch of Theorem 1 and 2.* The proof of Theorem 1 is devided into two parts (Section 3 and Section 4). Roughly speaking, for each equivalence therein, we first characterize the exact universal rates by ERM via the three aforementioned sequences (see Theorems 3-6 in Section 3). We have to prove a lower bound together with an upper bound for the sufficiency since we are showing the "exact" universal rates. The lower bound is established by constructing a realizable distribution on the existent infinite sequence, and the derivation of upper bound is strongly related to the classical PAC theory. To prove the necessity, we will use the method of contradiction. Then in Section 4, we establish equivalent characterizations via those well-known complexity measures, whenever possible. Theorem 2 is an associated target-dependent version, and is directly proved by those corresponding lemmas in Section 3. The complete proof structure for Theorem 1 can be summarized as follow:

**For the first bullet**, we start by proving that $\mathcal{H}$ is universally learnable with exact rate $e^{-n}$ if and only if $\mathcal{H}$ does not have an infinite eluder sequence (Theorem 3), and then we extend the equivalence by showing that $\mathcal{H}$ does not have an infinite eluder sequence if and only if $\mathcal{H}$ is a finite class (Lemma 8). **For the second bullet**, we prove that $\mathcal{H}$ is universally learnable with exact rate $1/n$ if and only if $\mathcal{H}$ has an infinite eluder sequence but does not have an infinite star-eluder sequence (Theorem 4). Then the desired equivalence follows immediately from the first bullet. **For the third bullet**, we prove that $\mathcal{H}$ is universally learnable with exact rate $\log{(n)}/n$ if and only if $\mathcal{H}$ has an infinite star-eluder sequence but does not have an infinite VC-eluder sequence (Theorem 5). The desired equivalence comes in conjunction with the claim that $\mathcal{H}$ has an infinite VC-eluder sequence if and only if $\mathcal{H}$ has infinite VC dimension (Lemma 9). Finally, **for the last bullet**, it suffices to prove that $\mathcal{H}$ requires at least arbitrarily slow rates to be universally learned by ERM if and only if $\mathcal{H}$ has an infinite VC-eluder sequence (Theorem 6). $\qquad\square$

## 3 Exact universal rates

Sections 3 and 4 of this paper are devoted to the proof ideas of Theorems 1 and 2 with further details. In this section, we give a complete characterization of the four possible exact universal rates by ERM ($e^{-n}, 1/n, \log{(n)}/n$ and arbitrarily slow rates) via the existence/nonexistence of the three combinatorial sequences defined in Section 2. For each of the following "if and only if" results (Theorems 3-6), we are required to prove both the sufficiency and the necessity. The sufficiency consists of both an upper bound and a lower bound since we are proving the exact universal rates. The

necessity also follows simply by the method of contradiction, given the rates are exact. All technical proofs are deferred to Appendix D.1.

## 3.1 Exponential rates

**Theorem 3.** *$\mathcal{H}$ is universally learnable by ERM with exact rate $e^{-n}$ if and only if $\mathcal{H}$ does not have an infinite eluder sequence.*

The lower bound for sufficiency is straightforward and was established by Schuurmans (1997).

**Lemma 1** ($e^{-n}$ **lower bound**). *Given a concept class $\mathcal{H}$, for any learning algorithm $\hat{h}_n$, there exists a realizable distribution $P$ with respect to $\mathcal{H}$ such that $\mathbb{E}[er_P(\hat{h}_n)] \geq 2^{-(n+2)}$ for infinitely many $n$, which implies that $\mathcal{H}$ is not universally learnable at rate faster than exponential rate $e^{-n}$.*

**Remark 8.** *Note that this lower bound is actually stronger than desired in a sense that it holds for any learning algorithm (not necessarily for ERM algorithms).*

**Lemma 2** ($e^{-n}$ **upper bound**). *If $\mathcal{H}$ does not have an infinite eluder sequence (centered at $h^*$), then $\mathcal{H}$ ($h^*$) is universally learnable by ERM at rate $e^{-n}$.*

*Proof of Theorem 3.* The sufficiency follows directly from the lower bound in Lemma 1 together with the upper bound in Lemma 2. Furthermore, Lemma 3 in Section 3.2 proves that the existence of an infinite eluder sequence leads to a linear lower bound of the ERM universal rates. Therefore, the necessity follows by using the method of contradiction. $\square$

## 3.2 Linear rates

**Theorem 4.** *$\mathcal{H}$ is universally learnable by ERM with exact rate $1/n$ if and only if $\mathcal{H}$ has an infinite eluder sequence but does not have an infinite star-eluder sequence.*

**Lemma 3** ($1/n$ **lower bound**). *If $\mathcal{H}$ has an infinite eluder sequence centered at $h^*$, then $h^*$ is not universally learnable by ERM at rate faster than $1/n$.*

**Lemma 4** ($1/n$ **upper bound**). *If $\mathcal{H}$ does not have an infinite star-eluder sequence (centered at $h^*$), then $\mathcal{H}$ ($h^*$) is universally learnable by ERM at rate $1/n$.*

*Proof of Theorem 4.* To prove the sufficiency, on one hand, the existence of an infinite eluder sequence implies a linear lower bound based on Lemma 3. On the other hand, if $\mathcal{H}$ does not have an infinite star-eluder sequence, Lemma 4 yields a linear upper bound. The necessity can be proved by the method of contradiction. Concretely, if either of the two conditions fail, the universal rates will be either $e^{-n}$ or at least $\log(n)/n$, based on Lemma 2 in Section 3.1 and Lemma 5 in Section 3.3. $\square$

## 3.3 $\log(n)/n$ rates

**Theorem 5.** *$\mathcal{H}$ is universally learnable by ERM with exact rate $\log(n)/n$ if and only if $\mathcal{H}$ has an infinite star-eluder sequence but does not have an infinite VC-eluder sequence.*

**Lemma 5** ($\log(n)/n$ **lower bound**). *If $\mathcal{H}$ has an infinite star-eluder sequence centered at $h^*$, then $h^*$ is not universally learnable by ERM at rate faster than $\log(n)/n$.*

**Remark 9.** *Note that the conclusion in Remark 5 explains why the intersection of "infinite Littlestone classes" and "classes with finite star number" is empty in Figure 1. However, we mention in Remark 3 that infinite star number does not guarantee an infinite star-eluder sequence (see Appendix B.3 for details). Hence, Remark 5 cannot explain why the intersection of "infinite Littlestone classes" and "classes that are learnable at linear rate by ERM" is also empty. To address this problem, we give out the following additional result:*

**Proposition 1.** *Any infinite concept class $\mathcal{H}$ has either an infinite star-eluder sequence or infinite Littlestone dimension.*

**Lemma 6** ($\log(n)/n$ **upper bound**). *If $\mathcal{H}$ does not have an infinite VC-eluder sequence (centered at $h^*$), then $\mathcal{H}$ ($h^*$) is universally learnable by ERM at $\log(n)/n$ rate.*

*Proof of Theorem 5.* To prove the sufficiency, on one hand, if $\mathcal{H}$ has an infinite star-eluder sequence, the universal rates have a $\log(n)/n$ lower bound based on Lemma 5. On the other hand, if $\mathcal{H}$ does not

have an infinite VC-eluder sequence, then Lemma 6 yields a $\log(n)/n$ upper bound. The necessity can be proved using the method of contradiction based on Lemma 4 in Section 3.2 and Lemma 7 in Section 3.4 below. □

### 3.4 Arbitrarily slow rates

**Theorem 6.** *$\mathcal{H}$ requires at least arbitrarily slow rates to be learned by ERM if and only if $\mathcal{H}$ has an infinite VC-eluder sequence.*

*Proof of Theorem 6.* Given the necessity proved by Lemma 6 in Section 3.3, it suffices to prove the sufficiency, which is completed by the following Lemma 7. □

**Lemma 7 (Arbitrarily slow rates).** *If $\mathcal{H}$ has an infinite VC-eluder sequence centered at $h^*$, then $h^*$ requires at least arbitrarily slow rates to be universally learned by ERM.*

## 4 Equivalent characterizations

In Section 3, it has been shown that the eluder sequence, the star-eluder sequence and the VC-eluder sequence are the correct characterizations of the exact universal learning rates by ERM. However, the definitions to them are somewhat non-intuitive. Therefore, in this section, we aim to build connections between these combinatorial sequences and some well-understood complexity measures, which will then give rise to our Theorem 1. Concretely, we have the following two equivalences (see Appendix D.2 for their complete proofs).

**Lemma 8.** *$\mathcal{H}$ has an infinite eluder sequence if and only if $|\mathcal{H}| = \infty$.*

**Lemma 9.** *$\mathcal{H}$ has an infinite VC-eluder sequence if and only if $VC(\mathcal{H}) = \infty$.*

Maybe surprisingly, unlike the above two equivalences, $\mathfrak{s}_{\mathcal{H}} = \infty$ is inequivalent to the existence of an infinite star-eluder sequence. Indeed, it is straightforward from definition that if $\mathcal{H}$ has an infinite star-eluder sequence, then it must have $\mathfrak{s}_{\mathcal{H}} = \infty$. However, the converse is not true.

**Proposition 2.** *$\mathfrak{s}_{\mathcal{H}} = \infty$ if $\mathcal{H}$ has an infinite star-eluder sequence. Moreover, there exist concept classes $\mathcal{H}$ with $\mathfrak{s}_{\mathcal{H}} = \infty$ but does not have any infinite star-eluder sequence.*

**Remark 10.** *Based on the results in Section 3, the proposition essentially states that the gap between $1/n$ and $\log(n)/n$ exact universal rates by ERM is not characterized by the star number $\mathfrak{s}_{\mathcal{H}}$. We wonder whether there is some other simple combinatorial quantity that is determinant to this gap, which would be an valuable direction for future work.*

Why is the case of star-eluder sequence different from the other two structures? We suspect that such a distinction may arise from the following: unlike the eluder sequence and the VC-eluder sequence, the centered concept of a star-eluder sequence is much more meaningful (see Remark 6). Concretely, within its definition, the set of the following $k$ points is not only required to be a star set of the version space $V_{n_k}(\mathcal{H})$, but is required to be centered at the same labelling target. This intuitively implies that there might exists a class such that for arbitrarily large integer $k$, it can witness a star set of size $k$, but with a $k$-specified center (for different $k$). Such a class (e.g. Examples 19, 20 in Appendix B.3) does have infinite star number but will not have an infinite star-eluder sequence. Indeed, the relations between those star-related notions (star number, star-eluder dimension, star set and star eluder sequence) turn out to be more complicated than expected, and we leave it to Appendix B.3.

## 5 Appendix Summary

Due to page limitation, we leave some interesting results as well as all the proofs to Appendices, which are briefly summarized below. Given extra required notations and definitions in Appendix A and related technical lemmas in Appendix E, the main body of supplements consists of three parts, namely, Appendices B, C and D.

Specifically, Appendix B contains three sub-parts. In Appendix B.1, we provide direct mathematical analysis (without using our characterization in Section 1.2) for those basic examples in Section 1.1. In Appendix B.2, we provide details of examples that appeared in Section 1.2. These examples are

carefully constructed, providing evidence that ERM algorithms cannot guarantee the optimal universal rates (Bousquet et al., 2021). In Appendix B.3, we construct nuanced examples to distinguish between the following notions: star number $\mathfrak{s}_{\mathcal{H}}$ (Definition 4), the star-eluder dimension SE($\mathcal{H}$) (Definition 9), star set (Definition 4) and star eluder sequence (Definition 6). These examples will convince the readers why our characterization in Theorem 1 uses the star eluder sequence rather than the star number (see our discussions in Remarks 3 and 10).

Appendix C presents a fine-grained analysis of the asymptotic rate of decay of the universal learning curves by ERM, whenever possible. This will be an analogy to the optimal fine-grained universal learning curves studied in Bousquet et al. (2023). Concretely, we provide a characterization of sharp distribution-free constant factors of the ERM universal rates. Our characterization of these constant factors is based on two newly-developed combinatorial dimensions, namely, the *star-eluder dimension* (or SE dimension) and the *VC-eluder dimension* (or VCE dimension) (Definition 9). We say "whenever possible" because distribution-free constants are unavailable for certain cases (see our discussion in Remark 16). Such a characterization can be considered as a refinement to the classical PAC theory, in a sense that it is sometimes better but only asymptotically-valid.

Finally, Appendix D includes all the missing proofs for the theorems and lemmas that have shown up in previous sections.

## 6    Conclusion and Future Directions

In this paper, we reveal a fundamental tetrachotomy of the universal learning rates by the ERM principle and provide a complete characterization of the exact universal rates via certain appropriate complexity structures. Additionally, by introducing new combinatorial dimensions, we are able to characterize sharp asymptotically-valid constant factors for these rates, whenever possible. While only the realizable case is considered in this paper, we believe analogous results can be extend to different learning scenarios such as the agnostic case. Generalizing the results from binary classification to multiclass classification would be another valuable future direction. Moreover, since this paper considers the "worst-case" ERM in its nature, studying the universal rates of the "best-case" ERM is also an interesting problem which we leave for future work.

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

# A  Preliminaries

**Notation 1.** *We denote by $\mathbb{N}$ the set of all natural numbers $\{0, 1, \ldots\}$. For any $n \in \mathbb{N}$, we denote $[n] := \{1, \ldots, n\}$.*

**Notation 2.** *For any $x > 0$, we redefine $\ln(x) := \ln(x \vee e)$ and $\log(x) := \log_2(x \vee 2)$. Moreover, for correctness, we also adopt the conventions that $\ln(0) = \log(0) = 0$, $0 \ln(\infty) = 0 \log(\infty) = 0$. After then, it is reasonable to define $0 \ln(0/0) = 0 \log(0/0) = 0$.*

**Notation 3.** *For any $\mathbb{R}$-valued functions $f$ and $g$, we write $f(x) \lesssim g(x)$ if there exists a finite numerical constant $c > 0$ such that $f(x) \leq c \cdot g(x)$ for all $x \in \mathbb{R}$. For example, $\ln(x) \lesssim \log(x)$ and $\log(x) \lesssim \ln(x)$.*

**Notation 4.** *Let $\mathcal{X}$ be an instance space, we write $h_{all\text{-}0's}$ and $h_{all\text{-}1's}$ to denote the hypotheses that output all zero labels and all one labels, respectively, that is, $h_{all\text{-}0's}(x) = 0, h_{all\text{-}1's}(x) = 1, \forall x \in \mathcal{X}$.*

**Notation 5.** *For an infinite union of spaces $(\mathcal{X}_1 \cup \mathcal{X}_2 \cup \cdots)$ and an integer $k$, we write $\mathcal{X}_{<k}$ to denote the finite union of prefix $(\mathcal{X}_1 \cup \cdots \cup \mathcal{X}_{k-1})$ and write $\mathcal{X}_{>k}$ to denote the infinite union of suffix $(\mathcal{X}_{k+1} \cup \mathcal{X}_{k+2} \cup \cdots)$.*

**Definition 8** (**Empirical risk minimization**). *Let $\mathcal{H}$ be a concept class on an instance space $\mathcal{X}$. For every $n \in \mathbb{N}$, let $S_n := \{(x_i, y_i)\}_{i=1}^n \in (\mathcal{X} \times \{0, 1\})^n$ be a set of samples. A learning algorithm that outputs $\{\hat{h}_n\}_{n \in \mathbb{N}}$ is called an $\underline{Empirical\ Risk\ Minimization}$ (ERM) algorithm, if it satisfies $\hat{h}_n \in \arg\min_{h \in \mathcal{H}} \hat{er}_{S_n}(h) := \arg\min_{h \in \mathcal{H}} \{\frac{1}{n} \sum_{i=1}^n \mathbb{1}(h(x_i) \neq y_i)\}$ for all $n \in \mathbb{N}$, where $\hat{er}_{S_n}(\hat{h}_n)$ is called the $\underline{empirical\ error\ rate}$ of $\hat{h}_n$ on $S_n$. It is clear that $\hat{er}_{S_n}(\hat{h}_n) = 0$ when $P \in RE(\mathcal{H})$.*

# B  Detailed examples

In this appendix section, we provide further examples. Specifically, in Appendix B.1, we present direct analysis (without using our newly-developed characterization) of each example illustrated in Section 1.1. The aim of the examples in Appendix B.2 is to reveal that ERM algorithms can sometimes be optimal but sometimes not in a universal learning framework, and compare their performance with the optimal universal learning algorithms. Finally, we also provide additional examples related to the star number in Appendix B.3 as complements to Proposition 2 in Section 4.

## B.1  Details of examples in Section 1.1

We provide direct analysis to the examples illustrated in Section 1.1 without using the characterization in Theorem 1. Concretely, Examples 8 and 9 illustrate scenarios where linear universal rates occur. Example 10 specifies a case where universal rate matches uniform rate by ERM as $\log(n)/n$. Example 11 specifies a case where extremely fast universal learning is achievable, but where some bad ERM algorithms can give rise to arbitrarily slow rates.

**Example 7** (**Example 1 restated**). *Any finite class $\mathcal{H}$ is universally learnable at exponential rate by ERM. To show this, for any realizable distribution $P$ with respect to $\mathcal{H}$, we have*

$$
\begin{aligned}
\mathbb{E}\left[er_P(\hat{h}_n)\right] \quad &\leq \quad \mathbb{P}\left(\exists h \in \mathcal{H} : er_P(h) > 0, \hat{er}_{S_n}(h) = 0\right) \\
&\overset{\text{union bound}}{\leq} \quad \sum_{h \in \mathcal{H}: er_P(h) > 0} \mathbb{P}\left(\hat{er}_{S_n}(h) = 0\right) \\
&= \quad \sum_{h \in \mathcal{H}: er_P(h) > 0} (1 - er_P(h))^n \\
&\leq \quad |\mathcal{H}| \cdot \left(1 - \min_{h \in \mathcal{H}: er_P(h) > 0} er_P(h)\right)^n \\
&\overset{1 - t \leq e^{-t}}{\leq} \quad |\mathcal{H}| \cdot \exp\left\{-\left(\min_{h \in \mathcal{H}: er_P(h) > 0} er_P(h)\right) \cdot n\right\}.
\end{aligned}
$$

**Example 8** (**Example 2 restated**). *Let $\mathcal{H}_{thresh,\mathbb{N}} := \{h_t : t \in \mathbb{N}\}$ be the class of all threshold classifiers on the space of natural numbers defined by $h_t(x) := \mathbb{1}(x \geq t)$. $\mathcal{H}_{thresh,\mathbb{N}}$ is universally learnable at exponential rate since this concept class does not have an infinite Littlestone tree*

*([Bousquet et al., 2021](#)). In the following part, we show that the worst-case ERM cannot achieve such exponential rate, but has a rate $1/n$.*

*Let $h_{t^*} \in \mathcal{H}_{thresh,\mathbb{N}}$ be the target hypothesis. Given a dataset $S_n$, let $h_{\hat{t}} = ERM(S_n)$ be the output of an ERM algorithm. For any realizable distribution $P$ satisfying $P\{(t,0)\} = 1$ for all $t < t^*$ and $P\{(t,1)\} = 1$ for all $t \geq t^*$, we define*

$$t_l := \max\{t < t^* : P(t) > 0\}.$$

*According to the definition of threshold classifiers, if the dataset $S_n$ contains at least a copy of both $t_l$ and $t^*$, then $er_P(h_{\hat{t}}) = 0$. Therefore, we have*

$$\mathbb{E}\left[er_P(h_{\hat{t}})\right] \leq \mathbb{P}\left(er_P(h_{\hat{t}}) > 0\right) \leq (1 - P(t^*))^n + (1 - P(t_l))^n.$$

*Note that for any $\epsilon \in (0,1)$, $1 - \epsilon \leq e^{-\epsilon}$, it follows immediately that*

$$\mathbb{E}\left[er_P(h_{\hat{t}})\right] \leq (1 - P(t^*))^n + (1 - P(t_l))^n \leq e^{-nP(t^*)} + e^{-nP(t_l)} \leq 2e^{-n\cdot\min\{P(t^*),P(t_l)\}}.$$

*However, let us consider a distribution $P$ satisfying $P\{(t,0)\} = 2^{-t}$ and $P\{(t,1)\} = 0$ for all $t \in \mathbb{N}$. Note that $P$ is also realizable with respect to $\mathcal{H}_{thresh,\mathbb{N}}$ according to the definition, that is*

$$\inf_{h \in \mathcal{H}_{thresh,\mathbb{N}}} er_P(h) = \inf_{t \in \mathbb{N}} er_P(h_t) = \inf_{t \in \mathbb{N}} 2^{-t} = 0.$$

*Given a dataset $S_n := \{(x_i, y_i)\}_{i=1}^n \sim P^n$, let $t_n := \max_{i \in [n]} x_i$ be the largest point in the dataset, it is straightforward that the worst-case ERM outputs $h_{\widehat{t_n+1}}$, and we have that*

$$\mathbb{E}\left[er_P\left(h_{\widehat{t_n+1}}\right)\right] = \sum_{t \geq 1} 2^{-t-1} \cdot \mathbb{P}\left(\max_{i \in [n]} x_i = t\right) = \sum_{t \geq 1} 2^{-t-1}\left[\left(1 - 2^{-t}\right)^n - \left(1 - 2^{-(t-1)}\right)^n\right].$$

*On one hand, we can lower bound the above infinite series by*

$$\sum_{t \geq 1} 2^{-t-1}\left[\left(1 - 2^{-t}\right)^n - \left(1 - 2^{-(t-1)}\right)^n\right]$$

$$\geq \sum_{t=1}^{\lfloor \log n \rfloor} 2^{-t-1}\left[\left(1 - 2^{-t}\right)^n - \left(1 - 2^{-(t-1)}\right)^n\right]$$

$$\geq \frac{1}{2n} \sum_{t=1}^{\lfloor \log n \rfloor}\left[\left(1 - 2^{-t}\right)^n - \left(1 - 2^{-(t-1)}\right)^n\right]$$

$$\geq \frac{1}{2n}\left(1 - \frac{2}{n}\right)^n \geq \frac{1}{18n}, \text{ for infinitely many } n.$$

*This implies that $\mathcal{H}_{thresh,\mathbb{N}}$ is not learnable by ERM at rate faster than $1/n$. On the other hand, we have the following upper bound for all $n$:*

$$\sum_{t \geq 1} 2^{-t-1}\left[\left(1 - 2^{-t}\right)^n - \left(1 - 2^{-(t-1)}\right)^n\right] \leq \sum_{t \geq 1} 2^{-t}\left(1 - 2^{-t}\right)^n \leq \int_0^1 \epsilon(1-\epsilon)^n d\epsilon = \frac{1}{n+1},$$

*which implies that $\mathcal{H}_{thresh,\mathbb{N}}$ is indeed learnable by ERM at linear rate $1/n$. In conclusion, $\mathcal{H}_{thresh,\mathbb{N}}$ is universally learnable by ERM with exact $1/n$ rate.*

**Example 9 (Threshold classifier on $\mathbb{R}$).** *This example serves as a complement to Example [8](#). Here, we show that ERM algorithms can sometimes be optimal for universal learning. Specifically, let $\mathcal{H}_{thresh,\mathbb{R}} := \{h_t : t \in \mathbb{R}\}$ be the class of all threshold classifiers on the real line defined by $h_t(x) := \mathbb{1}(x \geq t), \forall x \in \mathbb{R}$. It has been shown that $\mathcal{H}_{thresh,\mathbb{R}}$ is universally learnable with optimal linear rate ([Schuurmans, 1997](#)).*

*To show that $\mathcal{H}_{thresh,\mathbb{R}}$ is also universally learnable with exact linear rate by ERM, we only need to prove an upper bound. To this end, let $h_{t^*} \in \mathcal{H}_{thresh,\mathbb{R}}$ be the target hypothesis. For any realizable distribution $P$ satisfying $P\{(t,0)\} = 1$ for all $t < t^*$ and $P\{(t,1)\} = 1$ for all $t \geq t^*$, a dataset $S_n$ and $\epsilon \in (0,1)$, let $h_{\hat{t}} = ERM(S_n)$ be the output of an ERM algorithm. Now we define $A$ and $B$ be the minimal regions left and right to $t^* \in \mathbb{R}$ such that $\mathbb{P}(A) = \mathbb{P}(B) = \epsilon$. If at least one of $A$*

*and B does not contain any sample, then the worst-case ERM can output $h_{\hat{t}} \in \mathcal{H}_{thresh,\mathbb{R}}$ such that $er_P(h_{\hat{t}}) \geq \epsilon$. Therefore, it follows that*

$$\mathbb{E}\left[er_P(h_{\hat{t}})\right] = \int_0^1 \mathbb{P}\left(er_P(h_{\hat{t}}) \geq \epsilon\right) d\epsilon \leq \int_0^1 2(1-\epsilon)^n d\epsilon \leq \int_0^1 2e^{-n\epsilon} d\epsilon \leq \frac{2}{n}.$$

*Note that such analysis is also applicable to the realizable distribution with the target concept $h_{all\text{-}0's}$. Therefore, we have that $\mathcal{H}_{thresh,\mathbb{R}}$ is universally learnable by ERM with exact rate $1/n$.*

**Example 10 (Example 3 restated).** *Let $\mathcal{X} = \mathbb{N}$ and define $\mathcal{H}_{singleton,\mathbb{N}} := \{h_t : t \in \mathcal{X}\}$ be the class of all singletons on $\mathcal{X}$, where $h_t(x) := \mathbb{1}(x = t)$, for all $x \in \mathcal{X}$. It is clear that $VC(\mathcal{H}_{singleton,\mathbb{N}}) = 1$. Note that $\mathcal{H}_{singleton,\mathbb{N}}$ is universally learnable at exponential rate since it does not have an infinite Littlestone tree (Actually, we have $LD(\mathcal{H}_{singleton,\mathbb{N}}) = 1$). In the following part, we show that the worst-case ERM algorithm has an exact universal rate $\log(n)/n$.*

*To get the exact rate by ERM on universally learning $\mathcal{H}_{singleton,\mathbb{N}}$, we consider a marginal uniform distribution over $\{1, 2, \ldots, 1/\epsilon\}$ with all zero labels with $\epsilon \in (0, 1)$, if the dataset $S_n$ does not have a copy of a point $1 \leq x \leq 1/\epsilon$, the worst-case ERM can label 1 at $x$, and thus has an error rate $er_P(\hat{h}_n) \geq \epsilon$. Based on the Coupon Collector's Problem, we know that to have $\mathbb{E}[er_P(\hat{h}_n)] \leq \epsilon$, we need $n = \Omega(\epsilon^{-1} \log(1/\epsilon))$. In other words, $\mathbb{E}[er_P(\hat{h}_n)] \geq \Omega(\frac{\log n}{n})$, that is, $\mathcal{H}_{singleton,\mathbb{N}}$ is not universally learnable by ERM at rate faster than $\log(n)/n$. Finally, the classical PAC theory yields the same upper bound, and thus $\log(n)/n$ is tight.*

**Example 11 (Example 4 restated).** *Let $\mathcal{X} = \bigcup_{i \in \mathbb{N}} \mathcal{X}_i$ be the disjoint union of finite sets with $|\mathcal{X}_i| = 2^i$. For each $i \in \mathbb{N}$, let*

$$\mathcal{H}_i := \left\{ h_S := \mathbb{1}_S : S \subseteq \mathcal{X}_i, |S| \geq 2^{i-1} \right\},$$

*and consider the concept class $\mathcal{H} = \bigcup_{i \in \mathbb{N}} \mathcal{H}_i$. In the following part, we show that the worst-case ERM can be arbitrarily slow in learning this class.*

*Given any rate function $R(n) \to 0$, let $\{n_t\}_{t \geq 1}$ and $\{i_t\}_{t \geq 1}$ be two strictly increasing sequences such that $\{p_t := 2^{i_t-2}/n_t, \forall t \geq 1\}$ satisfies*

$$\{p_t\}_{t \geq 1} \text{ is decreasing}, \sum_{t \geq 1} p_t \leq 1 \text{ and } p_t \geq 4R(n_t).$$

*We consider any ERM algorithm with the following property: if the data $S_n = \{(x_i, y_i)\}_{i=1}^n$ satisfies $y_i = 0$ for all $i \in [n]$, outputs $\hat{h}_n \in \mathcal{H}_{i_{T_n}}$ with*

$$T_n := \min \left\{ t : \exists h \in \mathcal{H}_{i_t} \text{ s.t. } h(x_1) = \cdots = h(x_n) = 0 \right\}.$$

*We construct the following distribution $P$:*

$$P\{(x, 0)\} = 2^{-i_t} p_t, \text{ for all } x \in \mathcal{X}_{i_t}, t \in \mathbb{N},$$

*where we set $P\{(x', 0)\} = 1 - \sum_{t \geq 1} p_t$ for some arbitrary choice of $x' \notin \bigcup_{t \in \mathbb{N}} \mathcal{X}_{i_t}$. Since*

$$\inf_{h \in \mathcal{H}} er_P(h) = \inf_{i \in \mathbb{N}} \inf_{h \in \mathcal{H}_i} er_P(h) \leq \inf_{i \in \mathbb{N}} er_P(h_{\mathcal{X}_i}) \leq \inf_{i_t : t \in \mathbb{N}} er_P(h_{\mathcal{X}_{i_t}}) = \inf_{i_t : t \in \mathbb{N}} P\{(x, 0) : x \in \mathcal{X}_{i_t}\} = 0,$$

*we know that $P$ is realizable with respect to $\mathcal{H}$. Finally, we claim that the ERM defined above behave poorly on $P$ by showing $\mathbb{E}[er_P(\hat{h}_n)] \geq R(n)$ for infinitely many $n$. To this end, note that for a dataset $S_{n_t} = \{(x_i, y_i)\}_{i=1}^{n_t} \sim P^{n_t}$, and for any $t \in \mathbb{N}$, it holds*

$$\mathbb{P}\left(T_{n_t} \leq t\right) \geq \mathbb{P}\left(\left|\{j \in [n_t] : x_j \in \mathcal{X}_{i_t}\}\right| \leq 2^{i_t-1}\right) = \mathbb{P}\left(\sum_{j=1}^{n_t} \mathbb{1}\{x_j \in \mathcal{X}_{i_t}\} \leq 2^{i_t-1}\right) \geq \frac{1}{2},$$

*where the last inequality follows from the Markov's inequality. Therefore,*

$$
\begin{aligned}
\mathbb{E}\left[er_P(\hat{h}_{n_t})\right] \quad &\geq \quad 2R(n_t) \cdot \mathbb{P}\left(er_P(\hat{h}_{n_t}) \geq 2R(n_t)\right) \\
&\overset{p_t \geq 4R(n_t)}{\geq} 2R(n_t) \cdot \mathbb{P}\left(er_P(\hat{h}_{n_t}) \geq \frac{1}{2}p_t\right) \\
&\overset{LoFT}{\geq} \quad 2R(n_t) \cdot \mathbb{P}\left(er_P(\hat{h}_{n_t}) \geq \frac{1}{2}p_t\Big|T_{n_t} \leq t\right)\mathbb{P}\left(T_{n_t} \leq t\right) \\
&\geq \quad 2R(n_t) \cdot \mathbb{P}\left(er_P(\hat{h}_{n_t}) \geq \frac{1}{2}p_{T_{n_t}}\Big|T_{n_t} \leq t\right)\mathbb{P}\left(T_{n_t} \leq t\right) \\
&\geq \quad 2R(n_t) \cdot \mathbb{P}\left(T_{n_t} \leq t\right) \\
&\geq \quad R(n_t).
\end{aligned}
$$

## B.2 Optimal universal rates versus exact universal rates by ERM

In this section, we provide evidence that ERM algorithms cannot guarantee the best achievable universal learning rates. Recall that the optimal universal learning rates and the associated characterization have been fully understood by Bousquet et al. (2021), which we present first as follow:

**Theorem 7** (**Bousquet et al., 2021**, **Theorem 1.9**). *For every concept class $\mathcal{H}$ with $|\mathcal{H}| \geq 3$, the following hold:*

- *$\mathcal{H}$ is universally learnable with optimal rate $e^{-n}$ if $\mathcal{H}$ does not have an infinite Littlestone tree.*
- *$\mathcal{H}$ is universally learnable with optimal rate $1/n$ if $\mathcal{H}$ has an infinite Littlestone tree but does not have an infinite (strong) VCL tree.*
- *$\mathcal{H}$ requires arbitrarily slow rates if $\mathcal{H}$ has an infinite (strong) VCL tree.*

Based on our Theorem 1, to distinguish the optimal universal rates from the exact universal rates by ERM, we have to distinguish their corresponding characterizations. Indeed, those sequences we defined in Section 2 are strongly related to the Littlestone tree and the VCL tree in Theorem 7. According to the definitions, it is not hard to figure out all the following relations:

- Every branch of a Littlestone tree is an eluder sequence. Hence, if $\mathcal{H}$ does not have an infinite eluder sequence, then $\mathcal{H}$ must not have an infinite Littlestone tree, and thus can be universally learned with optimal exponential rate. However, there exists a class $\mathcal{H}$ having an infinite eluder sequence but no infinite Littlestone tree (see Example 12 below). This implies that such a class cannot be universally learned by ERM at rate faster than $1/n$, but can be learned by some other "optimal" learning algorithms at $e^{-n}$ rate.
- Every branch of a (strong) VCL tree is a VC-eluder sequence, and also a star-eluder sequence. Therefore, if $\mathcal{H}$ does not have an infinite star-eluder sequence, then it must not have an infinite VCL tree, and thus can be universally learned with optimal linear rate. However, there exists a concept class that has an infinite star-eluder sequence, but does not have an infinite VCL tree (see Example 14 below). Furthermore, there also exists a concept class that has an infinite star-eluder sequence, but does not even have an infinite Littlestone tree (see Example 13 below). These two examples imply that there exist classes that can not be universally learned by ERM at rate faster than $\log(n)/n$, but can be learned by some other "optimal" learning algorithms at $1/n$ or even $e^{-n}$ rates.
- Moreover, there exists a concept class that has an infinite VC-eluder sequence, but does not have an infinite VCL tree (see Example 16 below), or even no infinite Littlestone tree (see Example 15 below). Such examples imply that there exist classes that require arbitrarily slow rates to be universally learned by ERM, but can be learned by some other "optimal" learning algorithms at $1/n$ or even $e^{-n}$ rates.

To summarize, we are able to illustrate all the distinctions as in the following table.

| Optimal rate | Exact rate by ERM | Case | Example |
|---|---|---|---|
| $e^{-n}$ | $1/n$ | infinite eluder sequence but no infinite Littlestone tree | Example 12 |
| $e^{-n}$ | $\log(n)/n$ | infinite star-eluder sequence but no infinite Littlestone tree | Example 13 |
| $e^{-n}$ | arbitrarily slow | infinite VC-eluder sequence but no infinite Littlestone tree | Example 15 |
| $1/n$ | $\log(n)/n$ | infinite star-eluder sequence but no infinite VCL tree | Example 14 |
| $1/n$ | arbitrarily slow | infinite VC-eluder sequence but no infinite VCL tree | Example 16 |

**Example 12** (**Infinite eluder sequence but no infinite Littlestone tree**). *A simple example is given in Example 8, where $\mathcal{H} = \mathcal{H}_{thresh,\mathbb{N}}$ is the class of all threshold classifiers on $\mathbb{N}$. Note that $\mathcal{H}$ does not have an infinite Littlestone tree, but any infinite sequence $\{(x_1,0),(x_2,0),\ldots\}$ with $x_1 < x_2 < \ldots$ is an infinite eluder sequence of $\mathcal{H}$ centered at $h_{all\text{-}0's}$. In particular, $h_{all\text{-}0's}$ is the only realizable target that allows an infinite eluder sequence. In other words, for $\mathcal{H}$, all the realizable distribution with target concept $h^* \in \mathcal{H}$ is universally learnable by ERM at exponential rate, except that special one $h_{all\text{-}0's}$, which matches our analysis within Example 8.*

**Example 13** (**Infinite star-eluder sequence but no infinite Littlestone tree**). *Let $\mathcal{X} := \bigcup_{k \in \mathbb{N}} \mathcal{X}_k$ be the disjoint union of finite sets with $|\mathcal{X}_k| = k$ and $\mathcal{H} := \bigcup_{k \geq 1} \mathcal{H}_k$, where $\mathcal{H}_k := \{\mathbb{1}_x : x \in \mathcal{X}_k\}$. Note that this is exactly singletons on an infinite domain and we have the following hold:*

1. *$\mathcal{H}$ does not have an infinite Littlestone tree since for any root $x \in \mathcal{X}$, the subclass $\{h \in \mathcal{H} : h(x) = 1\}$ has only size 1, and thus the corresponding subtree of the Littlestone tree must be finite.*
2. *$\mathcal{H}$ has an infinite star-eluder sequence. Indeed, any infinite sequence $\{(x_1,0),(x_2,0),\ldots\}$ with $x_k \in \mathcal{X}_k$ for all $k \geq 1$, is an infinite star-eluder sequence. To see this, note that for any $k \in \mathbb{N}$, and any $n_k$, the version space $V_{n_k}(\mathcal{H})$ contains $\bigcup_{j > n_k} \mathcal{H}_j$. Therefore, $\{(x_{n_k+1},0),(x_{n_k+2},0),\ldots,(x_{n_k+k},0)\}$ is a star set of $V_{n_k}(\mathcal{H})$ centered at $h_{all\text{-}0's}$, witnessed by concepts $\{\mathbb{1}_{\{x_{n_k+1}\}},\mathbb{1}_{\{x_{n_k+2}\}},\ldots,\mathbb{1}_{\{x_{n_k+k}\}}\}$.*

**Example 14** (**Infinite star-eluder sequence but no infinite VCL tree**). *Let $\mathcal{X}_1$ and $\mathcal{H}_1$ be defined in Example 13, let $\mathcal{X}_2 = \mathbb{R}$ and $\mathcal{H}_2 = \mathcal{H}_{thresh,\mathbb{R}}$ be the class of all threshold classifiers on $\mathbb{R}$. Note that $\mathcal{H}_2$ has an infinite Littlestone tree. Now we define $\mathcal{X} := \mathcal{X}_1 \cup \mathcal{X}_2$ and $\mathcal{H} := \mathcal{H}_1 \cup \mathcal{H}_2$, and have the following hold:*

1. *$\mathcal{H}$ does not have an infinite VCL tree since for any fixed root $x \in \mathcal{X}$, the subclass $\{h \in \mathcal{H} : h(x) = 1\}$ has a VC dimension only 1, and thus the corresponding subtree of the VCL tree must be finite.*
2. *$\mathcal{H}$ has an infinite star-eluder sequence (see Example 13).*

**Example 15** (**Infinite VC-eluder sequence but no infinite Littlestone tree**). *Let $\mathcal{X} := \bigcup_{k \in \mathbb{N}} \mathcal{X}_k$ be the disjoint union of finite sets with $|\mathcal{X}_k| = k$ and $\mathcal{H} := \bigcup_{k \geq 1} \mathcal{H}_k$, where $\mathcal{H}_k := \{\mathbb{1}_S : S \subseteq \mathcal{X}_k\}$. We have the following hold:*

1. *$\mathcal{H}$ does not have an infinite Littlestone tree since for any root $x \in \mathcal{X}$, the subclass $\{h \in \mathcal{H} : h(x) = 1\}$ is finite, and thus the corresponding subtree of the Littlestone tree must be finite.*
2. *$\mathcal{H}$ has an infinite VC-eluder sequence. Indeed, any sequence $\{(x_1,0),(x_2,0),(x_3,0),\ldots\}$ with $x_{n_k+1},\ldots,x_{n_k+k} \in \mathcal{X}_k$ for all $k \geq 1$, is an infinite VC-eluder sequence. Furthermore, it has been argued that $VC(\mathcal{H}) = \infty$ (Ex.2.3 Bousquet et al., 2021), which is consistent with our Lemma 9 in Section 4.*

**Example 16** (**Infinite VC-eluder sequence but no infinite VCL tree**). *Let $\mathcal{X}_1$ and $\mathcal{H}_1$ be defined in Example 15, let $\mathcal{X}_2 = \mathbb{R}$ and $\mathcal{H}_2 = \mathcal{H}_{thresh,\mathbb{R}}$ be the class of all threshold classifiers on $\mathbb{R}$. Note that $\mathcal{H}_2$ has an infinite Littlestone tree. Now we define $\mathcal{X} := \mathcal{X}_1 \cup \mathcal{X}_2$ and $\mathcal{H} := \mathcal{H}_1 \cup \mathcal{H}_2$, and have the following hold:*

1. *$\mathcal{H}$ does not have an infinite VCL tree since for any fixed root $x \in \mathcal{X}$, the subclass $\{h \in \mathcal{H} : h(x) = 1\}$ has a bounded VC dimension, and thus the corresponding subtree of the VCL-tree must be finite.*
2. *$\mathcal{H}$ has an infinite VC-eluder sequence (see Example 15).*

### B.3 Star-related notions

In this section, we provide examples to distinguish between the following star-related notions: star number $\mathfrak{s}_{\mathcal{H}}$ (Definition 4), the star-eluder dimension $SE(\mathcal{H})$ (Definition 9), star set (Definition 4) and star eluder sequence (Definition 6).

In particular, Example 17 reveals that having an infinite star number of $h^*$ does not guarantee that $\mathcal{H}$ has an infinite star-eluder sequence centered at the same target $h^*$. Note that if $\mathfrak{s}_{\mathcal{H}} = \infty$ always yields an infinite star set, then we can simply choose this infinite star set to be an infinite star-eluder sequence. Unfortunately, Example 18 fails the conjecture. Furthermore, Proposition 2 in Section 4 is convinced by Example 19. Finally, Example 20 gives an instance that $\text{SE}(\mathcal{H}) = \infty$ and infinite star-eluder sequence are not equivalent as well. For comparison, we recall that $\text{E}(\mathcal{H}) = \infty$ is equivalent to an infinite eluder sequence, and $\text{VCE}(\mathcal{H}) = \infty$ is equivalent to an infinite VC-eluder sequence (see a discussion in Appendix C).

**Example 17** (**Infinite star number and infinite star-eluder sequence with different centers**). *Let us recall Example 3, where $\mathcal{H}_{singleton,\mathbb{N}}$ is the class of singletons on natural numbers. According to the analysis in Example 13, we know that $\mathcal{H}_{singleton,\mathbb{N}}$ has an infinite star number of $h_{all\text{-}0's}$, and also an infinite star-eluder sequence centered at $h_{all\text{-}0's}$.*

*Now we slightly change the setting: Let $\mathcal{X} := \bigcup_{k \in \mathbb{N}} \mathcal{X}_k$ be the disjoint union of finite sets with $|\mathcal{X}_k| = k$ (one may simply assume $\mathcal{X} := \mathbb{N}$). Denote $\mathcal{X}_k := \{x_{k,1}, \ldots, x_{k,k}\}$ and define $h_{k,i}(x) := \mathbb{1}\{x = x_{k,i} \text{ or } x \notin \mathcal{X}_k\}$, for all $1 \le i \le k$. We let $\mathcal{H} := \{h_{k,i}, k \in \mathbb{N}, 1 \le i \le k\}$, and have the following hold:*

1. *$\mathfrak{s}_{h_{all\text{-}0's}} = \infty$: Given arbitrarily large integer $k$, $\{(x_{k,1}, 0), (x_{k,2}, 0), \ldots, (x_{k,k}, 0)\}$ is a star set centered at $h_{all\text{-}0's}$, witnessed by hypotheses $\{h_{k,i}, 1 \le i \le k\}$.*
2. *$\mathfrak{s}_{h_{all\text{-}1's}} = \infty$: Given arbitrarily large integer $k$, $\{(x_{1,1}, 1), (x_{2,1}, 1), \ldots, (x_{k,1}, 1)\}$ is a star set centered at $h_{all\text{-}1's}$, witnessed by hypotheses $\{h_{i,2}, 1 \le i \le k\}$.*
3. *$\mathcal{H}$ has an infinite star-eluder sequence centered at $h_{all\text{-}1's}$: Indeed, $\{(x_{1,1}, 1), (x_{2,1}, 1), \ldots\}$ is an example of infinite star-eluder sequence.*
4. *$\mathcal{H}$ does not have an infinite star-eluder sequence centered at $h_{all\text{-}0's}$.*

**Example 18** (**Infinite star number but no infinite star set**). *We slightly change the setting in Example 17: Let $\mathcal{X} := \bigcup_{k \in \mathbb{N}} \mathcal{X}_k$ be the disjoint union of finite sets with $|\mathcal{X}_k| = k$ (one may again simply assume $\mathcal{X} := \mathbb{N}$). Denote $\mathcal{X}_k := \{x_{k,1}, \ldots, x_{k,k}\}$, let $h_{k,i}(x) := \mathbb{1}\{x = x_{k,i} \text{ or } x \in \mathcal{X}_{>k}\}$, for all $1 \le i \le k$ and $k \in \mathbb{N}$, and let $\mathcal{H} := \{h_{k,i}, 1 \le i \le k, k \in \mathbb{N}\}$. We have the following hold:*

1. *$\mathfrak{s}_{\mathcal{H}} = \infty$ since $\mathcal{H}$ has a star set of arbitrarily large finite size.*
2. *$\mathcal{H}$ does not have an infinite star set.*

*It is worthwhile to mention that in this example, $\mathcal{H}$ does have an infinite star-eluder sequence $\{(x_{1,1}, 0), (x_{2,1}, 0), (x_{2,2}, 0), \ldots\}$ centered at $h_{all\text{-}0's}$. Hence, an infinite star set is an infinite star-eluder sequence, but not the only possibility.*

**Example 19** (**Infinite star number but no infinite star-eluder sequence**). *We slightly change the setting in Example 18 as follow: Let $\mathcal{X} := \bigcup_{k \in \mathbb{N}} \mathcal{X}_k$ be the disjoint union of finite sets with $|\mathcal{X}_k| = k$ (one may again simply assume $\mathcal{X} := \mathbb{N}$). Denote $\mathcal{X}_k := \{x_{k,1}, \ldots, x_{k,k}\}$, let $h_{k,i}(x) := \mathbb{1}\{x = x_{k,i} \text{ or } x \in \mathcal{X}_{<k}\}$, for all $1 \le i \le k$ and $k \in \mathbb{N}$, and let $\mathcal{H} := \{h_{k,i}, 1 \le i \le k, k \in \mathbb{N}\}$. Then the following hold:*

1. *$\mathfrak{s}_{\mathcal{H}} = \infty$ since $\mathcal{H}$ has a star set of arbitrarily large finite size.*
2. *$\mathcal{H}$ does not have an infinite star-eluder sequence, and $SE(\mathcal{H}) < \infty$.*

**Example 20** (**Infinite star-eluder dimension but no infinite star-eluder sequence**). *For any $k \in \mathbb{N}$, let $\mathcal{X}_k := \bigcup_{t \in \mathbb{N}} \mathcal{X}_{k,t}$ be disjoint union of finite sets with $|\mathcal{X}_{k,t}| = k$ for all $t \in \mathbb{N}$. Let $\mathcal{X} := \bigcup_{k \in \mathbb{N}} \mathcal{X}_k$ also with disjoint subspaces $\{\mathcal{X}_k\}_{k \in \mathbb{N}}$. For notation simplicity, let us denote $\mathcal{X}_{k,t} := \{x_{k,t,1}, \ldots, x_{k,t,k}\}$ for all $k, t \in \mathbb{N}$. Now we can define a hypothesis class as follow: let $h_{k,t,j}(x) := \mathbb{1}\{(x = x_{k,t,j}) \vee (x \in \mathcal{X}_{k,>t}) \vee (x \in \mathcal{X}_{<k})\}$, for all $k, t \in \mathbb{N}$ and $1 \le j \le k$, and let $\mathcal{H} := \{h_{k,t,j}, 1 \le j \le k, k, t \in \mathbb{N}\}$. We have the following hold:*

1. *$SE(\mathcal{H}) = \infty$ since for arbitrarily large $k \in \mathbb{N}$, $\mathcal{H}$ has an infinite $k$-star-eluder sequence $\mathcal{X}_k$ with all labels 0.*
2. *$\mathcal{H}$ does not have an infinite (strong) star-eluder sequence.*

**Remark 11.** *Altogether, we have the follow relations*

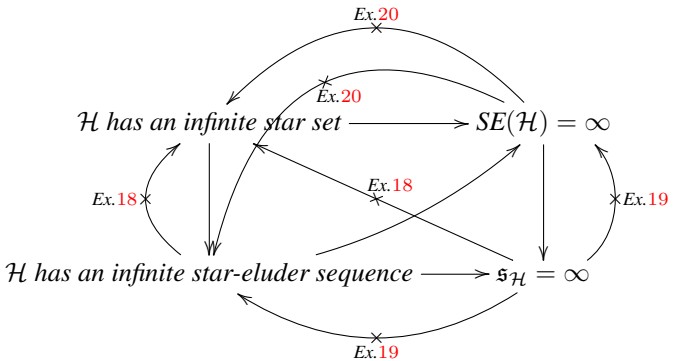

*Remarkably, a complete theory to the relations between these notions is still lacking here, which might be of independent interests.*

## C Fine-grained analysis

In this appendix section, we provide a fine-grained analysis of the asymptotic rate of decay of the universal learning curves by ERM, whenever possible. This will be an analogy to the optimal fine-grained universal learning curves studied in Bousquet et al. (2023). Our characterization of the sharp distribution-free constant factors is based on two newly-introduced combinatorial dimensions named the *star-eluder dimension* or SE dimension and the *VC-eluder dimension* or VCE dimension. We present their formal definitions first.

**Definition 9** (**Star(VC)-eluder dimension**)**.** *Let $\mathcal{H}$ be a concept class, we say $\mathcal{H}$ has an infinite*

- *d-star-eluder sequence $\{(x_1, y_1), (x_2, y_2), \ldots\}$ centered at $h$, if it is realizable and for every $k \in \mathbb{N}$, $\{x_{kd+1}, \ldots, x_{kd+d}\}$ is a star set of $V_{kd}(\mathcal{H})$ centered at $h$. Furthermore, the star-eluder dimension of $\mathcal{H}$, denoted by $SE(\mathcal{H})$, is defined to be the largest integer $d \geq 0$ such that $\mathcal{H}$ has an infinite $d$-star-eluder sequence. If $\mathcal{H}$ does not have any infinite 1-star-eluder sequence, we define $SE(\mathcal{H}) = 0$. If for arbitrarily large integer $d$, $\mathcal{H}$ has an infinite $d$-star-eluder sequence, we define $SE(\mathcal{H}) = \infty$.*
- *d-VC-eluder sequence $\{(x_1, y_1), (x_2, y_2), \ldots\}$ centered at $h$, if it is realizable, and for every $k \in \mathbb{N}$, $h(x_k) = y_k$ and $\{x_{kd+1}, \ldots, x_{kd+d}\}$ is a shattered set of $V_{kd}(\mathcal{H})$. Furthermore, the VC-eluder dimension of $\mathcal{H}$, denoted by $VCE(\mathcal{H})$, is defined to be the largest integer $d \geq 0$ such that $\mathcal{H}$ has an infinite $d$-VC-eluder sequence. If $\mathcal{H}$ does not have any infinite 1-VC-eluder sequence, we define $VCE(\mathcal{H}) = 0$. If for arbitrarily large integer $d$, $\mathcal{H}$ has an infinite $d$-VC-eluder sequence, we define $VCE(\mathcal{H}) = \infty$.*

**Remark 12.** *We recall that the eluder dimension $E(\mathcal{H})$ in Definition 5 represents the length of the longest eluder sequence that exists in $\mathcal{H}$. Indeed, an eluder sequence is exactly one branch of a Littlestone tree, and thus $E(\mathcal{H}) < \infty$ implies that $\mathcal{H}$ has no infinite Littlestone tree. The converse is not true, because $\mathcal{H}$ may have a finite Littlestone tree with some of the branches being infinitely long (see Example 12). Similarly, the star-eluder dimension $SE(\mathcal{H})$ and the VC-eluder dimension $VCE(\mathcal{H})$ here are also strongly related to certain combinatorial structures that have been studied before. In particular for $VCE(\mathcal{H})$, one may refer to the concepts of the (strong) VCL tree, d-VCL tree and the VCL dimension introduced by Bousquet et al. (2021, 2023). Indeed, an infinite (strong) VC-eluder sequence is exactly one branch of a strong VCL tree, and an infinite $d$-VC-eluder sequence is exactly one branch of an infinite $d$-VCL tree. Since an infinite 1-VCL-tree is exactly an infinite Littlestone tree, an infinite 1-VC-eluder sequence is thus exactly an infinite eluder sequence. Moreover, recall that $VCL(\mathcal{H}) = 0$ implies that $\mathcal{H}$ does not have an infinite Littlestone tree, and similarly, here we have $VCE(\mathcal{H}) = 0$ implies that $\mathcal{H}$ does not have an infinite eluder sequence.*

**Remark 13.** *For any concept class $\mathcal{H}$, the following hold:*

1. *$E(\mathcal{H}) \geq SE(\mathcal{H}) \geq VCE(\mathcal{H})$.*
2. *$VCE(\mathcal{H}) \geq 1 \iff SE(\mathcal{H}) \geq 1 \iff E(\mathcal{H}) = \infty$.*
3. *$VCE(\mathcal{H}) = 0 \iff SE(\mathcal{H}) = 0 \iff E(\mathcal{H}) < \infty$.*

We then state the formal definition of the fine-grained universal rates by ERM.

**Definition 10** (**Fine-grained universal rates by ERM**). *Let $\mathcal{H}$ be a concept class and $R(n) \to 0$ be a distribution-free rate function. We say*

- $\mathcal{H}$ *is universally learnable at fine-grained rate $R$ by ERM, if for every distribution $P \in RE(\mathcal{H})$, there exists a distribution-dependent rate $\lambda(n) = o\left(R(n)\right)$ such that for every ERM algorithm, $\mathbb{E}[er_P(\hat{h}_n)] \le R(n) + \lambda(n)$, for all $n \in \mathbb{N}$.*
- $\mathcal{H}$ *is not universally learnable at fine-grained rate faster than $R$ by ERM, if there exists a distribution $P \in RE(\mathcal{H})$ such that there is an ERM algorithm satisfying $\mathbb{E}[er_P(\hat{h}_n)] \ge R(n)$, for infinitely many $n \in \mathbb{N}$.*
- $\mathcal{H}$ *is universally learnable with exact fine-grained rate $R$ by ERM, if $\mathcal{H}$ is universally learnable at fine-grained rate $R$ by ERM, and is not universally learnable at fine-grained rate faster than $R$ by ERM.*

Note that the crucial difference between this definition and Definition 2 is that here $R(n)$ is independent of the data distribution $P$. In other words, the fine-grained rates provide optimal distribution-free upper and lower envelopes of the universal learning curves up to numerical constant factors.

**Remark 14.** *Definition 10 describes special cases of Definition 2 in the following sense: If $\mathcal{H}$ is universally learnable at fine-grained rate (no faster than) $R$ by ERM, then it is universally learnable at rate (no faster than) $R$ by ERM as well. Briefly speaking, the fine-grained analysis aims to find the correct characterization that captures the optimal distribution-free upper envelope and lower envelope of all the distribution-dependent learning curves, tight up to numerical constant factors.*

We now turn to state our results of fine-grained universal rates by ERM. All technical aspects of the proofs are deferred to Appendix D.3.

**Theorem 8** (**Fine-grained learning rates**). *For every class $\mathcal{H}$ with $|\mathcal{H}| \ge 3$, the following hold:*

- *If $VCE(\mathcal{H}) < \infty$, then $\mathcal{H}$ is universally learnable at fine-grained rate $\frac{VCE(\mathcal{H}) \log n}{n}$, and is not universally learnable at fine-grained rate faster than $\frac{VCE(\mathcal{H})}{n}$, by ERM.*
- *If $SE(\mathcal{H}) < \infty$, then $\mathcal{H}$ is universally learnable at fine-grained rate $\frac{VCE(\mathcal{H})}{n} \log\left(\frac{SE(\mathcal{H})}{VCE(\mathcal{H})}\right)$, but is not universally learnable at fine-grained rate faster than $\frac{VCE(\mathcal{H}) + \log(SE(\mathcal{H}))}{n}$, by ERM.*

*or equivalently, there exist finite numerical constants $\alpha, \beta > 0$ such that*

- *If $VCE(\mathcal{H}) < \infty$, then*

$$\mathbb{E}\left[er_P(\hat{h}_n)\right] \ge \alpha \cdot \frac{VCE(\mathcal{H})}{n}, \quad \text{for infinitely many } n \in \mathbb{N}, \tag{1}$$

$$\mathbb{E}\left[er_P(\hat{h}_n)\right] \le \beta \cdot \frac{VCE(\mathcal{H}) \log n}{n} + 2^{-\lfloor n/2\kappa \rfloor}, \ \forall n \in \mathbb{N}, \tag{2}$$

*where $\kappa = \kappa(P)$ is a distribution-dependent constant.*
- *If $SE(\mathcal{H}) < \infty$, then*

$$\mathbb{E}\left[er_P(\hat{h}_n)\right] \ge \alpha \cdot \frac{VCE(\mathcal{H}) + \log(SE(\mathcal{H}))}{n}, \quad \text{for infinitely many } n \in \mathbb{N}, \tag{3}$$

$$\mathbb{E}\left[er_P(\hat{h}_n)\right] \le \beta \cdot \frac{VCE(\mathcal{H})}{n} \log\left(\frac{SE(\mathcal{H})}{VCE(\mathcal{H})}\right) + 2^{-\lfloor n/2\hat{\kappa} \rfloor}, \ \forall n \in \mathbb{N}, \tag{4}$$

*where $\hat{\kappa} = \hat{\kappa}(P)$ is a distribution-dependent constant.*

**Remark 15.** *Our proofs use $\alpha = 1/20$ and $\beta = 160$.*

**Remark 16.** *When $SE(\mathcal{H}) = VCE(\mathcal{H}) = 0$, $E(\mathcal{H}) < \infty$, or $SE(\mathcal{H}) = VCE(\mathcal{H}) = 1$, $E(\mathcal{H}) = \infty$, we still have the bounds (3) and (4) since we define $\log(0) = 0$, $0 \log(0/0) = 0$ and $\log(x) := \log(x \vee 2)$ for any $x > 0$. Moreover, we remark that neither $VCE(\mathcal{H}) = \infty$ nor $SE(\mathcal{H}) = \infty$ is considered in these fine-grained rates. This is because when $VCE(\mathcal{H}) = \infty$, arbitrarily slow rates cannot admit distribution-free constants. And when $SE(\mathcal{H}) = \infty$, it is still impossible because it does not guarantee an infinite star-eluder sequence, that is, the lower bound of (1) cannot be increased to $\log(n)/n$, and is the sharpest one we can have here.*

**Remark 17.** *It is not hard to understand that a target-specified version of fine-grained universal rates by ERM is also derivable, based on a centered version of the star-eluder dimension $SE_{h^*}$ and VC-eluder dimension $VCE_{h^*}$.*

**Remark 18.** *It is worth noting that, when $SE(\mathcal{H}) < \infty$, there is a mismatch between the lower bound and the upper bound. This serves as an analogy to the mismatch between Cor.12 and Thm.13 in Hanneke (2016b), and certain demonstrating examples have been exhibited in Hanneke and Yang (2015). In the following two examples, we provide evidence that such a gap does exist, in a sense that both the upper bound and the lower bound can sometimes be tight for some classes. Roughly speaking, if an infinite $SE(\mathcal{H})$-star-eluder sequence in $\mathcal{H}$ is also an infinite $VCE(\mathcal{H})$-VC-eluder sequence (see Definition 9), then $\frac{VCE(\mathcal{H})}{n} \log \left(\frac{SE(\mathcal{H})}{VCE(\mathcal{H})}\right)$ is the optimal rate, otherwise $\frac{VCE(\mathcal{H}) + \log (SE(\mathcal{H}))}{n}$ is optimal.*

**Example 21** (**Optimal $(\mathbf{VCE}(\mathcal{H}) + \log (\mathbf{SE}(\mathcal{H})))/n$ rate**). *We construct a concept class $\mathcal{H}$ such that an infinite $VCE(\mathcal{H})$-VC-eluder sequence and an infinite $SE(\mathcal{H})$-star-eluder sequence cannot be realized by an infinite sequence. To this end, we slightly change the example presented in Appendix D.2 of Hanneke and Yang (2015), which yields the tightness of a lower bound $(VC(\mathcal{H}) + \log (\mathfrak{s}_{\mathcal{H}}))/n$.*

*Specifically, let $d, s > 0$ be two integers satisfying $d \leq s$. Let $\mathcal{X} := \mathbb{Z} \setminus \{0\} := \mathcal{X}_1 \cup \mathcal{X}_2$, where $\mathcal{X}_1 := \mathbb{N} \setminus \{0\}$ and $\mathcal{X}_2 := -\mathbb{N} \setminus \{0\} = -\mathcal{X}_1$. We can also write*

*$\mathcal{X}_1 = (\mathcal{X}_{1,0} \cup \mathcal{X}_{1,1} \cup \cdots)$, where $\mathcal{X}_{1,k} := \{ks + 1, \ldots, (k+1)s\}$ for all $k \in \mathbb{N}$,*

*$\mathcal{X}_2 = (\mathcal{X}_{2,0} \cup \mathcal{X}_{2,1} \cup \cdots)$, where $\mathcal{X}_{2,k} := \{-(k+1)d, \ldots, -kd - 1\}$ for all $k \in \mathbb{N}$.*

*Now we let $\mathcal{H} := \mathcal{H}_1 \cup \mathcal{H}_2$ satisfying $VCE(\mathcal{H}) = d$ and $SE(\mathcal{H}) = s$, where*

*$\mathcal{H}_1 := \{h_{k,j} : \forall j \in \mathcal{X}_{1,k}, \forall k \in \mathbb{N}\}$, where $h_{k,j}(x) := \mathbb{1}(x = j \text{ or } x \in \mathcal{X}_{1,>k})$.*

*$\mathcal{H}_2 := \{h_{k,S} : \forall S \subseteq \mathcal{X}_{2,k}, \forall k \in \mathbb{N}\}$, where $h_{k,S}(x) := \mathbb{1}(x \in S \text{ or } x \in \mathcal{X}_{2,>k})$.*

*In particular, $\mathcal{X}_1$ itself is an infinite $s$-star-eluder sequence centered at $h_{all-0's}$, and $\mathcal{X}_2$ itself is an infinite $d$-VC-eluder sequence, but they do not intersect. To show that the upper bound can be decreased to match the lower bound, we simply note that for any infinite $s$-star-eluder sequence, its associated VC-eluder dimension is exactly 1, resulting in a $\log (s)/n$ upper bound. For any infinite $d$-VC-eluder sequence, its associated star-eluder dimension is also $d$, resulting in a $d/n$ upper bound. The maximum of the two upper bounds yields the desired one.*

**Example 22** (**Optimal $(\mathbf{VCE}(\mathcal{H})/n) \log (\mathbf{SE}(\mathcal{H})/\mathbf{VCE}(\mathcal{H}))$ rate**). *We construct a concept class $\mathcal{H}$ such that there exists an infinite sequence in $\mathcal{H}$ which is both an infinite $VCE(\mathcal{H})$-VC-eluder sequence and an infinite $SE(\mathcal{H})$-star-eluder sequence. To this end, we slightly change the example presented in Appendix D.1 of Hanneke and Yang (2015), which yields the tightness of an upper bound $(VC(\mathcal{H})/n) \log (\mathfrak{s}_{\mathcal{H}}/VC(\mathcal{H}))$.*

*Specifically, let $d, s > 0$ be two integers satisfying $d \leq s$. Let $\mathcal{X} := \mathbb{N}$ and for every $k \in \mathbb{N}$, define $h_{k,S}(x) := \mathbb{1}(x \in S \text{ or } x > (k+1)s)$ for every subset $S \subseteq \{ks + 1, \ldots, (k+1)s\}$ with $|S| \leq d$. Let $\mathcal{H} := \{h_{k,S} : S \subseteq \{ks + 1, \ldots, (k+1)s\}, |S| \leq d, k \in \mathbb{N}\}$. Note that for this class, we have $VCE(\mathcal{H}) = d$, $SE(\mathcal{H}) = s$ and there exists an infinite sequence serving as an infinite $d$-VC-eluder sequence as well as an infinite $s$-star-eluder sequence. To show in this case that the lower bound can be increased to match the upper bound, the realizable distribution that witnesses this rate is referred to Appendix D.1.1 of Hanneke and Yang (2015).*

Now let us turn to the proof of Theorem 8, which is based on the following two lemmas and within each an upper bound as well as a lower bound are established.

**Lemma 10.** *For every concept class $\mathcal{H}$ with $|\mathcal{H}| \geq 3$, if $VCE(\mathcal{H}) < \infty$, then the following hold:*

$$\mathbb{E}\left[er_P(\hat{h}_n)\right] \geq \frac{VCE(\mathcal{H})}{18n}, \text{ for infinitely many } n \in \mathbb{N},$$

$$\mathbb{E}\left[er_P(\hat{h}_n)\right] \leq \frac{28 VCE(\mathcal{H}) \log n}{n} + 2^{-\lfloor n/2\kappa \rfloor}, \ \forall n \in \mathbb{N},$$

*where $\kappa = \kappa(P)$ is a distribution-dependent constant.*

**Remark 19.** *Note that a concept class with its VCE dimension finite can either have an infinite star-eluder sequence or not, which results in a difference (of a logarithmic factor) in the upper bound and lower bound stated in Lemma 10.*

Recall that $VC(\mathcal{H}) < \infty$ yields a uniform upper bound $VC(\mathcal{H}) \log(n)/n$. On one hand, $VCE(\mathcal{H}) = \infty$ implies that $\mathcal{H}$ has a shattered set of arbitrarily large size, which further implies an unbounded VC dimension. On the other hand, according to Lemma 9, $VC(\mathcal{H}) = \infty$ implies that $\mathcal{H}$ has an infinite VC-eluder sequence, and thus $VCE(\mathcal{H}) = \infty$ holds as well. Therefore, $VCE(\mathcal{H}) = \infty$ if and only if $VC(\mathcal{H}) = \infty$ if and only if $\mathcal{H}$ has an infinite VC-eluder sequence. Moreover, when $VCE(\mathcal{H}) < \infty$, a trivial observation is $VCE(\mathcal{H}) \leq VC(\mathcal{H}) < \infty$. However, the following example reveals that VCE and VC are not the same dimension, namely, there exists a class $\mathcal{H}$ having strictly $VCE(\mathcal{H}) < VC(\mathcal{H})$ (see the following Example 23). Therefore, Lemma 10 sometimes reflects an improvement over the classical uniform bound.

**Example 23** ($\mathbf{VCE}(\mathcal{H}) < \mathbf{VC}(\mathcal{H}) < \infty$)**.** *To make it more convincing, we provide an example of infinite classes here. Let $\mathcal{X}_1$ be a finite set of size $d$, and $\mathcal{X}_2$ be an infinite instance space that is disjoint with $\mathcal{X}_1$. For simplicity, one may assume that $\mathcal{X}_1 := \{-d, -(d-1), \dots, -1\}$ and $\mathcal{X}_2 := \mathbb{N}$. We define $\mathcal{X} := \mathcal{X}_1 \cup \mathcal{X}_2$ and let $\mathcal{H} := \{h_{S,k} := \mathbb{1}_{S \cup \{k\}}, \forall S \subseteq \mathcal{X}_1, \forall k \in \mathbb{N}\}$. This class has $VC(\mathcal{H}) = (d+1)$ but $VCE(\mathcal{H}) = 1$ since there is no infinite 2-VC-eluder sequence. Similarly, we can also construct an example that witnesses strictly $SE(\mathcal{H}) < \mathfrak{s}_{\mathcal{H}} < \infty$.*

**Lemma 11.** *For every concept class $\mathcal{H}$ with $|\mathcal{H}| \geq 3$, if $SE(\mathcal{H}) < \infty$, then the following hold:*

$$\mathbb{E}\left[er_P(\hat{h}_n)\right] \geq \frac{\log(SE(\mathcal{H}))}{12n}, \text{ for infinitely many } n \in \mathbb{N},$$

$$\mathbb{E}\left[er_P(\hat{h}_n)\right] \leq \frac{160 VCE(\mathcal{H})}{n} \log\left(\frac{SE(\mathcal{H})}{VCE(\mathcal{H})}\right) + 2^{-\lfloor n/2\hat{\kappa}\rfloor}, \ \forall n \in \mathbb{N},$$

*where $\hat{\kappa} = \hat{\kappa}(P)$ is a distribution-dependent constant.*

**Remark 20.** *Note that in Theorem 8, the lower bound appears as $\frac{VCE(\mathcal{H}) + \log(SE(\mathcal{H}))}{n}$. Indeed, $SE(\mathcal{H}) < \infty$ immediately implies $VCE(\mathcal{H}) < \infty$ and then the lower bound in Lemma 10 holds. Combining with the lower bound in Lemma 11 will give us the desired result in Theorem 8 with some sufficiently small constant, e.g. $\alpha = 1/20$.*

*Remarkably, when both $SE(\mathcal{H})$ and $VCE(\mathcal{H})$ are finite, either of $VCE(\mathcal{H})$ and $\log(SE(\mathcal{H}))$ can be larger than the other, and we provide the following examples for evidence. Therefore, none of the quantities can be removed in the lower bound.*

**Example 24** ($\mathbf{VCE}(\mathcal{H}) < \log(\mathbf{SE}(\mathcal{H})) < \infty$)**.** *Let $\mathcal{X} := \bigcup_{k \in \mathbb{N}} \mathcal{X}_k$ be the disjoint union of finite sets with $|\mathcal{X}_k| = d < \infty$. We denote $\mathcal{X}_k := \{x_{k,1}, \dots, x_{k,d}\}$, for every $k \in \mathbb{N}$, let $h_{k,j}(x) := \mathbb{1}\{x = x_{k,j} \text{ or } x \in \mathcal{X}_{>k}\}$, for every $k \in \mathbb{N}$ and every $1 \leq j \leq d$, and finally let $\mathcal{H} := \{h_{k,j}, 1 \leq j \leq d, k \in \mathbb{N}\}$. For this class, we have $VCE(\mathcal{H}) = 2 < \log(SE(\mathcal{H})) = \log d$ for a sufficiently large $d$.*

**Example 25** ($\log(\mathbf{SE}(\mathcal{H})) < \mathbf{VCE}(\mathcal{H}) < \infty$)**.** *Let $\mathcal{X} := \bigcup_{k \in \mathbb{N}} \mathcal{X}_k$ be the disjoint union of finite sets with $|\mathcal{X}_k| = d < \infty$. We denote $\mathcal{X}_k := \{x_{k,1}, \dots, x_{k,d}\}$, for every $k \in \mathbb{N}$, let $h_{k,S}(x) := \mathbb{1}\{x \in S \text{ or } x \in \mathcal{X}_{>k}\}$, for every $k \in \mathbb{N}$ and every subset $S \subseteq \mathcal{X}_k$, and finally let $\mathcal{H} := \{h_{k,S}, S \subseteq \mathcal{X}_k, k \in \mathbb{N}\}$. For this class, we have $\log(SE(\mathcal{H})) = \log d < VCE(\mathcal{H}) = d$.*

# D Proofs

## D.1 Omitted Proofs in Section 3

**Proposition 3** (**Proposition 1 restated**)**.** *Any infinite concept class $\mathcal{H}$ has either an infinite star-eluder sequence or infinite Littlestone dimension.*

To prove the proposition, we first introduce a new complexity structure named the *threshold sequence*.

**Definition 11** (**Threshold sequence**)**.** *Let $\mathcal{H}$ be a concept class, we say that $\mathcal{H}$ has an infinite threshold sequence $\{(x_1, y_1), (x_2, y_2), \dots\}$, if it is realizable and for every integer $k$, there exists $h_k \in \mathcal{H}$ such that $h_k(x_i) = y_i$ for all $i < k$ and $h_k(x_i) \neq y_i$ for all $i \geq k$. We say an infinite threshold sequence $\{(x_1, y_1), (x_2, y_2), \dots\}$ is underlined{centered at $h$}, if $h(x_i) = y_i$ for all $i \in \mathbb{N}$.*

The following claim turns out to be an alternative result to the proposition.

**Claim 1.** *Any infinite class $\mathcal{H}$ has either an infinite star set or an infinite threshold sequence.*

Given that the claim holds, the proof to the proposition is then straightforward. This is because an infinite star set itself is an infinite star-eluder sequence. Moreover, an infinite threshold sequence

gives rise to infinite Littlestone dimension since $\mathcal{H}$ can have a Littlestone tree of arbitrarily large depth (an easy example is $\mathcal{H}_{\text{thresh},\mathbb{N}}$). Therefore, it suffices to prove Claim 1. The remaining proof relies on a connection to the classical Ramsey theory, which we briefly introduced as follow.

The classical Ramsey's theorem states that one will find monochromatic cliques in any edge labelling (with colors) of a sufficiently large complete graph. Specifically, let $r$ be an positive integer, a simple 2-colors version of the Ramsey's theorem states that there exists a smallest positive integer $R(r, r)$, named the (diagonal) Ramsey number, such that every red-blue edge coloring of the complete graph on $R(r, r)$ vertices contains either a red clique on $r$ vertices or a blue clique on $r$ vertices. However, we will need the following extension of the theorem to an infinite graph.

**Theorem 9 (Infinite Ramsey's theorem, Ramsey, 1987).** *For any countably infinite set, if its induced complete graph is colored with finitely many colors, then there is an infinite monochromatic clique.*

*Proof of Claim 1.* Based on Lemma 8, we know that any infinite class $\mathcal{H}$ has an infinite eluder sequence. Let $\{(x_1, y_1), (x_2, y_2), \ldots\}$ be an infinite eluder sequence centered at $h^*$, that is, for any $j \in \mathbb{N}$, there exists $h_j \in \mathcal{H}$ such that $h_j(x_i) = y_i = h^*(x_i)$ for all $i < j$ and $h_j(x_j) \neq y_j = h^*(x_j)$. We aim to show that there exists an infinite subsequence $\{(x_{i_1}, y_{i_1}), (x_{i_2}, y_{i_2}), \ldots\}$ that is either an infinite star set centered at $h^*$ or an infinite threshold sequence centered at $h^*$. To this end, we consider the infinite eluder sequence $\{(x_1, y_1), (x_2, y_2), \ldots\}$ as a red-blue coloring of an infinite complete graph according to the following: let the vertices be indexed by $\mathbb{N}$, then for every edge $e_{i,j}$ with integers $i > j$, we color it red if $h_j(x_i) = y_i$ and blue otherwise.

Note that for any infinite subsequence $\{(x_{i_1}, y_{i_1}), (x_{i_2}, y_{i_2}), \ldots\}$, if the infinite subgraph comprised of the vertices $\{i_1, i_2, \ldots\}$ is monochromatically red, then $h_{i_j}(x_{i_k}) = y_{i_k} = h^*(x_{i_k})$ for all integers $k > j$. Since $h_{i_j}(x_{i_k}) = y_{i_k} = h^*(x_{i_k})$ for all integers $k < j$ and $h_{i_j}(x_{i_j}) \neq y_{i_j} = h^*(x_{i_j})$, it implies that $\{(x_{i_1}, y_{i_1}), (x_{i_2}, y_{i_2}), \ldots\}$ is an infinite star set centered at $h^*$, witnessed by $\{h_{i_j}\}_{j \in \mathbb{N}}$. Moreover, if the infinite subgraph comprised of the vertices $\{i_1, i_2, \ldots\}$ is monochromatically blue, it is not hard to verify that $\{(x_{i_1}, y_{i_1}), (x_{i_2}, y_{i_2}), \ldots\}$ is an infinite threshold sequence centered at $h^*$. The proof is completed by applying the infinite Ramsey's theorem. $\square$

**Lemma 12 (Lemma 1 restated).** *Given a concept class $\mathcal{H}$, for any learning algorithm $\hat{h}_n$, there exists a realizable distribution $P$ with respect to $\mathcal{H}$ such that $\mathbb{E}[er_P(\hat{h}_n)] \geq 2^{-(n+2)}$ for infinitely many $n$, which implies that $\mathcal{H}$ is not universally learnable at rate faster than exponential $e^{-n}$.*

*Proof of Lemma 12.* We prove the lemma by using the "probabilistic method". Let us consider non-trivially that $|\mathcal{H}| > 2$, let $h_1, h_2 \in \mathcal{H}$ and $x, x^{'} \in \mathcal{X}$ such that $h_1(x) = h_2(x) = y$ and $h_1(x^{'}) \neq h_2(x^{'})$. Now for any learning algorithm $\hat{h}_n$, we define the following two realizable distributions $P_0$ and $P_1$, where $P_i\{(x, y)\} = 0.5$ and $P_i\{(x^{'}, i)\} = 0.5$, $i \in \{0, 1\}$. Let $I \sim \text{Bernoulli}(0.5)$, and conditioned on $I$, let $S_n := \{(x_1, y_1), (x_2, y_2), \ldots, (x_n, y_n)\}$ and $(x_{n+1}, y_{n+1})$ be i.i.d. samples from $P_I$ that the learning algorithm $\hat{h}_n$ is trained on. We note that

$$\mathbb{E}\left[\mathbb{P}\left(\hat{h}_n(x_{n+1}) \neq y_{n+1} | S_n, I\right)\right] \geq \frac{1}{2}\mathbb{P}\left(x_1 = \ldots = x_n = x, x_{n+1} = x^{'}\right) = 2^{-(n+2)}.$$

Furthermore, by the law of total probability, we have

$$\mathbb{E}\left[\mathbb{P}\left(\hat{h}_n(x_{n+1}) \neq y_{n+1} | S_n, I\right)\right] = \frac{1}{2}\sum_{i \in \{0,1\}} \mathbb{E}\left[\mathbb{P}\left(\hat{h}_n(x_{n+1}) \neq y_{n+1} | S_n, I = i\right) | I = i\right]$$

$$\leq \max_{i \in \{0,1\}} \mathbb{E}\left[\mathbb{P}\left(\hat{h}_n(x_{n+1}) \neq y_{n+1} | S_n, I = i\right) | I = i\right].$$

The above two inequalities imply that for every $n$, there exists $i_n \in \{0, 1\}$ such that

$$\mathbb{E}\left[\mathbb{P}\left(\hat{h}_n(x_{n+1}) \neq y_{n+1} | S_n, I = i_n\right) | I = i_n\right] = \mathbb{E}[er_{P_{i_n}}(\hat{h}_n)] \geq 2^{-(n+2)}.$$

In particular, by the pigeonhole principle, there exists $i \in \{0, 1\}$ such that $i_n = i$ infinitely often, which completes the proof. $\square$

**Lemma 13 (Lemma 2 restated).** *If $\mathcal{H}$ does not have an infinite eluder sequence centered at $h^*$, then $h^*$ is universally learnable by ERM at rate $e^{-n}$.*

*Proof of Lemma 13.* Since $\mathcal{H}$ does not have an infinite eluder sequence centered at $h^*$, then for any realizable distribution $P$ centered at $h^*$ and data sequence $S := \{(x_1, h^*(x_1)), (x_2, h^*(x_2)), \ldots\} \sim P^{\mathbb{N}}$, we have

$$\# \left\{ t \in \mathbb{N} : \exists t^{'} > t \text{ s.t. } \exists h \in V_{S_t}(\mathcal{H}) : h(x_{t'}) \neq h^*(x_{t'}) \right\} < \infty.$$

For the largest such integer $t$, we further have $\mathbb{P}(\exists h \in V_{S_t}(\mathcal{H}) : h(x_{t'}) \neq h^*(x_{t'})) = 1$ for some $t^{'} := t^{'}(S) > t$. This is true because the probability decays exponentially. Therefore, we have

$$\lim_{n \to \infty} \mathbb{P}_{S \sim P^{\mathbb{N}}} \left( P \left( x \in \mathcal{X} : \exists h \in V_n(\mathcal{H}) \text{ s.t. } h(x) \neq h^*(x) \right) = 0 \right) = 1,$$

which implies that there is a distribution-dependent positive integer $k := k(P) < \infty$ such that

$$\mathbb{P} \left( P \left( x \in \mathcal{X} : \exists h \in V_k(\mathcal{H}) \text{ s.t. } h(x) \neq h^*(x) \right) = 0 \right) \geq 1/2.$$

Now for any integer $n > k$, we split the dataset $S_n \sim P^n$ into $\lfloor n/k \rfloor$ parts with each one sized at least $k$, denoted by $S_{n,1}, \ldots, S_{n,\lfloor n/k \rfloor}$. It holds then

$$\begin{aligned}
&\mathbb{P} \left( P \left( x \in \mathcal{X} : \exists h \in V_n(\mathcal{H}) \text{ s.t. } h(x) \neq h^*(x) \right) \neq 0 \right) \\
\leq &\mathbb{P} \left( \forall i \in \{1, \ldots, \lfloor n/k \rfloor\} : P \left( x \in \mathcal{X} : \exists h \in V_{S_{n,i}}(\mathcal{H}) \text{ s.t. } h(x) \neq h^*(x) \right) \neq 0 \right) \\
= &\prod_{i=1}^{\lfloor n/k \rfloor} \mathbb{P} \left( P \left( x \in \mathcal{X} : \exists h \in V_{S_{n,i}}(\mathcal{H}) \text{ s.t. } h(x) \neq h^*(x) \right) \neq 0 \right) \leq 2^{-\lfloor n/k \rfloor},
\end{aligned}$$

which also holds for $n \leq k$. Finally, it follows that

$$\begin{aligned}
\mathbb{E} \left[ \text{er}_P(\hat{h}_n) \right] \leq &\mathbb{P} \left( \exists h \in V_n(\mathcal{H}) : \text{er}_P(h) > 0 \right) \\
= &\mathbb{P} \left( P \left( x \in \mathcal{X} : \exists h \in V_n(\mathcal{H}) \text{ s.t. } h(x) \neq h^*(x) \right) \neq 0 \right) \leq 2^{-\lfloor n/k \rfloor}, \quad \forall n \in \mathbb{N}.
\end{aligned}$$

$\square$

**Lemma 14** (**Lemma 3 restated**). *If $\mathcal{H}$ has an infinite eluder sequence centered at $h^*$, then $h^*$ is not universally learnable by ERM at rate faster than $1/n$.*

*Proof of Lemma 14.* Let $\{(x_1, y_1), (x_2, y_2), \ldots\}$ be an infinite eluder sequence centered at $h^*$, we consider the following distribution $P$: $P\{(x_i, y_i)\} = 2^{-i}$ and $P\{(x_i, 1-y_i)\} = 0$ for all $i \in \mathbb{N}$. Note that $P$ is realizable (with respect to $\mathcal{H}$) with target $h^*$. Given a dataset $S_n := \{(x_i, y_i)\}_{i=1}^n \sim P^n$, let the worst-case ERM outputs $\hat{h}_n := \text{ERM}(S_n)$. For any $t \in \mathbb{N}$, if $S_n$ does not contain any copy of the points in $\{x_i, i > t\}$, we have $\text{er}_P(\hat{h}_n) \geq 2^{-t}$. The probability of such event is

$$\mathbb{P} \left( \sum_{i=1}^n \mathbb{1}\{X_i \in \{x_{t+1}, x_{t+2}, \ldots\}\} = 0 \right) = \prod_{i=1}^n \mathbb{P}(X_i \in \{x_1, \ldots, x_t\}) = \left( 1 - 2^{-t} \right)^n.$$

Therefore, it follows immediately that

$$\mathbb{E} \left[ \text{er}_P(\hat{h}_n) \right] \geq \sum_{t=1}^{\infty} 2^{-t} \left( 1 - 2^{-t} \right)^n \geq \frac{1}{n} \left( 1 - \frac{2}{n} \right)^n \geq \frac{1}{9n},$$

where the second inequality follows from choosing $t = \lfloor \log n \rfloor$. $\square$

**Lemma 15** (**Lemma 4 restated**). *If $\mathcal{H}$ does not have an infinite star-eluder sequence centered at $h^*$, then $h^*$ is universally learnable by ERM at rate $1/n$.*

Before proceeding to the proof of Lemma 15, we first introduce several useful tools. The following definition of the sample compression scheme was originally stated in Littlestone and Warmuth (1986).

**Definition 12** (**Sample compression scheme**). *Let $\mathcal{H}$ be a concept class and $S_n := \{(x_i, y_i)\}_{i=1}^n$. A sample compression scheme for $\mathcal{H}$ consists of two maps $(\kappa, \rho)$ such that the following hold:*

— *The compression map $\kappa$ takes $S_n$ to $T := \kappa(S_n)$, for some $T \in \bigcup_{t=0}^{\infty} \{(x, y) \in S_n\}^t$.*
— *The reconstruction function $\rho$ takes $T$ to $\rho(T) : \mathcal{X} \to \{0, 1\}$.*

*The size of the sample compression scheme $(\kappa, \rho)$ is defined as $\max_{S_n \in (\mathcal{X} \times \{0,1\})^n} |\kappa(S_n)|$. A sample compression scheme $(\kappa, \rho)$ is called sample-consistent for $\mathcal{H}$, if for any realizable distribution $P$ with respect to $\mathcal{H}$ and $S_n := \{(x_i, y_i)\}_{i=1}^n \sim P^n$, it holds that $\hat{er}_{S_n}(\rho(\kappa(S_n))) = 0$. A sample compression scheme $(\kappa, \rho)$ is called stable if for every subsequence $S'$ satisfying $\kappa(S_n) \subseteq S' \subset S_n$, it holds that $\rho(\kappa(S')) = \rho(\kappa(S_n))$, that is, removing any non-compression point from $S_n$ does not change the classifier returned by the sample compression scheme.*

**Definition 13** (**Version space compression set**). *Let $\mathcal{H}$ be a concept class and $P$ be a realizable distribution with respect to $\mathcal{H}$. For any $n \in \mathbb{N}$, and any dataset $S_n := \{(x_i, y_i)\}_{i=1}^n \sim P^n$, the version space compression set $\hat{\mathcal{C}}_n$ is defined to be the smallest subset of $S_n$ satisfying $V_{S_n}(\mathcal{H}) = V_{\hat{\mathcal{C}}_n}(\mathcal{H})$. Furthermore, we define the version space compression set size as $\hat{n}(S_n) := |\hat{\mathcal{C}}_n|$, which is a data-dependent quantity. Finally, we define $\hat{n}_{1:n} := \max_{1 \le m \le n} \hat{m}(S_m)$, which is also data-dependent. However, $\hat{n}_{1:n}$ is not only just dependent on the full sample, but is also dependent on any prefix of the sample (that is, the order of the sample).*

**Remark 21.** *It has been argued in* Wiener et al. (2015) *that the region of disagreement of the version space $DIS(V_n(\mathcal{H}))$ can be described as a compression scheme, where the size of the compression scheme is exactly the version space compression set size $\hat{n}(S_n)$.*

With these definitions in hand, we are now able to prove the lemma.

*Proof of Lemma 15.* Let $P$ be a realizable distribution with respect to $\mathcal{H}$ centered at $h^*$, let $S_n := \{(x_i, y_i)\}_{i=1}^n \sim P^n$ be a dataset and $\hat{C}_n \subseteq S_n$ be the corresponding version space compression set with size $|\hat{C}_n| = \hat{n}(S_n)$. We let $(\kappa, \rho)$ be a sample compression scheme of size $\hat{n}(S_n)$ defined by $\kappa(S_n) = \hat{C}_n$ and $\rho(\hat{C}_n) = \hat{h}_n$. Since any ERM algorithm will output predictors $\{\hat{h}_n\}_{n \in \mathbb{N}}$ satisfying $\hat{h}_n \in V_n(\mathcal{H})$, it is clear that

$$\hat{er}_{S_n}(\rho(\kappa(S_n))) = \sum_{i=1}^n \mathbb{1}\{\rho(\kappa(S_n))(x_i) \ne y_i\} = \sum_{i=1}^n \mathbb{1}\left\{\hat{h}_n(x_i) \ne y_i\right\} = 0,$$

and thus it is sample-consistent. Furthermore, let $S'$ be any subsequence satisfying $\hat{C}_n \subseteq S' \subset S_n$. On one hand, we have $V_{S_n}(\mathcal{H}) \subseteq V_{S'}(\mathcal{H})$. On the other hand, we also have $V_{S'}(\mathcal{H}) \subseteq V_{\hat{C}_n}(\mathcal{H}) = V_{S_n}(\mathcal{H})$. Therefore, we conclude $V_{S_n}(\mathcal{H}) = V_{S'}(\mathcal{H})$, and thus $\rho(\kappa(S')) = \rho(\kappa(S_n))$, that is, the compression scheme $(\kappa, \rho)$ is also stable. Now we can apply Lemma 24, and then obtain

$$\mathbb{E}\left[er_P(\hat{h}_n)\right] = \mathbb{E}\left[er_P(\rho(\kappa(S_n)))\right] \le \frac{\mathbb{E}[\hat{n}(S_n)]}{n+1}.$$

Indeed, it has been proved that $\hat{n}(S_n) \le \mathfrak{s}_{h^*}$ (Thm.13 Hanneke and Yang, 2015), and for completeness, we prove it as in Lemma 25 in Appendix E. The only remaining concern is that the fact "$\mathcal{H}$ does not have an infinite star-eluder sequence centered at $h^*$" does not guarantee $\mathfrak{s}_{h^*} < \infty$. However, it essentially states that the version space will eventually have a bounded star number centered at $h^*$. Since $\mathcal{H}$ does not have an infinite star-eluder sequence centered at $h^*$, for any sequence $S := \{(x_1, y_1), (x_2, y_2), \ldots\} \sim P^{\mathbb{N}}$, there exists a data-dependent integer $\bar{k} := \bar{k}(S) < \infty$ such that $\mathfrak{s}_{h^*}(V_{n_{\bar{k}}}(\mathcal{H})) < \bar{k}$. Moreover, we know there exists a distribution-dependent constant factor $k := k(P) < \infty$ such that $\bar{k}(S) \le k(P)$ with probability at least $1/2$. By using a similar argument in the proof of Lemma 15, we have

$$\mathbb{E}\left[er_P(\hat{h}_n)\right] \lesssim \frac{k}{n} + 2^{-\lfloor n/k \rfloor}, \forall n \in \mathbb{N},$$

which proves a target-specified linear upper bound. $\qquad\square$

**Lemma 16** (**Lemma 5 restated**). *If $\mathcal{H}$ has an infinite star-eluder sequence centered at $h^*$, then $h^*$ is not universally learnable by ERM at rate faster than $\log(n)/n$.*

*Proof of Lemma 16.* Suppose that $S := \{(x_1, y_1), (x_2, y_2), \ldots\}$ is an infinite star-eluder sequence in $\mathcal{H}$ centered at $h^*$, that is $h^*(x_i) = y_i$ for all $i \in \mathbb{N}$. For notation simplicity, let $\mathcal{X}_1 := \{x_1\}, \mathcal{X}_2 := \{x_2, x_3\}, \mathcal{X}_3 := \{x_4, x_5, x_6\}, \ldots, \mathcal{X}_k := \{x_{n_k+1}, \ldots, x_{n_k+k}\}, \ldots$, with $n_k := \binom{k}{2}$. We consider a strictly increasing sequence $\{k_t\}_{t \in \mathbb{N}}$ that will be specified later, and only put non-zero probability

masses on these disjoint sets $\mathcal{X}_{k_t}$ with $t \in \mathbb{N}$. Then, let $\mathcal{X} := \bigcup_{t \in \mathbb{N}} \mathcal{X}_{k_t}$ be a union of disjoint finite sets with $|\mathcal{X}_{k_t}| = k_t$, and consider the following marginal distribution $P_{\mathcal{X}}$ on $\mathcal{X}$:

$$P_{\mathcal{X}}\{x \in \mathcal{X}_{k_t}\} = 2^{-t} \text{ and } P_{\mathcal{X}}(x) = 2^{-t}/k_t, \ \forall x \in \mathcal{X}_{k_t}.$$

It immediately implies the joint distribution $P := P(P_{\mathcal{X}}, h^*)$ that is realizable with respect to $\mathcal{H}$:

$$P\{(x, h^*(x))\} = 2^{-t}/k_t, \ P\{(x, 1 - h^*(x))\} = 0, \ \forall x \in \mathcal{X}_{k_t}, \ \forall k \in \mathbb{N}.$$

Now for any $n \in \mathbb{N}$, we let $S_n := \{(x_i, y_i)\}_{i=1}^n \sim P^n$ and consider the event $\mathcal{E} := \mathcal{E}_1 \cap \mathcal{E}_2$, where

$$\mathcal{E}_1 := \{S_n \text{ does not contain a copy of any point in } \mathcal{X}_{k_{>t}}\},$$
$$\mathcal{E}_2 := \{S_n \text{ does not contain a copy of at least one point in } \mathcal{X}_{k_t}\}.$$

If $\mathcal{E}$ happens, the worst-case ERM can output some $\hat{h}_n \in V_{S_n}(\mathcal{H})$ such that $\mathrm{er}_P(\hat{h}_n) \geq 2^{-t}/k_t$. This is because: $V_{n_{k_t}}(\mathcal{H}) \subseteq V_{S_{n,k_{<t}}}(\mathcal{H})$, where $S_{n,k_{<t}}$ contains the samples of $S_n$ that falling into $\mathcal{X}_{k_1} \cup \cdots \cup \mathcal{X}_{k_{t-1}}$, and then $\mathcal{X}_{k_t} = \{x_{n_{k_t}+1}, \ldots, x_{n_{k_t}+k_t}\}$ is a star set of $V_{S_{n,k_{<t}}}(\mathcal{H})$ witnessed by a set of functions, denoted by $\{h_{n_{k_t}+1}, \ldots, h_{n_{k_t}+k_t}\}$. In other words, $V_{S_{n,k_{<t}}}(\mathcal{H})$ contains a size-$k_t$ "singletons" with point-wise probability mass $2^{-t}/k_t$. However, the remaining samples $S_n \cap \mathcal{X}_{k_t}$ does not contain a copy of every point in $\mathcal{X}_{k_t}$, which results in an error rate $\mathrm{er}_P(\hat{h}_n) \geq 2^{-t}/k_t$, with $\hat{h}_n := h_{n_{k_t}+j}$ for some $1 \leq j \leq k_t$.

Hence, it remains to characterize the probability of $\mathcal{E}$. To this end, we refer to the so-called *Coupon Collector's Problem*, and define a random variable

$$\hat{n}_{k_t} := \min\{n \in \mathbb{N} : \mathcal{X}_{k_t} \subseteq S_n\}.$$

Note that $\hat{n}_{k_t}$ can be represented as a sum $\sum_{j=1}^{k_t} G_j$ of independent geometric random variables $G_j \sim \text{Geometric}(\frac{k_t+1-j}{k_t}2^{-t})$ for $1 \leq j \leq k_t$, with

$$\begin{cases} \mathbb{E}[\hat{n}_{k_t}] = \sum_{j=1}^{k_t} \mathbb{E}[G_j] = \sum_{j=1}^{k_t} \frac{k_t \cdot 2^t}{k_t+1-j} = k_t \cdot 2^t \left(\sum_{j=1}^{k_t} \frac{1}{k_t+1-j}\right) = k_t \cdot 2^t \cdot H_{k_t}, \\ \mathrm{Var}[\hat{n}_{k_t}] = \sum_{j=1}^{k_t} \mathrm{Var}[G_j] < \sum_{j=1}^{k_t} \left(\frac{k_t+1-j}{k_t}2^{-t}\right)^{-2} < \frac{\pi^2 \cdot k_t^2 \cdot 2^{2t}}{6} \end{cases},$$

where $H_m$ is $m^{th}$ harmonic number satisfying $H_m \gtrsim \log(m)$. Then the standard Chebyshev's inequality implies that $\mathbb{P}(|\hat{n}_{k_t} - \mathbb{E}[\hat{n}_{k_t}]| > z) \leq \mathrm{Var}[\hat{n}_{k_t}] \cdot z^{-2}$. By choosing $z = \sqrt{2\mathrm{Var}[\hat{n}_{k_t}]}$, we have with probability at least $1/2$,

$$\hat{n}_{k_t} > \mathbb{E}[\hat{n}_{k_t}] - \sqrt{2\mathrm{Var}[\hat{n}_{k_t}]} \geq k_t \cdot 2^t \cdot \left(\log k_t - \frac{\pi}{\sqrt{3}}\right).$$

In particular, when $k_t \geq 38$, it holds that $\log k_t \geq 2\pi/\sqrt{3}$, and thus $\hat{n}_{k_t} > 2^{t-1} k_t \log k_t$ with probability at least $1/2$. Altogether, we have for any $n \leq 2^{t-1} k_t \log k_t$,

$$\mathbb{P}(\mathcal{E}_2) \geq \mathbb{P}(n < \hat{n}_{k_t}) \geq \mathbb{P}(n \leq 2^{t-1} k_t \log k_t, \ 2^{t-1} k_t \log k_t < \hat{n}_{k_t}) \geq 1/2.$$

Moreover, to characterize the probability of $\mathcal{E}_1$, note that for any $x \sim P_{\mathcal{X}}$, $\mathbb{P}(x \in \mathcal{X}_{k_{>t}}) = 2^{-k_t}$, which implies immediately that for any $n \in \mathbb{N}$, $\mathbb{P}(\mathcal{E}_1) = (1 - 2^{-k_t})^n$. Now for any integer $k_t \geq 38$, we let $n = 2^{t-1} k_t \log k_t$, and have (for infinitely many $n$) that

$$\mathbb{P}\left(\mathrm{er}_P(\hat{h}_n) \geq \frac{2^{-t}}{k_t}\right) \geq \mathbb{P}(\mathcal{E}) \geq \mathbb{P}(\mathcal{E}_1)\mathbb{P}(\mathcal{E}_2) \geq \frac{1}{2}(1 - 2^{-k_t})^n,$$

which implies further

$$\mathbb{E}\left[\mathrm{er}_P(\hat{h}_n)\right] \geq \frac{2^{-t}}{k_t}\mathbb{P}\left(\mathrm{er}_P(\hat{h}_n) \geq \frac{2^{-t}}{k_t}\right) \geq \frac{1}{k_t 2^{t+1}}(1 - 2^{-k_t})^n =: \eta_{n,t}.$$

Finally, by choosing $k_t = \Omega(2^t)$, we can guarantee that $n \geq \eta_{n,t}^{-1} \log \eta_{n,t}^{-1}$. Applying Lemma 29, we have $\mathbb{E}[\mathrm{er}_P(\hat{h}_n)] \geq \eta_{n,t} \geq \log(n)/n$, for infinitely many $n$. $\qquad\square$

**Lemma 17 (Lemma 6 restated).** *If $\mathcal{H}$ does not have an infinite VC-eluder sequence centered at $h^*$, then $h^*$ is universally learnable by ERM at $\log(n)/n$ rate.*

*Proof of Lemma 17.* We first prove that any class $\mathcal{H}$ is universally learnable by ERM at $\log(n)/n$ rate if $\mathrm{VC}(\mathcal{H}) < \infty$. For any realizable distribution $P$ with respect to $\mathcal{H}$, we let $S_{2n} := \{(x_i, y_i)\}_{i=1}^{2n} \sim P^{2n}$, and denote $S_n := \{(x_i, y_i)\}_{i=1}^{n}$ and $T_n := \{(x_i, y_i)\}_{i=n+1}^{2n}$, namely, the "ghost samples". Given $\epsilon \in (0,1)$, Lemma 31 states that for any $n \geq 8/\epsilon$,

$$\mathbb{P}\left(\exists h \in \mathcal{H} : \hat{\mathrm{er}}_{S_n}(h) = 0 \text{ and } \mathrm{er}_P(h) > \epsilon\right) \leq 2\mathbb{P}\left(\exists h \in \mathcal{H} : \hat{\mathrm{er}}_{S_n}(h) = 0 \text{ and } \hat{\mathrm{er}}_{T_n}(h) > \epsilon/2\right).$$

Moreover, Lemma 32 states that for any $n \geq \mathrm{VC}(\mathcal{H})/2$,

$$\mathbb{P}\left(\exists h \in \mathcal{H} : \hat{\mathrm{er}}_{S_n}(h) = 0 \text{ and } \hat{\mathrm{er}}_{T_n}(h) > \epsilon/2\right) \leq \left(\frac{2en}{\mathrm{VC}(\mathcal{H})}\right)^{\mathrm{VC}(\mathcal{H})} 2^{-n\epsilon/2}.$$

Altogether, we have

$$\mathbb{P}\left(\mathrm{er}_P(\hat{h}_n) > \epsilon\right) \leq \mathbb{P}\left(\exists h \in \mathcal{H} : \hat{\mathrm{er}}_{S_n}(h) = 0 \text{ and } \mathrm{er}_P(h) > \epsilon\right) \leq 2\left(\frac{2en}{\mathrm{VC}(\mathcal{H})}\right)^{\mathrm{VC}(\mathcal{H})} 2^{-\frac{n\epsilon}{2}}, \quad (5)$$

for any $n \geq \max\{8/\epsilon, \mathrm{VC}(\mathcal{H})/2\}$. Finally, the upper bound on the expectation can be derived via the follow analysis (which will be used several times later): let

$$\epsilon_n := \frac{2}{n}\left(\mathrm{VC}(\mathcal{H}) \log\left(\frac{2en}{\mathrm{VC}(\mathcal{H})}\right) + 1\right),$$

and then by letting the RHS of (5) $=: \delta$, we have

$$\epsilon = \frac{2}{n}\left(\mathrm{VC}(\mathcal{H}) \log\left(\frac{2en}{\mathrm{VC}(\mathcal{H})}\right) + \log\left(\frac{2}{\delta}\right)\right) > \epsilon_n.$$

When $\epsilon \leq \epsilon_n$, we of course still have $\mathbb{P}(\mathrm{er}_P(\hat{h}_n) > \epsilon) \leq 1$. It follows that for all $n \geq \mathrm{VC}(\mathcal{H})/2$,

$$\begin{aligned}
\mathbb{E}\left[\mathrm{er}_P(\hat{h}_n)\right] &= \int_0^1 \mathbb{P}\left(\mathrm{er}_P(\hat{h}_n) > \epsilon\right) d\epsilon \\
&= \int_{\frac{8}{n}}^1 \mathbb{P}\left(\mathrm{er}_P(\hat{h}_n) > \epsilon\right) d\epsilon + \int_0^{\frac{8}{n}} \mathbb{P}\left(\mathrm{er}_P(\hat{h}_n) > \epsilon\right) d\epsilon \\
&= \int_{\frac{8}{n}}^{\epsilon_n} \mathbb{P}\left(\mathrm{er}_P(\hat{h}_n) > \epsilon\right) d\epsilon + \int_{\epsilon_n}^1 \mathbb{P}\left(\mathrm{er}_P(\hat{h}_n) > \epsilon\right) d\epsilon + \int_0^{\frac{8}{n}} \mathbb{P}\left(\mathrm{er}_P(\hat{h}_n) > \epsilon\right) d\epsilon \\
&\overset{(5)}{\leq} \epsilon_n + \int_{\epsilon_n}^\infty 2\left(\frac{2en}{\mathrm{VC}(\mathcal{H})}\right)^{\mathrm{VC}(\mathcal{H})} 2^{-n\epsilon/2} d\epsilon \\
&= \frac{2\mathrm{VC}(\mathcal{H})}{n} \log\left(\frac{2en}{\mathrm{VC}(\mathcal{H})}\right) + \frac{2 + 2\ln(2)}{n} \lesssim \frac{\mathrm{VC}(\mathcal{H})}{n} \log\left(\frac{n}{\mathrm{VC}(\mathcal{H})}\right).
\end{aligned}$$

For $n \leq \mathrm{VC}(\mathcal{H})/2$, the result is trivial.

Now to prove a target-specified upper bound, let $P$ be any realizable distribution centered at the target concept $h^*$ with an associated marginal distribution denoted by $P_\mathcal{X}$. Suppose that $\mathcal{H}$ does not have an infinite VC-eluder sequence centered at $h^*$, then there is a largest target-dependent integer $d := d(h^*) < \infty$ such that there exists an infinite $d$-VC-eluder sequence centered at $h^*$, but no infinite $(d+1)$-VC-eluder sequence centered at $h^*$ (see Definition 9). We know that there exists a positive distribution-dependent integer $k := k(P) < \infty$ such that $\mathbb{P}(\mathrm{VC}(V_k(\mathcal{H})) \leq d) \geq 1/2$.

We let $S_n := \{(x_i, y_i)\}_{i=1}^n \sim P^n$ be a dataset, and consider the event $\mathcal{E}_n := \{\mathrm{VC}(V_{\lfloor n/2 \rfloor}(\mathcal{H})) \leq d\}$. The probability of this event can be characterized as follow: for any $n \in \mathbb{N}$, assume first $n \geq k$, we then split the dataset $S_n \sim P^n$ into $\lfloor n/k \rfloor$ parts with each one sized at least $k$, denoted by $S_{n,1}, \ldots, S_{n,\lfloor n/k \rfloor}$. For every $1 \leq i \leq \lfloor n/k \rfloor$, we know that the corresponding induced version space has VC dimension $\mathrm{VC}(V_{S_{n,i}}(\mathcal{H})) \leq d$ with probability at least $1/2$. Note that $V_n(\mathcal{H}) = \bigcap_{1 \leq i \leq \lfloor n/k \rfloor} V_{S_{n,i}}(\mathcal{H})$ satisfies $\mathrm{VC}(V_n(\mathcal{H})) \leq \mathrm{VC}(V_{S_{n,i}}(\mathcal{H}))$, for every $1 \leq i \leq \lfloor n/k \rfloor$. Therefore, we have $\mathbb{P}(\mathrm{VC}(V_n(\mathcal{H})) > d) \leq \mathbb{P}(\forall 1 \leq i \leq \lfloor n/k \rfloor : \mathrm{VC}(V_{S_{n,i}}(\mathcal{H})) > d) \leq 2^{-\lfloor n/k \rfloor}$. Note that this bound also holds when $n < k$ since a probability is always at most 1. Altogether, we obtain $\mathbb{P}(\neg\mathcal{E}_n) = \mathbb{P}(\mathrm{VC}(V_{\lfloor n/2 \rfloor}(\mathcal{H})) > d) \leq 2^{-\lfloor n/2k \rfloor}$. Conditioning on this event, the previous analysis

of the uniform rate $\log(n)/n$ can be applied since the version space has a bounded VC dimension $d$. Finally, we have that for all $n \in \mathbb{N}$,

$$
\begin{aligned}
\mathbb{E}\left[\mathrm{er}_P(\hat{h}_n)\right] &= \int_0^\infty \mathbb{P}\left(\mathrm{er}_P(\hat{h}_n) > \epsilon\right) d\epsilon \\
&\leq \int_0^\infty \left(\mathbb{P}\left(\mathrm{er}_P(\hat{h}_n) > \epsilon \Big| \mathcal{E}_n\right) + \mathbb{P}\left(\neg \mathcal{E}_n\right)\right) d\epsilon \lesssim \frac{d}{n} \log\left(\frac{n}{d}\right) + 2^{-\lfloor n/k \rfloor},
\end{aligned}
$$

where both $d := d(h^*)$ and $k := k(P)$ are distribution-dependent constants. In conclusion, $h^*$ is universally learnable by ERM at $\log(n)/n$ rate. $\qquad\square$

**Lemma 18 (Lemma 7 restated).** *If $\mathcal{H}$ has an infinite VC-eluder sequence centered at $h^*$, then $h^*$ requires at least arbitrarily slow rates to be universally learned by ERM.*

*Proof of Lemma 18.* Let $S := \{(x_1, y_1), (x_2, y_2), \ldots\}$ be an infinite VC-eluder sequence in $\mathcal{H}$ centered at $h^*$, that is, $h^*(x_i) = y_i$ for all $i \in \mathbb{N}$. We inherit the notations used in Lemma 5 by letting $\mathcal{X}_k := \{x_{n_k+1}, \ldots, x_{n_k+k}\}$ with $n_k := \binom{k}{2}$, for all $k \in \mathbb{N}$. Let $\mathcal{X} := \bigcup_{k \in \mathbb{N}} \mathcal{X}_k$ be a union of disjoint finite sets with $|\mathcal{X}_k| = k$, and consider the following marginal distribution $P_{\mathcal{X}}$ on $\mathcal{X}$:

$$
P_{\mathcal{X}}\{x \in \mathcal{X}_k\} = p_k \text{ and } P_{\mathcal{X}}(x) = p_k/k, \ \forall x \in \mathcal{X}_k,
$$

where $\{p_k\}_{k \in \mathbb{N}}$ is a sequence of probabilities satisfying $\sum_{k \geq 1} p_k \leq 1$ that will be specified later. It implies immediately the following realizable (joint) distribution $P := P(P_{\mathcal{X}}, h^*)$:

$$
P\{(x, h^*(x))\} = p_k/k, \ P\{(x, 1 - h^*(x))\} = 0, \ \forall x \in \mathcal{X}_k, \ \forall k \in \mathbb{N}.
$$

Our remaining target is to show that for any rate function $R(n) \to 0$, $\mathcal{H}$ cannot be universally learned by the worst-case ERM at rate faster than $R(n)$ under the distribution $P$. To this end, we let $S_n := \{(x_i, y_i)\}_{i=1}^n \sim P^n$. For any $t \in \mathbb{N}$ and any $j \in [k_t]$, we consider the following event

$$
\mathcal{E}_{n,k,t,j} := \left\{S_n \text{ does not contain a copy of any point in } \mathcal{X}_{k>t} \cup \{x_{n_{k_t}+j}\}\right\},
$$

where $\{k_t\}_{t \in \mathbb{N}}$ is an increasing sequence of integers that will be specified later. If $\mathcal{E}_{n,k,t,j}$ happens, then the worst-case ERM algorithm can output some $\hat{h}_n \in V_{n_{k_t}}(\mathcal{H})$ such that $\mathrm{er}_P(\hat{h}_n) \geq p_{k_t}/k_t$, that is the classifier that predicts incorrectly on the "missing" point in $\mathcal{X}_{k_t}$. Moreover, to characterize the probability of event $\mathcal{E}_{n,k,t,j}$, we have $\mathbb{P}(\mathcal{E}_{n,k,t,j}) = (1 - \sum_{k>k_t} p_k - p_{k_t}/k_t)^n$. Therefore, we make a construction by applying Lemma 30, and finally get for all $t \in \mathbb{N}$,

$$
\mathbb{E}\left[\mathrm{er}_P\left(\hat{h}_{n_t}\right)\right] \geq \sum_{j \in [k_t]} p_{k_t} \cdot \mathbb{P}\left(\mathcal{E}_{n_t,k,t,j}\right) \geq p_{k_t}\left(1 - \sum_{k>k_t} p_k - \frac{p_{k_t}}{k_t}\right)^{n_t} \geq p_{k_t}\left(1 - \frac{2}{n_t}\right)^{n_t} \gtrsim R(n_t).
$$

$\qquad\square$

### D.2 Omitted Proofs in Section 4

**Lemma 19 (Lemma 8 restated).** *$\mathcal{H}$ has an infinite eluder sequence if and only if $|\mathcal{H}| = \infty$.*

*Proof of Lemma 19.* The necessity is straightforward. To show the sufficiency, we construct such an infinite eluder sequence via the following procedure: pick some $x_1 \in \mathcal{X}$ such that both $V_{(x_1,0)}(\mathcal{H}) := \{h \in \mathcal{H} : h(x_1) = 0\}$ and $V_{(x_1,1)}(\mathcal{H}) := \{h \in \mathcal{H} : h(x_1) = 1\}$ are non-empty. Such a point $x_1$ must exist since otherwise we will have $|\mathcal{H}| = 1$. Furthermore, we know that at least one of them is infinite, since otherwise we will have $|\mathcal{H}| < \infty$. We assume, without loss of generality, that $|V_{(x_1,0)}(\mathcal{H})| = \infty$. Then we pick some $x_2 \in \mathcal{X}$ such that both $V_{\{(x_1,0),(x_2,0)\}}(\mathcal{H}) := \{h \in \mathcal{H} : h(x_1) = 0, h(x_2) = 0\}$ and $V_{\{(x_1,0),(x_2,1)\}}(\mathcal{H}) := \{h \in \mathcal{H} : h(x_1) = 0, h(x_2) = 1\}$ are non-empty. Note that such an $x_2$ exists, because otherwise we will have $|V_{(x_1,0)}(\mathcal{H})| < \infty$. Again, at least one of them is infinite for the same reason, and then we choose $x_3$ from that infinite one. Following a similar procedure, we can get an infinite sequence $\{(x_1, y_1), (x_2, y_2), \ldots\}$, where $\{x_1, x_2, \ldots\}$ are chosen to keep the separates of version space non-empty, and $\{y_1, y_2, \ldots\}$ are chosen to keep the version space infinite. Now note that for any $i \in \mathbb{N}$, we can find some $h_i \in V_{S_i}(\mathcal{H}) \neq \emptyset$, where $S_i = \{(x_1, 1 - y_1), (x_2, 1 - y_2), \ldots, (x_{i-1}, 1 - y_{i-1}), (x_i, y_i)\}$. According to Definition 5, we know that $\{(x_1, y_1), (x_2, y_2), \ldots\}$ is an infinite eluder sequence consistent with $\mathcal{H}$. $\qquad\square$

**Lemma 20 (Lemma 9 restated).** *$\mathcal{H}$ has an infinite VC-eluder sequence if and only if $VC(\mathcal{H}) = \infty$.*

*Proof of Lemma 20.* According to Definition 7, the necessity is straightforward, i.e. we must have $VC(\mathcal{H}) = \infty$ if $\mathcal{H}$ has an infinite VC-eluder sequence. It remains to prove the sufficiency, that is, $VC(\mathcal{H}) = \infty$ yields the existence of an infinite VC-eluder sequence.

We construct an infinite VC-eluder sequence via the following procedure: Let $x_1 \in \mathcal{X}$ be any point, then at least one of $V_{\{(x_1,0)\}}(\mathcal{H})$ and $V_{\{(x_1,1)\}}(\mathcal{H})$ has an infinite VC dimension, which is because otherwise $\mathcal{H} = V_{\{(x_1,y_1)\}}(\mathcal{H}) \cup V_{\{(x_1,1-y_1)\}}(\mathcal{H})$ will have a finite VC dimension based on Lemma 28. Let $y_1 \in \{0,1\}$ such that $V_{\{(x_1,y_1)\}}(\mathcal{H})$ has an infinite VC dimension, and let $\{x_2, x_3\}$ be a shattered set of $V_{\{(x_1,y_1)\}}(\mathcal{H})$. Similarly, we know least one of the following four subclasses $V_{\{(x_1,y_1),(x_2,0),(x_3,0)\}}(\mathcal{H})$, $V_{\{(x_1,y_1),(x_2,0),(x_3,1)\}}(\mathcal{H})$, $V_{\{(x_1,y_1),(x_2,1),(x_3,0)\}}(\mathcal{H})$, $V_{\{(x_1,y_1),(x_2,1),(x_3,1)\}}(\mathcal{H})$ has an infinite VC dimension since otherwise $VC(V_{\{(x_1,y_1)\}}(\mathcal{H})) < \infty$ will lead to a contradiction. We then pick labels $y_2, y_3 \in \{0,1\}$ such that $V_{\{(x_1,y_1),(x_2,y_2),(x_3,y_3)\}}(\mathcal{H})$ has an infinite VC dimension. For notation simplicity, let $S := \{(x_1,y_1),(x_2,y_2),\ldots\}$ and let $n_k := \binom{k}{2}$. Inductively, for any $k \in \mathbb{N}$, if $V_{S_{1+2+\cdots+(k-1)}}(\mathcal{H}) = V_{S_{n_k}}(\mathcal{H})$ has an infinite VC dimension, let $\{x_{n_k+1}, \ldots, x_{n_k+k}\}$ be a shattered set of $V_{S_{n_k}}(\mathcal{H})$. Lemma 28 yields the existence of a set of labels $\{y_{n_k+1}, \ldots, y_{n_k+k}\} \in \{0,1\}^k$ such that $V_{S_{n_{k+1}}}(\mathcal{H})$ has an infinite VC dimension. Otherwise,

$$\text{VC}\left(V_{S_{n_k}}(\mathcal{H})\right) = \text{VC}\left(\bigcup_{(y_{n_k+1},\ldots,y_{n_k+k}) \in \{0,1\}^k} V_{S_{n_{k+1}}}(\mathcal{H})\right) \leq 2k + 4 \max \text{VC}\left(V_{S_{n_{k+1}}}(\mathcal{H})\right) < \infty,$$

which leads us to a contradiction! By such a construction, the returned infinite sequence $S := \{(x_1,y_1),(x_2,y_2),\ldots\}$ is an infinite VC-eluder sequence consistent with $\mathcal{H}$. $\square$

### D.3 Omitted Proofs in Appendix C

**Lemma 21 (Lemma 10 restated).** *For every concept class $\mathcal{H}$ with $|\mathcal{H}| \geq 3$, if $VCE(\mathcal{H}) < \infty$, then the following hold:*

$$\mathbb{E}\left[er_P(\hat{h}_n)\right] \geq \frac{VCE(\mathcal{H})}{18n}, \text{ for infinitely many } n \in \mathbb{N},$$

$$\mathbb{E}\left[er_P(\hat{h}_n)\right] \leq \frac{28 VCE(\mathcal{H}) \log n}{n} + 2^{-\lfloor n/2\kappa \rfloor}, \ \forall n \in \mathbb{N},$$

*where $\kappa = \kappa(P)$ is a distribution-dependent constant.*

*Proof of Lemma 21.* Let $\mathcal{H}$ be a concept class with $VCE(\mathcal{H}) = d < \infty$. Note that when $VCE(\mathcal{H}) = 0$, the results hold trivially, and hence we consider only $d \geq 1$ in the remaining part of the proof.

To show the upper bound, let $P$ be a realizable distribution with respect to $\mathcal{H}$, and for any $n \in \mathbb{N}$, let $S_n := \{(x_1,y_1),\ldots,(x_n,y_n)\} \sim P^n$ be a dataset. Since $VCE(\mathcal{H}) = d$, for any infinite sequence $S := \{(x_1,y_1),(x_2,y_2),\ldots\} \sim P^{\mathbb{N}}$, there exists a (smallest) non-negative integer $k = k(S) < \infty$ such that $VC(V_k(\mathcal{H})) \leq d$. For any ERM algorithm $\mathcal{A}$, let $\hat{h}_n := \mathcal{A}_{\mathcal{H}}(S_n) \in V_n(\mathcal{H})$, which can also be written as $\hat{h}_n := \hat{h}_{n,k} := \mathcal{A}_{V_k(\mathcal{H})}(S_{k+1:n})$, for every $k \in [n]$. Recall that for any $\epsilon \in (0,1)$, using the same argument as in the proof of Lemma 6, we can get

$$\mathbb{P}\left(er_P(\hat{h}_{n,k}) > \epsilon\right) \leq 2\left(\frac{2e(n-k)}{d}\right)^d 2^{-(n-k)\epsilon/2}, \ \forall n \geq k + \max\{8/\epsilon, d/2\}. \tag{6}$$

Now for any $n \in \mathbb{N}$, we consider the event $\mathcal{E}_n := \{VC(V_{\lfloor n/2 \rfloor}(\mathcal{H})) \leq d\}$. Applying the inequality $\mathbb{P}(A) \leq \mathbb{P}(A|B) + \mathbb{P}(\neg B)$, we have

$$\mathbb{P}\left(er_P(\hat{h}_n) > \epsilon\right) \leq \mathbb{P}\left(er_P(\hat{h}_{n,\lfloor n/2 \rfloor}) > \epsilon \Big| \mathcal{E}_n\right) + \mathbb{P}(\neg \mathcal{E}_n). \tag{7}$$

Let the RHS of (6) $=: \delta \in (0,1)$, then for the first probability in (7), we have

$$\mathbb{P}\left(er_P(\hat{h}_n) > \epsilon \Big| \mathcal{E}_n\right) = \mathbb{P}\left(er_P(\hat{h}_{n,\lfloor n/2 \rfloor}) > \epsilon \Big| VC(V_{\lfloor n/2 \rfloor}(\mathcal{H})) \leq d\right) \leq \delta,$$

for all $n \geq \max\{16/\epsilon, d\} \geq \lfloor n/2 \rfloor + \max\{8/\epsilon, d/2\}$, and also

$$\epsilon = \frac{2}{n - \lfloor \frac{n}{2} \rfloor}\left(d\log\left(\frac{2e(n - \lfloor \frac{n}{2} \rfloor)}{d}\right) + \log\left(\frac{2}{\delta}\right)\right) \gtrsim \frac{4}{n}\left(d\log\left(\frac{ne}{d}\right) + 1\right) =: \epsilon_n. \qquad (8)$$

To characterize the second probability in (7), we define $\kappa = \kappa(P)$, a distribution-dependent quantity, to be the smallest integer such that $k(S) \leq \kappa$ with probability at least $1/2$, where the randomness is from the data sequence $S$. Note that such an integer $\kappa$ exists since otherwise there will exist at least an infinite $(d+1)$-VC-eluder sequence. We then prove:

**Claim 2.** *For any* $n \in \mathbb{N}$, $\mathbb{P}(\neg\mathcal{E}_n) \leq 2^{-\lfloor n/2\kappa \rfloor}$.

*Proof of Claim 2.* For any $n \in \mathbb{N}$, assume first that $n \geq \kappa$, we then split the dataset $S_n \sim P^n$ into $\lfloor n/\kappa \rfloor$ parts with each one sized $\kappa$, denoted by $S_{n,1}, \ldots, S_{n,\lfloor n/\kappa \rfloor}$. According to the definition, for every $1 \leq i \leq \lfloor n/\kappa \rfloor$, we know that the induced version space has VC dimension $\mathrm{VC}(V_{S_{n,i}}(\mathcal{H})) \leq d$ with probability at least $1/2$. Note that $V_n(\mathcal{H}) = \bigcap_{1 \leq i \leq \lfloor n/\kappa \rfloor} V_{S_{n,i}}(\mathcal{H})$ satisfies $\mathrm{VC}(V_n(\mathcal{H})) \leq \mathrm{VC}(V_{S_{n,i}}(\mathcal{H}))$, for all $1 \leq i \leq \lfloor n/\kappa \rfloor$. Therefore, we have

$$\mathbb{P}\left(\mathrm{VC}(V_n(\mathcal{H})) > d\right) \leq \mathbb{P}\left(\forall 1 \leq i \leq \lfloor n/\kappa \rfloor : \mathrm{VC}(V_{S_{n,i}}(\mathcal{H})) > d\right) \leq 2^{-\lfloor n/\kappa \rfloor},$$

which also holds when $n < \kappa$. Finally, $\mathbb{P}(\neg\mathcal{E}_n) = \mathbb{P}(\mathrm{VC}(V_{\lfloor n/2 \rfloor}(\mathcal{H})) > d) \leq 2^{-\lfloor n/2\kappa \rfloor}$. $\qquad\square$

Putting together, we have that for all $n \geq d$,

$$\mathbb{E}\left[\mathrm{er}_P(\hat{h}_n)\right] = \int_{16/n}^1 \mathbb{P}\left(\mathrm{er}_P(\hat{h}_n) > \epsilon\right)d\epsilon + \int_0^{16/n} \mathbb{P}\left(\mathrm{er}_P(\hat{h}_n) > \epsilon\right)d\epsilon$$

$$\overset{(7)}{\leq} \int_{16/n}^1 \mathbb{P}\left(\mathrm{er}_P(\hat{h}_{n,\lfloor n/2 \rfloor}) > \epsilon\Big|\mathcal{E}_n\right)d\epsilon + \int_0^{16/n} \mathbb{P}\left(\mathrm{er}_P(\hat{h}_{n,\lfloor n/2 \rfloor}) > \epsilon\Big|\mathcal{E}_n\right)d\epsilon + \int_0^1 \mathbb{P}\left(\neg\mathcal{E}_n\right)d\epsilon$$

$$\overset{(8)}{\leq} \epsilon_n + \int_{\epsilon_n}^1 \mathbb{P}\left(\mathrm{er}_P(\hat{h}_{n,\lfloor n/2 \rfloor}) > \epsilon\Big|\mathcal{E}_n\right)d\epsilon + \int_0^1 \mathbb{P}\left(\neg\mathcal{E}_n\right)d\epsilon$$

$$\overset{(6)}{\leq} \epsilon_n + 3\int_{\epsilon_n}^\infty 2\left(\frac{ne}{d}\right)^d 2^{-n\epsilon/4}d\epsilon + \int_0^1 \mathbb{P}\left(\neg\mathcal{E}_n\right)d\epsilon$$

$$\leq \frac{4d}{n}\log\left(\frac{ne}{d}\right) + \frac{4 + 12\ln(2)}{n} + \int_0^1 \mathbb{P}\left(\neg\mathcal{E}_n\right)d\epsilon$$

$$\overset{\text{Claim 2}}{\leq} \frac{4d}{n}\log\left(\frac{ne}{d}\right) + \frac{4 + 12\ln(2)}{n} + 2^{-\lfloor n/2\kappa \rfloor} \leq \frac{28d\log n}{n} + 2^{-\lfloor n/2\kappa \rfloor}.$$

When $n \leq d$, the upper bound is trivial.

To show the lower bound, let $S := \{(x_1, y_1), (x_2, y_2), \ldots\}$ be any infinite $d$-VC-eluder sequence that is consistent with $\mathcal{H}$. We denote $\mathcal{X}_k := \{x_{kd-d+1}, \ldots, x_{kd}\}$ for all integers $k \geq 1$, and consider the following realizable distribution $P$:

$$P\{(x_{kd-d+j}, y_{kd-d+j})\} = p_k/d, \; P\{(x_{kd-d+j}, 1 - y_{kd-d+j})\} = 0, \; \forall 1 \leq j \leq d, \; \forall k \geq 1,$$

where $\{p_k\}_{k \geq 1}$ is a sequence of probabilities satisfying $\sum_{k \geq 1} p_k \leq 1$, which will be specified later.

We use a similar argument in the proof of Lemma 7, but instead of considering an arbitrarily slow rate function $R(n) \to 0$, we consider here $R(n) := d/n$. Specifically, let $S_n := \{(x_i, y_i)\}_{i=1}^n \sim P^n$ be a dataset. Note that for any $k \in \mathbb{N}$ and any $j \in [d]$, if the dataset $S_n$ does not contain any copy of the points in $\mathcal{X}_{>k} \cup \{x_{kd-d+j}\} := \bigcup_{t>k} \mathcal{X}_t \cup \{x_{kd-d+j}\}$, the worst-case ERM can have an error rate $\mathrm{er}_P(\hat{h}_n) \geq p_k/d$. The probability of such event is $\mathbb{P}(\sum_{i=1}^n \mathbb{1}\{X_i \in \mathcal{X}_{>k} \cup \{x_{kd-d+j}\}\} = 0) = \prod_{i=1}^n \mathbb{P}(X_i \notin \mathcal{X}_{>k} \cup \{x_{kd-d+j}\}) = (1 - \sum_{t>k} p_t - p_k/d)^n$.

Based on Lemma 30, for the rate function $R(n) := d/n$, there exist probabilities $\{p_k\}_{k \geq 1}$ satisfying $\sum_{k \geq 1} p_k = 1$, two increasing sequences of integers $\{k_t\}_{t \geq 1}$ and $\{n_t\}_{t \geq 1}$, and a constant $1/2 \leq C \leq 1$ such that $\sum_{k > k_t} p_k \leq 1/n_t$ and $p_{k_t} = C \cdot d/n_t$. Therefore, it follows that for all integers $t \geq 1$ (and thus for infinitely many $n \in \mathbb{N}$),

$$\mathbb{E}\left[\mathrm{er}_P\left(\hat{h}_{n_t}\right)\right] \geq C \cdot \frac{d}{n_t}\sum_{t \geq 1}\left(1 - \frac{C+1}{n_t}\right)^{n_t} \geq \frac{d}{2n_t}\sum_{t \geq 1}\left(1 - \frac{2}{n_t}\right)^{n_t} \geq \frac{d}{18n_t}.$$

$\square$

**Lemma 22 (Lemma 11 restated).** *For every concept class $\mathcal{H}$ with $|\mathcal{H}| \geq 3$, if $1 \leq SE(\mathcal{H}) < \infty$, then the following hold:*

$$\mathbb{E}\left[er_P(\hat{h}_n)\right] \geq \frac{\log\left(SE(\mathcal{H})\right)}{12n}, \text{ for infinitely many } n \in \mathbb{N},$$

$$\mathbb{E}\left[er_P(\hat{h}_n)\right] \leq \frac{160 VCE(\mathcal{H})}{n} \log\left(\frac{SE(\mathcal{H})}{VCE(\mathcal{H})}\right) + 2^{-\lfloor n/2\hat{\kappa}\rfloor}, \ \forall n \in \mathbb{N},$$

*where $\hat{\kappa} = \hat{\kappa}(P)$ is a distribution-dependent constant.*

*Proof of Lemma 22.* Let $\mathcal{H}$ be a concept class with $SE(\mathcal{H}) = s < \infty$ and $VCE(\mathcal{H}) = d < \infty$.

To prove the upper bound, let $P$ be a realizable distribution with respect to $\mathcal{H}$ centered at $h$, and for any $n \in \mathbb{N}$, let $S_n := \{(x_1, y_1), \ldots, (x_n, y_n)\} \sim P^n$ be a dataset. Indeed, a distribution-free upper bound for any consistent learning rule has been proved in Hanneke (2016b) (see Lemma 26 in Appendix E), which states that for any $\delta \in (0,1)$ and any $n \in \mathbb{N}$, with probability at least $1 - \delta$,

$$\sup_{h \in V_n(\mathcal{H})} er_P(h) \leq \frac{8}{n}\left(VC(\mathcal{H})\ln\left(\frac{49e\mathfrak{s}_h}{VC(\mathcal{H})} + 37\right) + 8\ln\left(\frac{6}{\delta}\right)\right). \tag{9}$$

Since $\mathcal{H}$ does not have an infinite $(d+1)$-VC-eluder sequence, and also does not have an infinite $(s+1)$-star-eluder sequence, for any infinite sequence $S := \{(x_1, y_1), (x_2, y_2), \ldots\} \sim P^{\mathbb{N}}$, there exists a (smallest) non-negative integer $k = k(S) < \infty$ such that $VC(V_k(\mathcal{H})) \leq d$ and $\mathfrak{s}_h(V_k(\mathcal{H})) \leq s$. Following a similar argument in the proof of Lemma 10, we define $\hat{\kappa} = \hat{\kappa}(P)$, a distribution-dependent quantity, to be the smallest integer such that $k(S) \leq \hat{\kappa}$ with probability at least $1/2$, and then consider the following event $\hat{\mathcal{E}}_n := \{VC(V_{\lfloor n/2\rfloor}(\mathcal{H})) \leq d, \mathfrak{s}_h(V_{\lfloor n/2\rfloor}(\mathcal{H})) \leq s\}$ with probability $\mathbb{P}(\neg\hat{\mathcal{E}}_n) \leq 2^{-\lfloor n/2\hat{\kappa}\rfloor}$. For notation simplicity, let us denote by $\epsilon_n := \frac{8}{n}(d\ln(\frac{49es}{d} + 37) + 8\ln(6))$, then conditioning on $\hat{\mathcal{E}}_n$, we have that for all $n \in \mathbb{N}$,

$$\mathbb{E}\left[\sup_{\hat{h}_n \in V_n(\mathcal{H})} er_P(\hat{h}_n)\right] = \int_0^1 \mathbb{P}\left(er_P(\hat{h}_n) > \epsilon\right) d\epsilon$$

$$\leq \int_0^1 \mathbb{P}\left(er_P(\hat{h}_n) > \epsilon \middle| \hat{\mathcal{E}}_n\right) d\epsilon + \int_0^1 \mathbb{P}\left(\neg\hat{\mathcal{E}}_n\right) d\epsilon$$

$$= \int_0^{\epsilon_n} \mathbb{P}\left(er_P(\hat{h}_n) > \epsilon \middle| \hat{\mathcal{E}}_n\right) d\epsilon + \int_{\epsilon_n}^1 \mathbb{P}\left(er_P(\hat{h}_n) > \epsilon \middle| \hat{\mathcal{E}}_n\right) d\epsilon + \int_0^1 \mathbb{P}\left(\neg\hat{\mathcal{E}}_n\right) d\epsilon$$

$$\overset{(9)}{\leq} \epsilon_n + \int_{\epsilon_n}^1 6\exp\left(\frac{d}{8}\ln\left(\frac{49es}{d} + 37\right) - \frac{n\epsilon}{64}\right) d\epsilon + 2^{-\lfloor n/2\hat{\kappa}\rfloor}$$

$$= \epsilon_n + \frac{384}{n}\exp\left(\frac{d}{8}\ln\left(\frac{49es}{d} + 37\right) - \frac{n\epsilon_n}{64}\right) + 2^{-\lfloor n/2\hat{\kappa}\rfloor}$$

$$\leq \frac{8}{n}\left(d\ln\left(\frac{49es}{d} + 37\right) + 8\ln(6)\right) + \frac{64}{n} + 2^{-\lfloor n/2\hat{\kappa}\rfloor}$$

$$\leq \frac{8d}{n}\log\left(\frac{(49e + 37)s}{d}\right) + \frac{64\ln(6) + 64}{n} + 2^{-\lfloor n/2\hat{\kappa}\rfloor} \leq \frac{160d}{n}\log\left(\frac{s}{d}\right) + 2^{-\lfloor n/2\hat{\kappa}\rfloor}.$$

To show the lower bound, let $S := \{(x_1, y_1), (x_2, y_2), \ldots\}$ be any infinite $d$-star-eluder sequence that is consistent with $\mathcal{H}$. We denote $\mathcal{X}_k := \{x_{kd-d+1}, \ldots, x_{kd}\}$ for every integer $k \geq 1$, and consider the following realizable distribution $P$ (with the same center of $S$):

$$P\{(x_{kd-d+j}, y_{kd-d+j})\} = p_k/d, \ P\{(x_{kd-d+j}, 1 - y_{kd-d+j})\} = 0, \ \forall 1 \leq j \leq d, \ \forall k \geq 1,$$

where $\{p_k\}_{k\geq 1}$ is a sequence of probabilities satisfying $\sum_{k\geq 1} p_k \leq 1$, which will be specified later. Let $S_n := \{(x_i, y_i)\}_{i=1}^n \sim P^n$ be a dataset and consider the event $\mathcal{E} := \mathcal{E}_1 \cap \mathcal{E}_2$, where

$$\mathcal{E}_1 := \{S_n \text{ does not contain a copy of any point in } \mathcal{X}_{>k}\},$$
$$\mathcal{E}_2 := \{S_n \text{ does not contain a copy of at least one point in } \mathcal{X}_k\}.$$

If the event $\mathcal{E}$ happens, the worst-case ERM can have an error rate $\mathrm{er}_P(\hat{h}_n) \geq p_k/d$. This is because $\{x_{kd-d+1}, \ldots, x_{kd}\}$ is a star set (with the same center) of $V_n(\mathcal{H}) \supseteq V_{kd-d}(\mathcal{H})$, and so we know that for any $1 \leq j \leq d$, there exists $h_{k,j} \in V_n(\mathcal{H})$ such that $h_{k,j}(x_{kd-d+j}) \neq y_{kd-d+j}$, and the ERM outputting $h_{k,j}$ will have such an error rate. The probability of $\mathcal{E}_1$ follows simply as $\mathbb{P}(\mathcal{E}_1) = (1 - \sum_{t>k} p_t)^n$. Moreover, characterizing the probability of $\mathcal{E}_2$ can be approached as an instance of the so-called *Coupon Collector's Problem*. Specifically, we let

$$\hat{n}_k := \min\{n \in \mathbb{N} : \mathcal{X}_k \subseteq S_n\},$$

which can be represented as a sum $\sum_{j=1}^{d} G_j$ of independent geometric random variables $G_j \sim$ Geometric($\frac{d+1-j}{d} p_k$) for $1 \leq j \leq d$, with the following properties

$$\begin{cases} \mathbb{E}[\hat{n}_k] = \sum_{j=1}^{d} \mathbb{E}[G_j] = \sum_{j=1}^{d} \frac{d \cdot p_k^{-1}}{d+1-j} = \frac{d}{p_k}\left(\sum_{j=1}^{d} \frac{1}{d+1-j}\right) = \frac{d}{p_k} \cdot H_d \\ \mathrm{Var}[\hat{n}_k] = \sum_{j=1}^{d} \mathrm{Var}[G_j] < \sum_{j=1}^{d} \left(\frac{d+1-j}{d} p_k\right)^{-2} < \frac{\pi^2 d^2}{6 p_k^2} \end{cases},$$

where $H_d$ is $d^{th}$ harmonic number satisfying $H_d \geq \log d$, for all $d \geq 1$. Then the standard Chebyshev's inequality implies that $\mathbb{P}(|\hat{n}_k - \mathbb{E}[\hat{n}_k]| > z) \leq \mathrm{Var}[\hat{n}_k] \cdot z^{-2}$. By choosing $z = \sqrt{2\mathrm{Var}[\hat{n}_k]}$, we have with probability at least $1/2$,

$$\hat{n}_k > \mathbb{E}[\hat{n}_k] - \sqrt{2\mathrm{Var}[\hat{n}_k]} \geq \frac{d}{p_k}\left(\log d - \frac{\pi}{\sqrt{3}}\right).$$

In particular, when $d \geq 38$, it holds that $\log d \geq 2\pi/\sqrt{3}$, and thus $\hat{n}_k > p_k^{-1} d \log d/2$, with probability at least $1/2$. Altogether, we have for any $n \leq p_k^{-1} d \log d/2$,

$$\mathbb{P}(\mathcal{E}_2) \geq \mathbb{P}(n < \hat{n}_k) \geq \mathbb{P}\left(n \leq p_k^{-1} d \log d/2, \; p_k^{-1} d \log d/2 < \hat{n}_k\right) \geq 1/2.$$

Now for all $d \geq 38$, it follows from the proceeding analysis that

$$\mathbb{P}\left(\mathrm{er}_P(\hat{h}_n) \geq \frac{p_k}{d}\right) \geq \mathbb{P}(\mathcal{E}) \geq \mathbb{P}(\mathcal{E}_1)\mathbb{P}(\mathcal{E}_2) \geq \frac{1}{2}\left(1 - \sum_{t>k} p_t\right)^n,$$

which further implies that for all $d \geq 38$,

$$\mathbb{E}\left[\mathrm{er}_P(\hat{h}_n)\right] \geq \frac{p_k}{d}\mathbb{P}\left(\mathrm{er}_P(\hat{h}_n) \geq \frac{p_k}{d}\right) \geq \frac{p_k}{2d}\left(1 - \sum_{t>k} p_t\right)^n,$$

for all $n \leq p_k^{-1} d \log d/2$. By letting $n_k = p_k^{-1} d \log d/2$, we have for all $k \in \mathbb{N}$ (infinitely many $n \in \mathbb{N}$),

$$\mathbb{E}\left[\mathrm{er}_P\left(\hat{h}_{n_k}\right)\right] \geq \frac{p_k}{2d}\left(1 - \sum_{t>k} p_t\right)^{n_k} = \frac{\log d}{4n_k}\left(1 - \sum_{t>k} p_t\right)^{\frac{d \log d}{2p_k}} \geq \frac{\log d}{4en_k},$$

where the last inequality follows from choosing probabilities $\{p_k\}_{k \geq 1}$ satisfying $\sum_{t>k} p_t \leq 1/n_k$. When $1 \leq d < 38$, the result is trivial. $\qquad\square$

## E    Technical lemmas

**Lemma 23 (Chernoff's bound).** *Let $Z_1, \ldots, Z_n$ be independent random variables in $\{0,1\}$, let $\bar{Z} := \frac{1}{n}\sum_{i=1}^{n} Z_i$. For all $t \in (0,1)$, we have*

$$\mathbb{P}\left(\bar{Z} \leq (1-t)\mathbb{E}[\bar{Z}]\right) \leq e^{-\frac{n\mathbb{E}[\bar{Z}]t^2}{2}}.$$

**Lemma 24.** *Let $\mathcal{H}$ be a concept class, and $(\kappa, \rho)$ be a stable sample compression scheme of size $\hat{n}(S_n) < n$ that is sample-consistent for $\mathcal{H}$ given data $S_n$. For any realizable distribution $P$ with respect to $\mathcal{H}$ and $S_n := \{(x_i, y_i)\}_{i=1}^{n} \sim P^n$, let $\hat{h}_n := \rho(\kappa(S_n))$. Then it holds*

$$\mathbb{E}\left[\mathrm{er}_P(\hat{h}_n)\right] \leq \frac{\mathbb{E}[\hat{n}(S_n)]}{n+1}.$$

*Proof of Lemma 24.* We claim that if $(x_{n+1}, y_{n+1})$ satisfies $\rho(\kappa(S_n))(x_{n+1}) \neq y_{n+1}$, we must have $(x_{n+1}, y_{n+1}) \in \kappa(S_{n+1})$. Suppose not, then there is a subsequence $S_n := S_{n+1} \setminus \{(x_{n+1}, y_{n+1})\}$ satisfying $\kappa(S_{n+1}) \subseteq S_n \subset S_{n+1}$ and $\rho(\kappa(S_n))(x_{n+1}) \neq y_{n+1} = \rho(\kappa(S_{n+1}))(x_{n+1})$ based on the sample-consistency of the compression scheme, which contradicts to our assumption that $(\kappa, \rho)$ is stable. Now by the exchangeability of random variables $\{x_i\}_{i \geq 1}$, we have

$$
\begin{aligned}
\mathbb{E}\left[ \mathrm{er}_P(\hat{h}_n) \right] &= \mathbb{E}_{S_n} \left[ \mathbb{P} \left\{ \rho\left(\kappa\left(S_n\right)\right)\left(x_{n+1}\right) \neq y_{n+1} \right\} \right] \\
&= \mathbb{E}_{S_{n+1}} \left[ \mathbb{1} \left\{ \rho\left(\kappa\left(S_n\right)\right)\left(x_{n+1}\right) \neq y_{n+1} \right\} \right] \\
&= \frac{1}{n+1} \sum_{i=1}^{n+1} \mathbb{E}\left[ \mathbb{1} \left\{ \rho\left(\kappa\left(S_{n+1} \setminus \{(x_i, y_i)\}\right)\right)\left(x_i\right) \neq y_i \right\} \right] \\
&\leq \frac{1}{n+1} \sum_{i=1}^{n+1} \mathbb{E}\left[ \mathbb{1} \left\{ (x_i, y_i) \in \kappa\left(S_{n+1}\right) \right\} \right] \leq \frac{\mathbb{E}[\hat{n}(S_n)]}{n+1}.
\end{aligned}
$$

$\square$

**Lemma 25** ([Hanneke and Yang](), [2015](), Lemma 44). *Let $\mathcal{H}$ be a concept class, and $P$ be a realizable distribution centered at the target $h$. For any $n \in \mathbb{N}$, let $S_n := \{(x_i, y_i)\}_{i=1}^n \sim P^n$ be a dataset, and let $\hat{\mathcal{C}}_n$ be the version space compression set (Definition 13), that is, the smallest subset of $S_n$ such that $V_{\hat{\mathcal{C}}_n}(\mathcal{H}) = V_{S_n}(\mathcal{H})$. We have $|\hat{\mathcal{C}}_n| \leq \mathfrak{s}_h$.*

*Proof of Lemma 25.* We assume that the dataset $S_n$ is consistent with the target $h$, i.e. $h(x_j) = y_j$ for all $j \in [n]$. Note that, if there exists $(x_j, y_j) \in \hat{\mathcal{C}}_n$ such that every hypothesis $g \in V_{\hat{\mathcal{C}}_n \setminus \{(x_j, y_j)\}}(\mathcal{H})$ satisfies $g(x_j) = h(x_j)$, then we have $V_{\hat{\mathcal{C}}_n \setminus \{(x_j, y_j)\}}(\mathcal{H}) = V_{\hat{\mathcal{C}}_n}(\mathcal{H}) = V_{S_n}(\mathcal{H})$, which contradicts the definition of the version space compression set as the smallest subset. Therefore, for any $(x_j, y_j) \in \hat{\mathcal{C}}_n$, there exists $g \in V_{\hat{\mathcal{C}}_n \setminus \{(x_j, y_j)\}}(\mathcal{H})$ such that $g(x_j) \neq h(x_j)$. Moreover, note that "$g \in V_{\hat{\mathcal{C}}_n \setminus \{(x_j, y_j)\}}(\mathcal{H})$" is equivalent to saying "$g(x) = y = h(x)$, for all $(x, y) \in \hat{\mathcal{C}}_n \setminus \{(x_j, y_j)\}$", which precisely matches the definition of a star set centered at $h$, that is, $\hat{\mathcal{C}}_n$ is a star set for $\mathcal{H}$ centered at $h$, witnessed by those hypotheses $g$'s. We must have $|\hat{\mathcal{C}}_n| \leq \mathfrak{s}_h$. $\square$

**Lemma 26** ([Hanneke](), [2016b](), Theorem 11). *Let $\mathcal{H}$ be a concept class, and $P$ be a realizable distribution with respect to $\mathcal{H}$ centered at $h$, let $S_n := \{(x_i, y_i)\}_{i=1}^n \sim P^n$ be a dataset, for any $n \in \mathbb{N}$. Then for any $\delta \in (0, 1)$ and any $n \in \mathbb{N}$, we have with probability at least $1 - \delta$,*

$$
\sup_{h \in V_n(\mathcal{H})} \mathrm{er}_P(h) \leq \frac{8}{n} \left( VC(\mathcal{H}) \ln\left( \frac{49 e \hat{n}_{1:n}}{VC(\mathcal{H})} + 37 \right) + 8 \ln\left( \frac{6}{\delta} \right) \right),
$$

*where the data-dependent quantity $\hat{n}_{1:n}$ is defined in Definition 13 satisfying $\hat{n}_{1:n} \leq \mathfrak{s}_h$.*

**Lemma 27** (Sauer's lemma, [Sauer](), [1972]()). *Let $\mathcal{H}$ be a concept class with $VC(\mathcal{H}) < \infty$ defined on $\mathcal{X}$ and $S_n := \{(x_i, y_i)\}_{i=1}^n \in (\mathcal{X} \times \{0, 1\})^n$. Then for all $n \in \mathbb{N}$, it holds that*

$$
\left| \mathcal{H}(S_n) \right| \leq \sum_{i=0}^{VC(\mathcal{H})} \binom{n}{i}.
$$

*In particular, if $n \geq VC(\mathcal{H})$,*

$$
\left| \mathcal{H}(S_n) \right| \leq \left( \frac{en}{VC(\mathcal{H})} \right)^{VC(\mathcal{H})}.
$$

**Lemma 28** (**VC dimension of unions**). *Let $N, T \in \mathbb{N}$ and $\mathcal{H}_1, \ldots, \mathcal{H}_N$ be concept classes with $\max_{1 \leq i \leq N} VC(\mathcal{H}_i) \leq T$, then it holds*

$$
VC\left( \bigcup_{i=1}^N \mathcal{H}_i \right) \leq 2 \log N + 4T.
$$

*Proof of Lemma 28.* According to Sauer's lemma (Lemma 27), for any $i \leq N$ and $S_n := \{(x_i, y_i)\}_{i=1}^n$ with $n \geq T$, we have

$$|\mathcal{H}_i(S_n)| \leq \sum_{i=0}^{\text{VC}(\mathcal{H}_i)} \binom{n}{i} \leq \sum_{i=0}^{T} \binom{n}{i} \leq \left(\frac{en}{T}\right)^T. \tag{10}$$

Then we can upper bound the number of possible classifications (of $S_n$) by the union $\bigcup_{i=1}^N \mathcal{H}_i$ as

$$\left|\left(\bigcup_{i=1}^N \mathcal{H}_i\right)(S_n)\right| \leq \sum_{i=1}^N |\mathcal{H}_i(S_n)| \overset{(10)}{\leq} \sum_{i=1}^N \left(\frac{en}{T}\right)^T = N\left(\frac{en}{T}\right)^T. \tag{11}$$

Let $n = \text{VC}(\bigcup_{i=1}^N \mathcal{H}_i)$ and $S_n$ be a set shattered by $\bigcup_{i=1}^N \mathcal{H}_i$, the LHS of (11) is exactly $2^n$, and thus

$$2^n \leq N\left(\frac{en}{T}\right)^T \Rightarrow n \leq \log N + T\log\left(\frac{en}{T}\right) \Rightarrow n \leq 2\log N + 4T,$$

where the last step follows from the fact that $m \leq s + q\log(em/q)$ implies $m \leq 2s + 4q$, for any $s \geq 0$ and $m \geq q \geq 1$. $\qquad\square$

**Lemma 29 (Shalev-Shwartz and Ben-David, 2014, Lemma A.1).** *Let $a > 0$, then $x \geq 2a\log a$ implies $x \geq a\log x$. Conversely, $x < a\log x$ implies $x < 2a\log a$.*

**Lemma 30 (Bousquet et al., 2021, Lemma 5.12).** *For any function $R(n) \to 0$, there exist probabilities $\{p_t\}_{t\in\mathbb{N}}$ satisfying $\sum_{t\geq 1} p_t = 1$, two increasing sequences of integers $\{n_t\}_{t\in\mathbb{N}}$ and $\{k_t\}_{t\in\mathbb{N}}$, and a constant $1/2 \leq C \leq 1$ such that the following hold for all $t \in \mathbb{N}$:*

*(1) $\sum_{k>k_t} p_k \leq \frac{1}{n_t}$.*
*(2) $n_t p_{k_t} \leq k_t$.*
*(3) $p_{k_t} = CR(n_t)$.*

**Lemma 31 (Ghost samples).** *Let $\mathcal{H}$ be a concept class and $P$ be a realizable distribution with respect to $\mathcal{H}$. Let $S_{2n} := \{(x_i, y_i)\}_{i=1}^{2n} \sim P^{2n}$, $S_n := \{(x_i, y_i)\}_{i=1}^n$ and $T_n := \{(x_i, y_i)\}_{i=n+1}^{2n}$. Then for any $\epsilon \in (0,1)$ and $n \geq 8/\epsilon$, it holds*

$$\mathbb{P}\left(\exists h \in \mathcal{H}: \hat{er}_{S_n}(h) = 0 \text{ and } er_P(h) > \epsilon\right) \leq 2\mathbb{P}\left(\exists h \in \mathcal{H}: \hat{er}_{S_n}(h) = 0 \text{ and } \hat{er}_{T_n}(h) > \epsilon/2\right).$$

*Proof of Lemma 31.* If there exists $h \in \mathcal{H}$ such that $\hat{er}_{S_n}(h) = 0$ and $er_P(h) > \epsilon$, since $T_n$ is independent of $S_n$, by applying the Chernoff's bound (Lemma 23), we have

$$\mathbb{P}\left(\hat{er}_{T_n}(h) \leq \epsilon/2 \middle| h\right) = \mathbb{P}\left(\frac{1}{n}\sum_{i=n+1}^{2n} \mathbb{1}\{h(x_i) \neq y_i\} \leq \frac{\epsilon}{2} \middle| er_P(h) > \epsilon\right) < \exp\left\{-\frac{n\epsilon}{8}\right\}.$$

Then for any $n \geq 8/\epsilon$, it follows

$$\mathbb{P}\left(\hat{er}_{T_n}(h) > \epsilon/2 \middle| h\right) = 1 - \mathbb{P}\left(\hat{er}_{T_n}(h) \leq \epsilon/2 \middle| h\right) > 1 - \exp\left\{-\frac{n\epsilon}{8}\right\} \geq \frac{1}{2},$$

which completes the proof. $\qquad\square$

**Lemma 32 (Random swaps).** *Let $\mathcal{H}$ be a concept class with $VC(\mathcal{H}) < \infty$ and $P$ be a realizable distribution with respect to $\mathcal{H}$. Let $S_{2n} := \{(x_i, y_i)\}_{i=1}^{2n} \sim P^{2n}$, $S_n := \{(x_i, y_i)\}_{i=1}^n$ and $T_n := \{(x_i, y_i)\}_{i=n+1}^{2n}$. Then for any $\epsilon \in (0,1)$ and $n \geq VC(\mathcal{H})/2$, it holds*

$$\mathbb{P}\left(\exists h \in \mathcal{H}: \hat{er}_{S_n}(h) = 0 \text{ and } \hat{er}_{T_n}(h) > \epsilon/2\right) \leq \left(\frac{2en}{VC(\mathcal{H})}\right)^{VC(\mathcal{H})} 2^{-n\epsilon/2}.$$

*Proof of Lemma 32.* We prove the lemma by using the "random swaps" technique. Specifically, we define $\sigma_1, \ldots, \sigma_n$ to be independent random variables with $\sigma_i \sim \text{Unif}(\{i, n+i\})$ for all $1 \leq i \leq n$, which are also independent of $S_{2n}$. For notation simplicity, we denote by $\sigma_{n+i}$ to be the remaining element in $\{i, n+i\} \setminus \{\sigma_i\}$. Now we let $S_\sigma := \{(x_{\sigma_1}, y_{\sigma_1}), \ldots, (x_{\sigma_n}, y_{\sigma_n})\}$ and

$T_\sigma := \{(x_{\sigma_{n+1}}, y_{\sigma_{n+1}}), \ldots, (x_{\sigma_{2n}}, y_{\sigma_{2n}})\}$, and note that $S_\sigma \cup T_\sigma$ follows the same distribution as $S_{2n}$. Hence, we have

$$
\begin{aligned}
&\mathbb{P}\left(\exists h \in \mathcal{H} : \hat{\mathrm{er}}_{S_n}(h) = 0 \text{ and } \hat{\mathrm{er}}_{T_n}(h) > \epsilon/2\right) \\
=\ &\mathbb{P}\left(\exists h \in \mathcal{H} : \hat{\mathrm{er}}_{S_\sigma}(h) = 0 \text{ and } \hat{\mathrm{er}}_{T_\sigma}(h) > \epsilon/2\right) \\
=\ &\mathbb{P}\left(\exists (Y_1, \ldots, Y_{2n}) \in \mathcal{H}(S_{2n}) : \begin{array}{l} \frac{1}{n}\sum_{i=1}^{n} \mathbb{1}\left\{y_{\sigma_i} \neq Y_{\sigma_i}\right\} = 0, \\ \frac{1}{n}\sum_{i=1}^{n} \mathbb{1}\left\{y_{\sigma_{n+i}} \neq Y_{\sigma_{n+i}}\right\} > \epsilon/2 \end{array}\right) \\
\overset{\text{LoTP}}{=}\ &\mathbb{E}\left[\mathbb{P}\left(\exists (Y_1, \ldots, Y_{2n}) \in \mathcal{H}(S_{2n}) : \begin{array}{l} \frac{1}{n}\sum_{i=1}^{n} \mathbb{1}\left\{y_{\sigma_i} \neq Y_{\sigma_i}\right\} = 0, \\ \frac{1}{n}\sum_{i=1}^{n} \mathbb{1}\left\{y_{\sigma_{n+i}} \neq Y_{\sigma_{n+i}}\right\} > \epsilon/2 \end{array} \,\middle|\, S_{2n}\right)\right] \\
\overset{\text{Union bound}}{\leq}\ &\mathbb{E}\left[\sum_{(Y_1, \ldots, Y_{2n}) \in \mathcal{H}(S_{2n})} \mathbb{P}\left( \begin{array}{l} \frac{1}{n}\sum_{i=1}^{n} \mathbb{1}\left\{y_{\sigma_i} \neq Y_{\sigma_i}\right\} = 0, \\ \frac{1}{n}\sum_{i=1}^{n} \mathbb{1}\left\{y_{\sigma_{n+i}} \neq Y_{\sigma_{n+i}}\right\} > \epsilon/2 \end{array} \,\middle|\, S_{2n}\right)\right].
\end{aligned}
\tag{12}
$$

Next, we consider that given $S_{2n}$, how possibly that the following event happens

$$
\mathcal{E}_Y := \left\{ \frac{1}{n}\sum_{i=1}^{n} \mathbb{1}\left\{y_{\sigma_i} \neq Y_{\sigma_i}\right\} = 0 \text{ and } \frac{1}{n}\sum_{i=1}^{n} \mathbb{1}\left\{y_{\sigma_{n+i}} \neq Y_{\sigma_{n+i}}\right\} > \frac{\epsilon}{2} \right\},
$$

for a given labeling $Y := (Y_1, \ldots, Y_{2n}) \in \mathcal{H}(S_{2n})$. Indeed, if $\mathcal{E}_Y$ happens, then there must exist at least $\lceil n\epsilon/2 \rceil$ indices $i \leq n$ such that either $y_i = Y_i, y_{n+i} \neq Y_{n+i}$ or $y_i \neq Y_i, y_{n+i} = Y_{n+i}$, otherwise, the difference between $\frac{1}{n}\sum_{i=1}^{n} \mathbb{1}\{y_{\sigma_i} \neq Y_{\sigma_i}\}$ and $\frac{1}{n}\sum_{i=1}^{n} \mathbb{1}\{y_{\sigma_{n+i}} \neq Y_{\sigma_{n+i}}\}$ is less than $\epsilon/2$. Based on this and the distribution of $\sigma_i$'s, we have

$$
\mathbb{P}\left( \begin{array}{l} \frac{1}{n}\sum_{i=1}^{n} \mathbb{1}\left\{y_{\sigma_i} \neq Y_{\sigma_i}\right\} = 0, \\ \frac{1}{n}\sum_{i=1}^{n} \mathbb{1}\left\{y_{\sigma_{n+i}} \neq Y_{\sigma_{n+i}}\right\} > \epsilon/2 \end{array} \,\middle|\, S_{2n}\right) \leq 2^{-\lceil \frac{n\epsilon}{2} \rceil}.
$$

Plugging into (12), we finally get for all $n \geq \mathrm{VC}(\mathcal{H})/2$,

$$
\begin{aligned}
\mathbb{P}\left(\exists h \in \mathcal{H} : \hat{\mathrm{er}}_{S_n}(h) = 0 \text{ and } \hat{\mathrm{er}}_{T_n}(h) > \epsilon/2\right) \ &\leq\ \mathbb{E}\left[\sum_{(Y_1, \ldots, Y_{2n}) \in \mathcal{H}(S_{2n})} 2^{-\lceil \frac{n\epsilon}{2} \rceil}\right] \\
&\leq\ \mathbb{E}\left[\mathcal{H}(S_{2n})\right] \cdot 2^{-n\epsilon/2} \\
&\overset{\text{Lemma } 27}{\leq}\ \left(\frac{2en}{\mathrm{VC}(\mathcal{H})}\right)^{\mathrm{VC}(\mathcal{H})} \cdot 2^{-n\epsilon/2}.
\end{aligned}
$$

$\square$

