# OpenReview forum: "Universal Rates of Empirical Risk Minimization"
_NeurIPS.cc/2024/Conference — NeurIPS 2024 poster_

### Official Review · Reviewer_NPxP · 2024-07-10

**Soundness:** 4
**Presentation:** 4
**Contribution:** 3
**Rating:** 9
**Confidence:** 4

**Summary:**

In this paper, the authors build on earlier work in learning theory that provides an alternative to the classical PAC theory in the setting of realizable binary classification.  That earlier work considered *universal* learning rates, where a rate is universal if the data source is fixed across sample sizes (as opposed to being allowed to depend on the sample size &n&) and demonstrated a trichotomy in the possible rates of learning.  The present work considers a similar setting, where the data distribution is fixed and $n$ increases, but analyzes only the natural algorithm ERM.  As opposed to the earlier setting where arbitrary algorithms are allowed, the universal learning rates of ERM satisfy a tetrachotomy, depending on several technical combinatorial notions of complexity of the hypothesis class.  The paper complements their theoretical results with a number of concrete examples of hypothesis classes that fall into different cases of the tetrachotomy as well as comparing these cases to the universal rates attainable with more general algorithms than ERM.

**Strengths:**

The paper presents interesting and valuable contributions to statistical learning theory, following in the footsteps of an earlier work that seeks to understand learning beyond the classical PAC setting.  The algorithm considered, ERM, is classical and the standard approach to virtually all supervised learning problems and thus is an important setting to understand, hewing closer to practice than the more involved algorithms of the earlier work to study universal rates.  Furthermore, the presentation is excellent, with a number of helpful concrete examples included for grounding and clear exposition on the technical combinatorial details.

**Weaknesses:**

The primary weakness of the work is the restrictive assumption of realizability, as well as the focus purely on binary classification.  In the setting studied, however, the paper does a good job of essentially completely answering the question of the universal rates.  There is a minor gap in the combinatorial characterization of when rates $1/n$ vs $\log(n)/n$ are expected, but this limitation is adequately discussed in Remark 3.

**Questions:**

1. Are the algorithms that Bousquet et al 2021 use to demonstrate universal learning rates in the absence of computational considerations oracle-efficient with respect to an ERM oracle?  If not, it might be good to emphasize this point.  If so, what are the computational advantages of considering ERM?

2. A minor typo: in example 6, the VC dimension of halfspaces is $d+1$ not $d$.

3. In the conclusion, you suggest the study of "best-case ERM," but I am confused as to what that means?

**Limitations:**

The authors have adequately addressed the limitations.

---

> ### Author Rebuttal · Authors · 2024-08-06
>
> Thank you for your encouraging comments and insightful suggestions on our paper. Below, we provide detailed responses to your concerns and questions:
>
> 1. For the assumption of realizability, as is traditional in the learning theory literature, we first focus on the realizable case to build insights due to its simplicity, and the next natural step is to extend to the non-realizable (agnostic) setting.
> 2. The algorithm of [Bousquet et al., 2021] is not expressed as a reduction-to-ERM, and it is really unclear whether such a reduction is possible. Their algorithm is expressed in terms of an extension of Littlestone's SOA algorithm allowing for infinite ordinal values of the Littlestone dimension. Moreover, for finite Littlestone dimension, the recent work of [Assos, Attias, Dagan, Daskalakis, \& Fishelson, 2023] have found that online learning of Littlestone classes is achievable via a reduction to ERM, but this reduction relies on the relation between the Littlestone dimension and threshold dimension, a relation which breaks down for infinite ordinal Littlestone dimensions, since thresholds on the natural numbers do not admit an infinite Littlestone tree.
> 3. Thank you for pointing out this typo. The VC dimension of halfspaces is d+1.
> 4. To get an intuition of the so-called "best-case ERM", consider the following alternative to our Definition 2: for the upper bound, "for every ERM algorithm" $\rightarrow$ "there exists an ERM algorithm"; and for the lower bound, "there exists an ERM algorithm" $\rightarrow$ "for every ERM algorithm". In particular for the realizable case, studying the "best-case ERM" is equivalent to studying the learning rates of general proper learners.

---

> > ### Comment · Reviewer_NPxP · 2024-08-10
> > **Thank you for the response**
> >
> > Thank you for the clarifying points, especially on the reduction to ERM of prior work.  I would encourage you to include this discussion in the paper.  I maintain my (quite high) score.  Great work!

---

### Official Review · Reviewer_qCB2 · 2024-07-11

**Soundness:** 3
**Presentation:** 3
**Contribution:** 3
**Rating:** 5
**Confidence:** 1

**Summary:**

This paper studies the performance of ERM on realizable binary classification problems in the "Universal learning" framework of [1]. Specifically, the original work [1] showed that the optimal universal rate of convergence was in general not achieved by ERM procedures. Nevertheless, characterizing the universal rate of convergence of ERM procedures as a function of the hypothesis class in this setting is still interesting as these procedures are widely used in practice.

This paper tackles this question and shows that if a hypothesis class $\mathcal{H}$ is universally learnable by ERM procedures, then the universal rate of convergence of the worst ERM procedure is either $\exp(-n)$, $1/n$, $\log(n)/n$ or arbitrarily slow. Necessary and sufficient conditions in terms of properties of $\mathcal{H}$ are given that indicate which of these cases occurs. A second result provides a more refined such characterization as a function of the target function $h_{*} \in \mathcal{H}$ as well.



[1]: Bousquet, O., Hanneke, S., Moran, S., Van Handel, R., and Yehudayoff, A. (2021), “A theory of
universal learning,” in Proceedings of the 53rd Annual ACM SIGACT Symposium on Theory of
Computing, pp. 532–541.

**Strengths:**

+ The problem studied as well as the results are interesting.

**Weaknesses:**

+ The writing of the paper is at times uncomfortably close to that [1].

**Questions:**

minor comments:
+ lines 69-70: "unlike that" -> "while".
+ line 72: "to the characterization of" -> "characterizing".
+ line 82: "necessary" -> "necessarily".
+ line 107: "exist" -> "exists".
+ line 108: "it requires us" -> "we need"

**Limitations:**

A main limitation of this work is that it does not characterize when a class is universally learnable by ERM. This is mentionned in lines 147-150. More discussion about why this is difficult to achieve would enhance the paper.

---

> ### Author Rebuttal · Authors · 2024-08-06
>
> Thank you for your comments on our paper. We are happy to know that you found the problem and our results interesting.
>
> 1. For the weakness, we adopted the same universal learning framework of the work of [Bousquet et al., 2021] as well as some related definitions. However, our theory differs a lot from theirs, including but not limited to a different problem setting (we focus on ERM learners), different possible universal rates, new-developed combinatorial structures and plenty of concrete examples for nuanced analysis.
> 2. For the questions, we appreciate your careful reading. These are very fair comments on grammar, allowing us to enhance the quality of our manuscript. We will go through the paper to correct other potential errors.
> 3. For the limitation you mentioned, we would like to extend our sincere thanks to you for pointing out areas where our paper could be improved. In our manuscript, we left the universal learnability by ERM an open question for future work, which is not within the scope of this work. One might guess that the universal Glivenko-Cantelli property might be the correct characterization but it turns out that it is not (see Example 5). We suspect such a characterization would be of a significantly different nature compared to the combinatorial structures we find for the characterizing the rates. For instance, note that the set $\mathcal{H}$ of all concepts with finitely-many 1's or finitely-many 0's is learnable by ERM (albeit at arbitrarily slow rates) if $\mathcal{X}$ is countable but not if $\mathcal{X}$ is the real line. These cases won't be distinguished by a discrete combinatorial structure.

---

> > ### Comment · Reviewer_qCB2 · 2024-08-10
> >
> > I thank the authors for their response. I would like to emphasis that my evaluation is an educated guess, I do not know enough about the work [1] to fairly judge the current work.

---

### Official Review · Reviewer_MghQ · 2024-07-11

**Soundness:** 4
**Presentation:** 4
**Contribution:** 3
**Rating:** 6
**Confidence:** 3

**Summary:**

The main goal of the proposed work is to understand the learning procedure with a focus on Empirical risk minimization. The authors claim to provide a complete picture of the four possibilities of different learning rates by ERM. The work also introduces many new concepts related to combinatorial dimensions.

**Strengths:**

The paper is extremely well-written. I appreciate the effort put in by the authors to provide such a clear exposition. One of this paper's key strengths is exploring and finally characterizing all possible universal learning rates by ERM for the first time. I particularly like the detailed picture featuring the dichotomies between several learning rates with particular examples.

**Weaknesses:**

The paper is overall an enjoyable read, but I am not entirely sure how well it fits NeurIPS since the work is mostly theoretical and centred around learning theory. I believe the paper would be more suitable for a venue entirely dedicated to learning theory, such as COLT.

**Questions:**

Can the authors explain what do they mean by "learning scenarios such as the agnostic case" in l421?

---

> ### Author Rebuttal · Authors · 2024-08-06
>
> Thank you for your insightful comments and positive remarks on our paper.
>
> For the weakness you mentioned, we would like to point out that, in addition to numerous practical works, the interdisciplinary NeurIPS conference also has a rich history of many seminal purely-theoretical works in learning theory.
>
> Here is a response for your question. Mathematically, the agnostic case stands for the situation $\inf_{h\in\mathcal{H}}\text{er}_{P}(h)>0$, which is more practical than realizable case since we usually have no knowledge about the ground-truth model in real-world applications.

---

> > ### Comment · Reviewer_MghQ · 2024-08-09
> >
> > I thank the authors for their explanation.

---

### Official Review · Reviewer_EqSd · 2024-07-12

**Soundness:** 3
**Presentation:** 3
**Contribution:** 4
**Rating:** 7
**Confidence:** 2

**Summary:**

The work contributes to a recent line of work on universal learning rates, i.e. rates with distribution-dependent constants. It characterizes the universal learning rates for the ERM learning rule (specially, the "worst-case" version thereof), showing a partition into four possible optimal universal ERM rates -- which of these rates the "worst-case" ERM follows determined by combinatorial measures on the hypothesis class.

**Strengths:**

Given the ubiquity of ERM, and the looseness of the more classical minimax analyses, the work makes a clear contribution towards the understanding of learning in practice. The work is comprehensive, points out interesting future directions and potential improvements to their theory (e.g. the potential for a simple combinatorial measure separating the $1/n$ and $\log(n)/n$ regimes).

**Weaknesses:**

While the examples of different rates in Section 1.1 are expanded on mathematically in the Appendix, they don't add much as written in the body and could use a bit more intuitive framing. The presentation of section 3 feels a bit pedantic at items.

**Questions:**

What does a `"bad ERM algorithm" (line 135) look like? Random selection from the version space?

Do you have any conjectures as to if and where the rates for ``best-case'' ERM differ from the rates presented here?

**Limitations:**

Yes

---

> ### Author Rebuttal · Authors · 2024-08-06
>
> Thank you for your insightful feedback and helpful suggestions on our paper. Below, we provide detailed responses to your comments and questions:
>
> 1. For the weakness, we will definitely try to improve the writing and add intuitive explanations in the final version.
> 2. To answer the first question: the notion of a "worst-case ERM" arises because of the non-uniqueness of the minimizer of the 0-1 loss in classification, i.e., there may be many different ways of breaking this tie, and our analysis aims to provide a rate that holds for all such choices. The notion of "random selection" requires first that there is some well-defined measure on the concept class, and such an idea may require a different analysis. In particular for the Example 4 that you pointed out, we actually give out the explicit form of a bad (the worst) ERM algorithm in Appendix B.1 (see Example 11, lines 584-585), which is a deterministic algorithm.
> 3. For the second question, in particular for the realizable case, studying the ``best-case ERM" is equivalent to studying the optimal rates of general "proper learners", which is a fascinating and important question for future work. We suspect the categorization of rates will somewhat differ from those we establish for the worst-case ERM, analogously to the uniform analysis in [Bousquet, Hanneke, Moran, \& Zhivotovskiy, 2020].

---

> > ### Comment · Reviewer_EqSd · 2024-08-10
> >
> > Thanks for the reply and the pointer to Appendix B.1.

---

### Official Review · Reviewer_q1H2 · 2024-07-23

**Soundness:** 3
**Presentation:** 3
**Contribution:** 3
**Rating:** 7
**Confidence:** 4

**Summary:**

This paper studies the universal rates for ERM learners in the realizable case. While a complete characterization of universal rates for PAC has been studied, it was previously not clear what are the universal rates for the popular ERM learners. This papers presents a tetrachotomy for the ERM learners. In doing so, the authors also develop some new combinatorial complexity measures.

**Strengths:**

This paper presents a tetrachotomy for universal rates of the ERM learners: there are only four possible learning rates, i.e., e^{-n}, 1/n, \log(n)/n, or arbitrarily slow. The authors also provide results for target specified universal rates and introduce some new complexity measures.

**Weaknesses:**

It seems that lot of techniques used in this paper were borrowed from "A Theory of Universal Learning" by Bousquet et al. Can authors highlight what are the main difficulties of proving universal rates for ERM learners (compared to PAC learners), and what are new techniques developed for that?

Also, it's not clear to me what is the "worst-case" ERM algorithm (line 101). Can author elaborate on that?

**Questions:**

See above.

---

> ### Author Rebuttal · Authors · 2024-08-06
>
> Thank you for your insightful feedback on our paper. Below, we provide detailed responses to your questions:
>
> 1. ERM learners are quite different from the designed optimal learners constructed in [Bousquet et al., 2021]. Our analysis reveals a completely different characterization of when ERM learners achieve each of the possible rates, and indeed also reveals a $\log(n)/n$ rate is possible for ERM learners, which does not appear as an optimal universal rate. The main difficulty in establishing our theory comes from identifying the correct characterizations, which are different from those characterize the optimal universal rates. As for the proofs, most of them are actually built on the classic PAC theory of the ERM principle, together with arguments connecting combinatorial structures from PAC theory to appropriate infinite structures, such as eluder sequences, star-eluder sequences, and VC-eluder sequences, instead of the techniques in [Bousquet et al., 2021]. Moreover, the universal learning by ERM has many nuanced cases (those constructed examples in the paper) that we have to consider.
>
> 2. The notion of a "worst-case ERM" arises because of the non-uniqueness of the minimizer of the 0-1 loss in classification, i.e., there may be many different ways of breaking this tie, and our analysis aims to provide a rate that holds for all such choices. The so-called "worst-case ERM" is actually reflected in our Definition 2. Note that for the upper bound, if the worst-case ERM can achieve some learning rate, then every ERM algorithm can achieve that rate as well. For the lower bound, if there exists an ERM algorithm that fail to achieve some learning rate, then it must be that this is also true for the worst-case ERM. Another way to define the worst-case ERM is presented in Remark 2, i.e., $\sup_{h\in V_{n}(\mathcal{H})}\text{er}_{P}(h)$, which reflects the error rate of the worst-case ERM. We would like to emphasize that the classic PAC theory of ERM also considers the worst-case ERM, one of the equivalences in the Fundamental Theorem of Statistical Learning is stated as "Any ERM rule is a successful PAC learner for $\mathcal{H}$", which is essentially the same as saying "The worst-case ERM rule is a successful PAC learner for $\mathcal{H}$".

---

> > ### Comment · Reviewer_q1H2 · 2024-08-09
> >
> > Thank you for your reply. I have increased my score to 7.

---

### Decision · Program_Chairs · 2024-09-25

**Decision:**

Accept (poster)

**Comment:**

This paper applies the universal learning framework to ERM learners for the realizable case. Instead of 3 cases, ERM learners can have 4 different rates. The majority of the reviewers think this is a good quality work and therefore I recommend acceptance.